# High-dimensional Analysis of Synthetic Data Selection

**Parham Rezaei**[1]**, Filip Kovačević**[1]**, Francesco Locatello**[1][*]**, Marco Mondelli**[1][*]
[1] Institute of Science and Technology Austria (ISTA)
`parhamix@gmail.com, filip.kovacevic@ist.ac.at`
`francesco.locatello@ist.ac.at, marco.mondelli@ist.ac.at`

## Abstract

Despite the progress in the development of generative models, their usefulness in creating synthetic data that improve prediction performance of classifiers has been put into question. Besides heuristic principles such as "synthetic data should be close to the real data distribution", it is actually not clear which specific properties affect the generalization error. Our paper addresses this question through the lens of high-dimensional regression. Theoretically, we show that, for linear models, the *covariance shift* between the target distribution and the distribution of the synthetic data affects the generalization error but, surprisingly, the mean shift does not. Furthermore we prove that, in some settings, matching the covariance of the target distribution is optimal. Remarkably, the theoretical insights from linear models carry over to deep neural networks and generative models. We empirically demonstrate that the *covariance matching* procedure (matching the covariance of the synthetic data with that of the data coming from the target distribution) performs well against several recent approaches for synthetic data selection, across training paradigms, architectures, datasets and generative models used for augmentation.

## 1 Introduction

The controllable generation of arbitrary amounts of synthetic data for training machine learning models has long been considered as one of the key implications unlocked by more capable generative models (Kingma & Welling, 2014; Goodfellow et al., 2014; Shrivastava et al., 2017; Nikolenko et al., 2021). After all, synthetic data can not only be abundant, which would already be tremendously impactful in data-scarce applications such as medicine (Esteban et al., 2017; van Breugel et al., 2024), but it can also address other difficulties of observational data, such as privacy (Jordon et al., 2018), imbalancedness (Parihar et al., 2024; Ramaswamy et al., 2021) and overall difficulty to collect, as the domain can be specific (Dunlap et al., 2023) or the task complex (Wang et al., 2023). At the same time, while generative models have progressed significantly, experimental results are still mixed. Several works are promising (Trabucco et al., 2024; He et al., 2023; Azizi et al., 2023; Dunlap et al., 2023), steering and sometimes filtering the sampling by appropriately conditioning a generative model towards the target training distribution; others outright question whether synthetic data has any advantage over simply selecting some more data which is anyway used to train the generative model (Fan et al., 2024; Burg et al., 2023; Geng et al., 2024); some even warn that training on synthetic data may not only do worse, but also lead to unwanted effects such as model collapse (Shumailov et al., 2024) or additional bias (Wyllie et al., 2024). What emerges here is a broad challenge which consists of understanding *how extra synthetic data, for example from a generative model, helps training predictors*. Our paper tackles this challenge theoretically and empirically.

To do so, we assume access to a training dataset $(X_t, y_t)$ with i.i.d. samples, as well as to an additional *synthetic dataset* $(X_s, y_s)$. The samples from the synthetic dataset are also i.i.d., but they come from a different distribution, since they are obtained from a generative model and not from the training dataset. We perform empirical risk minimization (ERM) using the augmentation $((X_t, X_s), (y_t, y_s))$, and evaluate the performance on an independent test sample $(X_{\text{test}}, y_{\text{test}})$ with the same distribution as $(X_t, y_t)$. In this context, the challenge above leads to the following concrete question:

---

[*]Equal advising

$$\text{How to select the dataset } (X_s, y_s) \text{ in order to minimize the test error?} \tag{Q}$$

By studying this question, we can identify which properties of the distribution of $(X_s, y_s)$ improve generalization, thus guiding the selection of data obtained in practice from generative models.

**Formalization of the problem.** Let us first describe how we model the setting in the theoretical analysis. We assume that the distributions of both the original training dataset and the additional synthetic one are mixture models. The number of mixtures corresponds to the number of classes in the datasets, with each mixture component corresponding to a single class. As common in practice (Burg et al., 2023), the data augmentation via the synthetic dataset occurs class-by-class: for a problem with $K$ classes, the number of mixtures is $K$ and we add synthetic data of each class using a generative model.

We then address the question (Q) when $(X_t, y_t)$ and $(X_s, y_s)$ correspond to a single class, focusing on linear models and high-dimensional ridgeless regression. More precisely, we model $y_t = X_t\beta + \varepsilon_t$ and $y_s = X_s\beta + \varepsilon_s$, where rows of $X_t$ are i.i.d. with mean $\mu_t$ and covariance $\Sigma_t$, rows of $X_s$ are i.i.d. with mean $\mu_s$ and covariance $\Sigma_s$, and entries of $\varepsilon_t, \varepsilon_s$ are i.i.d. with zero mean and variance $\sigma^2$. Here, the difference between the distributions of $(X_t, y_t)$ and $(X_s, y_s)$ is captured by the mean shift $\mu_t \neq \mu_s$ and the covariance shift $\Sigma_t \neq \Sigma_s$. We also consider model shift (different $\beta$ between synthetic and real samples), deferring the details to Appendix B. Our formalization deals with a single class in isolation, fitting a regression model to the class label and neglecting interactions between classes. While this is a strong assumption chosen for mathematical tractability, we highlight that the resulting data selection procedure is extensively tested in practical settings where it performs well against existing baselines.

**Main contributions.** The *surprising* finding from our theoretical analysis is that, while the covariance shift affects the test error, *the mean shift does not*. This is the case as long as the training dataset $(X_t, y_t)$ is not too small compared to the synthetic dataset $(X_s, y_s)$, and it is especially surprising since the mean shift does affect the test error when using only synthetic data. From this insight, we show that the problem of selecting $(X_s, y_s)$ can be reduced to an optimization problem over the covariance $\Sigma_s$ and, in some settings, *matching the covariances* ($\Sigma_s \propto \Sigma_t$) leads to optimal performance. Most importantly, these theoretical insights are valid in practice: matching the covariance, without worrying about the mean shift, performs on par—or even outperforms—several recent approaches for synthetic data selection. We summarize our contributions below:

- We give a precise characterization of the test error of the min-norm least squares regression estimator, when the dimensions of $\beta, y_t, y_s$ are all large and scale proportionally. Our results hold in under-parameterized (Theorem 4.1) and over-parameterized regimes (Theorem 4.4), showing that the test error approaches a deterministic quantity that depends *only on the covariances* $\Sigma_t, \Sigma_s$ and *not on the means* $\mu_t, \mu_s$. As a comparison, we also analyze training only over synthetic data, showing that in this case the test error depends on both covariances $\Sigma_t, \Sigma_s$ and means $\mu_t, \mu_s$, see Proposition 4.2.

- Our characterization implies that we can select synthetic data minimizing the test error based on their covariance. We then show that, under some conditions, taking $\Sigma^s \propto \Sigma^t$, i.e., *covariance matching*, is optimal (Theorems 4.3 and 4.5 for under-parameterized and over-parameterized regimes).

- We validate the effectiveness of covariance matching as a way to select synthetic data obtained from generative models in several practical scenarios. We show that this simple approach performs on par—and, actually, it often outperforms—a variety of baselines proposed in the recent literature (He et al., 2023; Lin et al., 2023; Hulkund et al., 2025). This conclusion consistently holds across training paradigms (training from scratch, distilling a bigger model, fine-tuning a model trained from a larger dataset), across architectures (ResNets, transformers), across datasets (CIFAR-10, ImageNet-100, RxRx1), and across generative models used to obtain synthetic data (StyleGAN2-Ada, SANA1.5, PixArt-$\alpha$, StableDiffusion1.4, MorphGen).

## 2 RELATED WORK

**On the theoretical side**, we focus on the high-dimensional regime in which both the number of features (i.e., dimension of $\beta$) and the number of samples (i.e., dimensions of $y_s, y_t$) are large and scale proportionally. This setup was considered by a line of research using random matrix theory to characterize test error and various associated phenomena (e.g., benign overfitting (Bartlett et al., 2020) and double descent (Belkin et al., 2019)). More precisely, the test error of ridge(less) regression was studied by Hastie et al. (2022); Wu & Xu (2020); Richards et al. (2021); Cheng & Montanari (2024),

the distribution of the ERM solution by Montanari et al. (2019); Chang et al. (2021); Han & Xu (2023), and the impact of spurious correlations by Bombari & Mondelli (2025). This motivates us to look for practical insights into synthetic data selection by performing a high-dimensional regression analysis. Closer to our work are specific analyses involving more than one distribution, which in our case are the training/test distribution and the synthetic one used for augmentation. More precisely, the test error under distribution shift was analyzed by Patil et al. (2024); Mallinar et al. (2024), but this assumes training on one distribution and testing on the other, as opposed to training on both and testing on one. Training on surrogate data was considered by Ildiz et al. (2025); Kolossov et al. (2024); Jain et al. (2024): Ildiz et al. (2025) assume that the surrogate data comes from a teacher model and study the phenomenon of weak-to-strong generalization; Kolossov et al. (2024) consider data selection given unlabeled samples plus access to a surrogate model that predicts the labels better than random guessing; Jain et al. (2024) integrate surrogate and real data, but the analysis is limited to isotropic covariance. Most closely related to our theoretical setting is when training occurs on multiple data distributions and testing occurs on a single one of them, which was analyzed both in under-parameterized (Yang et al., 2025) and over-parameterized (Song et al., 2024) regimes. However, Yang et al. (2025); Song et al. (2024) assume that the data distributions have zero mean, which is unrealistic in our context. In fact, centering the data would require access to the mean of the test sample, which is equivalent to having access to its unknown label. Related work by El Firdoussi et al. (2025) analyzes high-dimensional binary classification with isotropic covariance, with a different objective from ours. Namely, its goal is to understand the factors influencing performance when generating synthetic data using estimated statistics from real data, pointing to degrading performance in case of bad covariance estimation. Additional related works by Seddik et al. (2024); Bertrand et al. (2024) consider distribution shift of a mix of real and synthetic data through the lens of statistical approximation error, a mechanism linked to model collapse by Shumailov et al. (2024). Both Seddik et al. (2024) and Bertrand et al. (2024) examine the phenomenon within iterative retraining loops, considering a setting complementary to the one of this paper.

**On the practical side**, several papers studied how to incorporate synthetic data into training predictors. Besides simply training better generative models, empirical work focused on upgrading the sampling process itself, under the assumption that better conditional generation would lead to more accurate predictors. More precisely, the CLIP model (Radford et al., 2021) underpins many filtering and selection algorithms for generative data. He et al. (2023) propose using CLIP similarity to labels to prune low-quality samples from augmentations. Lin et al. (2023) introduce sampling and filtering strategies based on CLIP similarity to either labels or the mean representation of real data, incorporating diversity via clustering. Almost concurrently, other works argued that synthetic images underperform in scaling laws (Fan et al., 2024) and, if the generative model is pre-trained on external data, simple retrieval baselines can be better (Geng et al., 2024; Burg et al., 2023). Our work can be interpreted as a more fine-grained investigation of the same problem, characterizing which properties of the generated data improve generalization. At the same time, our results do not preclude that the extra data is real data from another dataset, as tested in Figure 2 in Appendix C. Closer to our solution, Hulkund et al. (2025) explore the problem of data selection given a fixed test set and, taking a purely empirical stance, compare several filtering methods, including an approach inspired by Gadre et al. (2023) that selects clusters of image embeddings. As a heuristic, we find that this works rather well but has shortcomings, as empirically demonstrated in Table 5 in Appendix C.

## 3 PRELIMINARIES

**Data model.** We consider data augmentation in the context of linear models. Formally, we observe two datasets $(X_t, y_t)$ and $(X_s, y_s)$, denoting training data and augmenting synthetic data, such that
$$y_{(i)} = X_{(i)}\beta + \varepsilon_{(i)}, \qquad (i) \in \{t, s\}, \tag{3.1}$$
where $X_{(i)} \in \mathbb{R}^{n_{(i)} \times p}$, $\beta \in \mathbb{R}^p$, and $\varepsilon_{(i)} \in \mathbb{R}^{n_{(i)}}$. Thus, we are given $n_t$ training samples and $n_s$ synthetic samples, all of which are $p$ dimensional. We denote the total number of samples as $n := n_t + n_s$. Each entry of the noise vector $\varepsilon_{(i)}$ is sampled i.i.d. from a random variable with mean zero and variance $\sigma^2$. The row vectors of $X_{(i)}$, for $(i) \in \{t, s\}$, are independent random vectors with $p \times p$ population covariance matrix $\Sigma_{(i)}$ and mean $\mu_{(i)}$. This can be written as:
$$X_{(i)} = Z^{(i)}(\Sigma_{(i)})^{1/2} + 1_{n_{(i)}}\mu_{(i)}^{\top} \in \mathbb{R}^{n_i \times p}, \tag{3.2}$$
where $Z^{(i)} \in \mathbb{R}^{n_{(i)} \times p}, \mu_{(i)} \in \mathbb{R}^p$, $1_{n_i} \in \mathbb{R}^{n_i}$ is the all-ones vector, and all entries $[Z_{jk}^{(i)}]$ are independent with zero mean and unit variance. By omitting subscripts, we denote by $(X, y)$ the two

datasets $(X_t, y_t)$ and $(X_s, y_s)$ stacked, i.e., $X := \begin{bmatrix} X_t \\ X_s \end{bmatrix} \in \mathbb{R}^{n \times p}$, $y := \begin{bmatrix} y_t \\ y_s \end{bmatrix} \in \mathbb{R}^n$. The vector $\beta$ is assumed to be the same for $(X_t, y_t)$ and $(X_s, y_s)$, which corresponds to assuming that the conditional distribution of the labels $y$ given the features $X$ is the same for training and synthetic data.

Note that we also consider a data model in which model shift is present, i.e. training and synthetic data have different hidden parameters $\beta_t$ and $\beta_s$, respectively. For more details, see Appendix B.

**Assumptions.** We make some assumptions on the data distribution which are common in related work (Yang et al., 2025; Song et al., 2024). Let $\tau > 0$ be a small constant. We assume that, for $\psi > 4$, the $\psi$-th moment of $Z_{jk}^{(i)}$ is upper bounded by $1/\tau$, i.e., $\mathbb{E}[|Z_{jk}^{(i)}|^\psi] \leq \tau^{-1}$, which means that the tails do not decay too slowly. The eigenvalues of $\Sigma_{(i)}$, denoted as $\lambda_1^{(i)}, \cdots, \lambda_p^{(i)}$, are all bounded between $\tau$ and $\tau^{-1}$, i.e., $\tau \leq \lambda_p^{(i)} \leq \cdots \leq \lambda_2^{(i)} \leq \lambda_1^{(i)} \leq \tau^{-1}$, which means that the covariance matrix is well-conditioned (i.e., the distribution is well-spread). Furthermore, the entries of $\varepsilon_{(i)} \in \mathbb{R}^{n_i}$ have bounded moments up to any order, i.e., for any $k \in \mathbb{N}$, there exists a constant $C_k > 0$ s.t. $\mathbb{E}[|\varepsilon_{(i)_j}|^k] \leq C_k$ (noise is not heavy tailed). The sample sizes are comparable with the dimension $p$, i.e., $\gamma := n/p$, $\gamma_t := n_t/p$, and $\gamma_s := n_s/p$, with $0 \leq \gamma_t \leq 1/\tau$ and $\tau \leq \gamma$, $\gamma_s \leq 1/\tau$. Lastly, let $\|\mu_{(i)}\|_2 = r_{(i)} \sqrt{p}$, where $r_{(i)}$[1] is a constant, with a constant angle between them $\varphi := |\langle \mu_s, \mu_t \rangle| / (\|\mu_s\|_2 \|\mu_t\|_2)$.[2]

**Risk and estimator.** We test estimators on data sampled from the same distribution as the training dataset $(X_t, y_t)$ and, given an estimator $\hat{\beta}$, its out-of-sample excess risk is defined as

$$R_X(\hat{\beta}; \beta) := \mathbb{E}[(x_t^\top \hat{\beta} - x_t^\top \beta)^2 \mid X] = \mathbb{E}\left[\|\hat{\beta} - \beta\|_{\Sigma_t + \mu_t \mu_t^\top}^2 \mid X\right],$$

where $x_t$ has the same distribution as $Z^t (\Sigma_t)^{1/2} + \mu_t$ and $\|x\|_M^2 := x^\top M x$. This definition differs from similar ones appeared in (Yang et al., 2025; Song et al., 2024; Hastie et al., 2022) as the test distribution is not zero-mean (test data cannot be centered as knowing the mean is equivalent to knowing the label). The test error is then equal to the excess risk plus the noise variance $\sigma^2$, which corresponds to the Bayes error. Since $\sigma^2$ is a constant, minimizing excess risk and test error is the same, and we minimize the former. The excess risk is decomposed into bias and variance as

$$R_X(\hat{\beta}; \beta) = \|\mathbb{E}[\hat{\beta} \mid X] - \beta\|_{\Sigma_t + \mu_t \mu_t^\top}^2 + \text{Tr}[\text{Cov}(\hat{\beta} \mid X)(\Sigma_t + \mu_t \mu_t^\top)] := B_X(\hat{\beta}; \beta) + V_X(\hat{\beta}; \beta). \quad (3.3)$$

Let $\hat{\beta}$ be the min-norm least squares regression estimator of $y$ on the whole dataset available $X$, i.e.,

$$\hat{\beta} := \arg\min \left\{ \|b\|_2 : b \text{ minimizes } \|y - Xb\|_2^2 \right\} = (X^\top X)^+ X^\top y, \quad (3.4)$$

where $(\cdot)^+$ denotes the pseudo-inverse. We note that gradient descent converges to the interpolator which is the closest in $\ell_2$ norm to the initialization (see Equation (33) in Bartlett et al. (2021)) and, as such, (3.4) corresponds to the gradient descent solution starting from 0 initialization. The results in the next section readily extend to weighted versions of objective optimized in (3.4), see Appendix A.8 for details. For this reason, we present our main findings for the unweighted objective without loss of generality. Substituting (3.4) into the excess risk decomposition (3.3) yields closed-form expressions for bias and variance:

$$B_X(\hat{\beta}; \beta) = \beta^\top \Pi(\Sigma_t + \mu_t \mu_t^\top)\Pi\beta \quad \text{and} \quad V_X(\hat{\beta}; \beta) = \frac{\sigma^2}{n} \text{Tr}[\hat{\Sigma}^+(\Sigma_t + \mu_t \mu_t^\top)], \quad (3.5)$$

where $\hat{\Sigma} = X^\top X/n$ and $\Pi = I - \hat{\Sigma}^+ \hat{\Sigma}$ (projection on the null space of $X$).

## 4 THEORETICAL RESULTS

We characterize the excess risk of the min-norm interpolator using both training and augmenting synthetic data. We then use the explicitly derived formulas to optimize the data selection process,

---

[1]Taking $\|\mu_{(i)}\|_2 \sim \sqrt{p}$ ensures that the mean and variance of $y_{(i)}$ are of same order. In fact, $\|\mu_{(i)}\|_2 \ll \sqrt{p}$ would imply $\langle \mu_t, \beta \rangle \ll 1$, so the mean would have a vanishing effect on the risk. Furthermore, if $\|\mu_{(i)}\|_2 \gg \sqrt{p}$, then $\langle \mu_{(i)}, \beta \rangle \gg 1$, and the mean would dominate the risk entirely. In both cases the problem trivializes.

[2]This is a technical assumption to simplify the proof notation. If $\varphi$ is allowed to depend on $n, p$, all results (and corresponding proofs) still hold verbatim, as long as either $\varphi < 1 - \delta$ for some constant $\delta > 0$ or $\varphi = 1$.

in which, surprisingly, distribution means play no role. We contrast this setting with having only synthetic data available, where means instead impact the excess risk. Our findings hold in both the under-parameterized and over-parameterized regimes. For clarity, we present the two regimes separately, as the precise statements and proofs rely on different technical arguments.

## 4.1 UNDER-PARAMETERIZED REGIME

Let us assume that $1 + \tau \leq \gamma \leq 1/\tau$, implying that $n > p$, which makes the setting under-parametrized. Thus, $\hat{\Sigma} = X^\top X/n$ is full rank almost surely, which implies that $\Pi = I - \hat{\Sigma}^+ \hat{\Sigma} = I - \hat{\Sigma}^{-1}\hat{\Sigma} = 0$. From (3.5), it follows that $B_X(\hat{\beta}; \beta) = 0$, so the risk is only characterized by the variance $V_X(\hat{\beta}; \beta)$. We additionally constrain the number of samples as $1 + \tau \leq \gamma_t, \gamma_s \leq 1/\tau$ and $0 < \gamma_s/\gamma_t \leq 1/\tau$.

The following result provides a precise asymptotic characterization of the excess risk and, in doing so, it extends results by Yang et al. (2025) to non-zero centered data. Its proof is deferred to Appendix A.1 and we give a brief sketch of the argument below.

**Theorem 4.1.** *Let $M = \Sigma_s^{1/2}\Sigma_t^{-1/2}$ and denote the eigenvalues of $M^\top M$ as $\lambda_1 \geq \cdots \geq \lambda_p$. Then, under the assumptions from Section 3 and the start of this section, it holds that, with high probability,*

$$\lim_{n \to \infty} \left| R_X(\hat{\beta}; \beta) - \frac{\sigma^2}{n} \operatorname{Tr}\left[ \left( \alpha_1 M^\top M + \alpha_2 I_p \right)^{-1} \right] \right| = 0, \quad (4.1)$$

*where $\alpha_1$ and $\alpha_2$ are the unique positive solutions to the following two equations*

$$\alpha_1 + \alpha_2 = 1 - \frac{p}{n}, \quad \alpha_1 + \frac{1}{n}\sum_{i=1}^{p} \frac{\lambda_i \alpha_1}{\lambda_i \alpha_1 + \alpha_2} = \frac{n_s}{n}. \quad (4.2)$$

*Proof sketch.* As seen from (3.5), $R_X(\hat{\beta}; \beta)$ is related to spectral properties of the sample covariance matrix $\hat{\Sigma}$, dictated by its local laws. The core of our argument is to connect the spectrum of $\hat{\Sigma}$ for non-centered data to its zero-centered counterpart. This is done by factoring out the means $\mu_t, \mu_s$ as a rank-2 perturbation of a random matrix with i.i.d. entries, see Propositions A.1, A.2, and A.3 in Appendix A.1. We then apply anisotropic local laws for the zero-centered case and conclude. We finally note that this strategy gives a convergence rate of $O(\sigma^2 p^{-1/2})$ for the LHS of (4.1). $\square$

Theorem 4.1 gives a *deterministic equivalent* of the test error obtained using training and synthetic data in the under-parameterized regime. In fact, $R_X(\hat{\beta}; \beta)$ is a random quantity (the data is random), while $\frac{\sigma^2}{n}\operatorname{Tr}[(\alpha_1 M^\top M + \alpha_2 I_p)^{-1}]$ is deterministic as it depends on properties of the data distributions. Remarkably, the deterministic equivalent depends only on the covariances $\Sigma_t, \Sigma_s$ (via $M = \Sigma_s^{1/2}\Sigma_t^{-1/2}$) and it does not depend on the means $\mu_t, \mu_s$. This is highlighted in Figure 1a, showing that the excess risk is unchanged upon varying the cosine similarity between the means. Two points are now in order, which are elaborated upon in the next two paragraphs.

(a) The independence of the test error on the mean shift is surprising, and it is in stark contrast with the setting in which we only train on $(X_s, y_s)$, where the performance does depend on $\mu_s, \mu_t$.

(b) The deterministic equivalent can be optimized to find the covariance $\Sigma_s$ minimizing the error.

**(a) Training only on synthetic data.** We now adjust our assumption at the beginning of this section. Namely, we assume that $\gamma_t = 0$, $1 + \tau \leq \gamma_s = \gamma \leq 1/\tau$, which means that we are training on data from a single distribution that is different from the one we are testing on.

**Proposition 4.2.** *In the setting described above, it holds that, with high probability,*

$$\lim_{n \to \infty} \left| R_X(\hat{\beta}; \beta) - \frac{\sigma^2}{n} \cdot \frac{\gamma}{\gamma - 1} \cdot \left[ \operatorname{Tr}[\Sigma_t \Sigma_s^{-1}] + \|\Sigma_s^{-1/2}\mu_t\|_2^2 - \left( \frac{\mu_t^\top \Sigma_s^{-1}\mu_s}{\|\Sigma_s^{-1/2}\mu_s\|_2} \right)^2 \right] \right| = 0. \quad (4.3)$$

This result (proved in Appendix A.2) extends the zero-centered expression by Hastie et al. (2022). We observe consistency if we disregard means ($\mu_s = \mu_t = 0$) and covariance shift ($\Sigma_t \Sigma_s^{-1} = I_p$). Proposition 4.2 also extends the zero-centered anisotropic setting of Yang et al. (2025) to the case

without samples from the training distribution, and consistency follows after setting $\mu_s = \mu_t = 0$. The effect of the mean shift is captured by $\|\Sigma_s^{-1/2}\mu_t\|_2^2 - (\mu_t^\top \Sigma_s^{-1}\mu_s/\|\Sigma_s^{-1/2}\mu_s\|_2)^2$: what matters is *(i)* the cosine similarity between $\Sigma_s^{-1/2}\mu_s$ and $\Sigma_s^{-1/2}\mu_t$, and *(ii)* the alignment of the principal directions of $\Sigma_s$ with $\mu_t$. In other words, the excess risk decreases as *(i)* the mean of synthetic training data aligns with the mean of test data in the directions of the synthetic covariance matrix, and *(ii)* the principal directions of the synthetic covariance matrix align with the test mean.

**(b) Synthetic data selection.** Let us denote the deterministic quantity from (4.1) as

$$\mathcal{R}_u(M) := \frac{\sigma^2}{n} \operatorname{Tr}\left[\left(\alpha_1 M^\top M + \alpha_2 I_p\right)^{-1}\right], \tag{4.4}$$

where $\alpha_1$ and $\alpha_2$ satisfy (4.2). This corresponds to the limit of the risk $R_X(\hat{\beta}; \beta)$ due to Theorem 4.1. Note that $\mathcal{R}_u(M)$ depends only on the covariance matrices of the original training ($\Sigma_t$) and the augmenting synthetic data ($\Sigma_s$) via $M = \Sigma_s^{1/2}\Sigma_t^{-1/2}$. Thus, in the under-parameterized setting, the guiding question (Q) posed in the introduction can be formalized as:

*Given $\Sigma_t$, what is the optimal $\Sigma_s$ that minimizes $\mathcal{R}_u(M)$?*

The following theorem exactly treats this. Its proof is in Appendix A.3 and a brief sketch is below.

**Theorem 4.3.** *Let $\mathcal{M} := \{M \in \mathbb{R}^{p \times p} : \operatorname{rank}(M) = p, \operatorname{Tr}[M^\top M] = p\}$. Then, for $M_{opt} \in \mathcal{M}$ minimizing the limit risk of Theorem 4.1, i.e., $M_{opt} := \operatorname{arginf}_{M \in \mathcal{M}} \mathcal{R}_u(M)$, it holds that*

$$\lambda_i(M_{opt}^\top M_{opt}) = 1, \qquad \forall i \in \{1, \ldots, p\}. \tag{4.5}$$

*Proof sketch.* From the first equation in (4.2), $\mathcal{R}_u(M)$ can be expressed in terms of a single parameter, e.g., $\alpha_1$. A key insight is that $\mathcal{R}_u(M)$ is increasing in $\alpha_1$, which simplifies the optimization. Denoting with $\lambda_1 \geq \cdots \geq \lambda_p$ the eigenvalues of $M$ in decreasing order, we show that transformations of the form $(\lambda_i, \lambda_j) \to (\lambda_i - c, \lambda_j + c)$ for $c > 0$, can only lower $\alpha_1$. Thus, a majorization argument allows us to conclude that the most balanced solution (namely, (4.5)) is optimal. $\qquad \square$

Theorem 4.3 proves that, having fixed $\operatorname{Tr}[M^\top M]$, the limit risk $\mathcal{R}_u(M)$ is minimized when $M$ has all eigenvalues equal. Thus, given a training covariance $\Sigma_t$, choosing synthetic data with $\Sigma_s \propto \Sigma_t$, i.e., *matching the covariances*, is optimal. This is highlighted in Figure 1b, showing that the excess risk decreases as $\Sigma_s$ aligns with $\Sigma_t$. Increasing the scale of $\Sigma_s$ also reduces the risk, i.e., for any $M \in \mathbb{R}^{p \times p}$ s.t. $\operatorname{rank}(M) = p$ and any constant $\eta > 1$, it holds that $\mathcal{R}_u(\eta M) \leq \mathcal{R}_u(M)$, see Appendix A.4 for the proof and Figure 1c for an illustration. Recalling $M = \Sigma_s^{1/2}\Sigma_t^{-1/2}$, this suggests that greater diversity in synthetic data is advantageous. However, as Theorem 4.1 relies on bounds on the spectra of $\Sigma_t, \Sigma_s$ (see Section 3), $\eta$ must be of constant order, i.e., it cannot grow with $n$ and $p$ (otherwise, the error between $R_X(\hat{\beta}; \beta)$ and $\mathcal{R}_u(\eta M)$ may not vanish as in (4.1)). This motivates the trace normalization ($\operatorname{Tr}[M^\top M] = p$) in Theorem 4.3. While other normalizations exist (e.g., on the determinant in (Yang et al., 2025)), they overly restrict the search space and make interpretation for synthetic data selection less clear.

## 4.2 OVER-PARAMETERIZED REGIME

As opposed to Section 4.1, let us assume that $\tau \leq \gamma, \gamma_s, \gamma_t \leq 1/(1 + \tau)$, so that $n < p$ and we are in the over-parameterized regime. We sample $\beta$ from a sphere of constant radius, independently from $X, \varepsilon_t, \varepsilon_s$. We also assume that $\Sigma_s$ and $\Sigma_t$ are simultaneously diagonalizable. This assumption is of technical nature and common in related work (Song et al., 2024; Mallinar et al., 2024; Ildiz et al., 2025). Writing out this condition, we have the SVDs $\Sigma_s = U\Lambda^s U^\top, \Sigma_t = U\Lambda^t U^\top$. Let us denote by $\lambda_i^s := \Lambda_{i,i}^s, \lambda_i^t := \Lambda_{i,i}^t$ and introduce the spectral probability distributions used in our claims:

$$\hat{H}_p(\lambda^s, \lambda^t) := \frac{1}{p} \sum_{i=1}^p \mathbf{1}_{\{(\lambda^s, \lambda^t) = (\lambda_i^s, \lambda_i^t)\}}, \quad \hat{G}_p(\lambda^s, \lambda^t) := \sum_{i=1}^p \langle \beta, u_i \rangle^2 \, \mathbf{1}_{\{(\lambda^s, \lambda^t) = (\lambda_i^s, \lambda_i^t)\}}. \tag{4.6}$$

This section follows the same blueprint as Section 4.1 for the under-parameterized regime. Namely, Theorem 4.4 gives a *deterministic equivalent* of the excess risk using training and synthetic data and, in doing so, it extends results by Song et al. (2024) to non-zero centered data. The deterministic equivalent depends only on regression coefficients $\beta$ and covariances $\Sigma_t, \Sigma_s$, and it does not depend

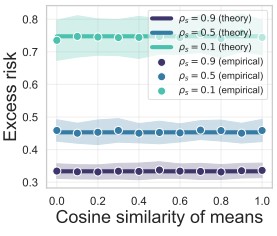 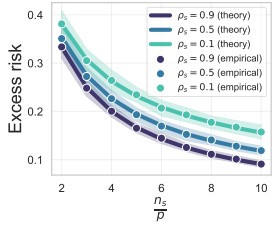 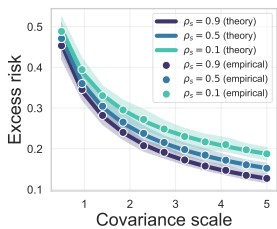

(a) Aligning the means   (b) More synthetic data   (c) Scaling the covariance

Figure 1: Excess risk using training data from $\mathcal{N}(\mu_t, \Sigma_t)$ and synthetic data from $\mathcal{N}(\mu_s, \Sigma_s)$, where $\Sigma_t, \Sigma_s$ are Kac–Murdock–Szegö matrices (Toeplitz matrices with geometrically decaying entries) with parameters $\rho_t, \rho_s$, scaled so that $\mathrm{Tr}[M^\top M] = p$. We pick $\|\mu_t\|_2 = \|\mu_s\|_2 = 2\sqrt{p}$, $\rho_t = 0.9$, $p = 600$, $n_t = 1200$, $n_s = 1200$, unless varying the parameters in the plot. Each value is computed from 100 i.i.d. trials, the error band is at 1 standard deviation, and theoretical predictions are continuous lines. Different curves correspond to different values of $\rho_s$. *(a)* Changing the cosine similarity of the mean does not impact the risk (here, $\Sigma_s$ is scaled by $\eta := \rho_s$). *(b)* Larger $\rho_s$ gives lower risk since $\Sigma_s$ is closer to $\Sigma_t$. *(c)* Scaling $\Sigma_s$ reduces the risk.

on means $\mu_t, \mu_s$. Then, Theorem 4.5 finds $\Sigma_s$ that minimizes the limit risk from Theorem 4.4 when $\Sigma_t = I_p$, thus showing the optimality of *covariance matching* ($\Sigma_s \propto \Sigma_t$) with isotropic training data. The proofs of these results follow a similar argument chain as in Section 4.1, although they tend to be more technically involved. We briefly discuss differences, deferring the full arguments of Theorems 4.4 and 4.5 to Appendices A.6 and A.7, respectively.

**Theorem 4.4.** *Under the assumptions from Section 3 and the start of this section, it holds that, with high probability,*

$$\lim_{n \to \infty} \left| R_X(\hat{\beta}; \beta) - \mathcal{V}(\Sigma_s, \Sigma_t) - \mathcal{B}(\Sigma_s, \Sigma_t, \beta) \right| = 0, \tag{4.7}$$

*where*

$$\mathcal{V}(\Sigma_s, \Sigma_t) := \frac{\sigma^2}{\gamma} \int \frac{-\lambda^t(a_3\lambda^s + a_4\lambda^t)}{(a_1\lambda^s + a_2\lambda^t + 1)^2} d\hat{H}_p(\lambda^s, \lambda^t), \quad \mathcal{B}(\Sigma_s, \Sigma_t, \beta) := \int \frac{b_3\lambda^s + (b_4+1)\lambda^t}{(b_1\lambda^s + b_2\lambda^t + 1)^2} d\hat{G}_p(\lambda^s, \lambda^t),$$

*and $a_i$, $b_i$ ($i \in \{1, 2, 3, 4\}$) are the unique solutions to the equations reported in Appendix A.5.*

We highlight two additional difficulties in the proof of Theorem 4.4 arising from the over-parameterized regime: *(1)* the inverse does not replace the pseudo-inverse in (3.5), and *(2)* the bias term does not vanish. We address the former by introducing the $\lambda$-regularized ridge estimator $\hat{\beta}_\lambda$, which approximates $\hat{\beta}$ for small $\lambda$ and admits inverse-based formulas similar to (3.5). Addressing the latter requires a delicate control of the inverse, obtained via Woodbury formula.

**Theorem 4.5.** *Let $\mathcal{S} := \{\Sigma \in \mathbb{R}^{p \times p}_{\succ 0} : \mathrm{Tr}(\Sigma) = p\}$, where $\mathbb{R}^{p \times p}_{\succ 0}$ denotes the set of $p \times p$ positive definite matrices. Recall the definitions of $\mathcal{V}(\Sigma_s, \Sigma_t)$, $\mathcal{B}(\Sigma_s, \Sigma_t, \beta)$ from Theorem 4.4, and define $\mathcal{R}_o(\Sigma_s, \Sigma_t, \beta) := \mathcal{V}(\Sigma_s, \Sigma_t) + \mathcal{B}(\Sigma_s, \Sigma_t, \beta)$. Then, for any $\Sigma_s \in \mathcal{S}$, with high probability over the sampling of $\beta$ over a sphere of constant radius, it holds that*

$$\mathcal{R}_o(I_p, I_p, \beta) \le \mathcal{R}_o(\Sigma_s, I_p, \beta) + o(1),$$

*where $o(1)$ denotes a quantity that vanishes as $n, p \to \infty$.*

Due to the complexity of the expressions for $\mathcal{V}(\Sigma_s, \Sigma_t)$ and $\mathcal{B}(\Sigma_s, \Sigma_t, \beta)$, the optimality of covariance matching ($\Sigma_s \propto \Sigma_t$) in the over-parameterized regime is shown for isotropic training data ($\Sigma_t = I_p$). At the technical level, we note that the bias generally depends on the eigenspace decomposition of the covariance matrices via $\hat{G}_p$, as defined in (4.6). However, when $\Sigma_t = I_p$, cancellations in the equations for $b_i$ ($i \in \{1, 2, 3, 4\}$) give that the bias $\mathcal{B}(\Sigma_s, I_p, \beta)$ is close to $\frac{p-n}{p} \|\beta\|_2$ for any $\Sigma_s$. Having obtained that, the variance is then optimized following the approach of Theorem 4.3.

## 5 EXPERIMENTAL RESULTS

Theorems 4.3 and 4.5 show the optimality of *covariance matching* ($\Sigma_s \propto \Sigma_t$) in both under-parameterized and over-parameterized regimes. We now extensively test the applicability of this

Table 1: *Covariance matching* outperforms all baselines across three training paradigms on CIFAR-10, when the synthetic data is generated via five truncated StyleGAN2-Ada models.

| Method | Scratch | Distillation | Pretrained |
|---|---|---|---|
| No synthetic | $44.36 \pm 1.51$ | $47.33 \pm 0.57$ | $63.40 \pm 1.33$ |
| Center matching (He et al., 2023) | $50.04 \pm 2.84$ | $53.83 \pm 0.59$ | $67.01 \pm 0.89$ |
| Center sampling (Lin et al., 2023) | $50.48 \pm 2.03$ | $54.91 \pm 1.07$ | $67.71 \pm 0.90$ |
| DS3 (Hulkund et al., 2025) | $52.83 \pm 2.19$ | $58.32 \pm 0.43$ | $68.21 \pm 0.66$ |
| K-means (Lin et al., 2023) | $50.74 \pm 1.77$ | $56.06 \pm 0.68$ | $66.50 \pm 1.11$ |
| Random | $49.38 \pm 2.43$ | $54.89 \pm 0.91$ | $67.65 \pm 0.77$ |
| Text matching (Lin et al., 2023) | $50.94 \pm 1.40$ | $55.17 \pm 0.57$ | $67.81 \pm 0.76$ |
| Text sampling (Lin et al., 2023) | $50.28 \pm 1.18$ | $54.82 \pm 0.72$ | $67.45 \pm 1.02$ |
| Covariance matching (ours) | $54.00 \pm 1.89$ | $59.77 \pm 0.61$ | $69.20 \pm 0.56$ |
| Real upper bound | $61.08 \pm 2.54$ | $65.38 \pm 0.51$ | $74.35 \pm 0.56$ |

synthetic data selection criterion in a range of practical settings. We consider classification problems, assume access to a large pool of synthetic samples obtained from generative models, and perform the augmentation per class. We implement *covariance matching* via a greedy algorithm: we initialize $S = \varnothing$ and, until $|S| = n_s$, we add the $x$ from the generated pool that minimizes $\|\widehat{\Sigma}(S \cup \{x\}) - \widehat{\Sigma}_t\|_F$, where $\widehat{\Sigma}(\cdot)$ and $\widehat{\Sigma}_t$ denote the sample covariance of CLIP features of the synthetic samples and real samples respectively and $\|\cdot\|_F$ is the Frobenius norm. To accelerate the selection, we compute covariances in a 32-dimensional PCA space fit on the $n_t$ real reference features. After the selection, we train a classifier on the union of real and selected synthetic samples.

**Experimental setup.** When using CIFAR-10, we evaluate three training paradigms. *(1) Scratch*: train a ResNet-18 (He et al., 2016) from scratch on the available data. *(2) Distillation*: train a ResNet-18 using soft targets (logits) from a ResNet-50 trained on full CIFAR-10, following Hinton et al. (2015). *(3) Pretrained*: fine-tune an ImageNet-pretrained ResNet-18 with a new classification head. We also repeat the *Scratch* and *Distillation* experiments replacing the ResNet with two transformer models (ViT and Swin-T). Unless stated otherwise, we use $n_t = 200$ real images and augment with $n_s = 800$ synthetic images per class. The features for the selection algorithms are extracted with CLIP ViT-B, yielding a $p = 512$-dimensional feature space, which places us in an under-parameterized regime. We report in Table 9 in Appendix C an additional experiment for $n_s + n_t = 400$, which places us in an over-parameterized regime. We additionally consider ImageNet-100 as a more diverse dataset, and RxRx1 (Sypetkowski et al., 2023) as a specialized one. For RxRx1, we use a small subset of $n_t = 30$ images from four common perturbations (1108, 1124, 1137, 1138) on HUVEC cells. We consider the task of perturbation classification and augment with $n_s = 60$ samples chosen from 500 images generated by MorphGen (Demirel et al., 2025). Further details are in Appendix C.

**Baselines.** We compare *Covariance matching* with the following baselines. *(1) Center matching* (He et al., 2023): select the $n_s$ images nearest to the centroid of the $n_t$ real training features. *(2) Center sampling* (Lin et al., 2023): sample with probability proportional to the cosine similarity to the $n_t$ real training features. *(3) DS3* (Hulkund et al., 2025): cluster the generated pool into 200 clusters; for each of the $n_t$ real images, retain its nearest cluster; then, sample $n_s$ images uniformly from the retained set. *(4) K-means* (Lin et al., 2023): cluster the generated pool into $n_s$ clusters and choose one random representative per cluster. *(5) Random*: uniformly sample $n_s$ images from the generated pool. The methods "No-filtering" (Hulkund et al., 2025), "Match-dist" (Hulkund et al., 2025), and "Match-label" (Hulkund et al., 2025) are all equivalent to *Random* in our setting due to having the same number of data for each class. *(6) Text matching* (Lin et al., 2023): select the $n_s$ images nearest to the class text embedding. *(7) Text sampling* (Lin et al., 2023): sample with probability proportional to the cosine similarity to the class text embedding. We also report a baseline, *No synthetic*, corresponding to using only $n_t$ samples from the training distribution (synthetic data discarded), as well as a baseline, *Real upper bound*, corresponding to using $n_t + n_s$ samples from the training distribution (synthetic data replaced by in-distribution data). All experiments are repeated over 10 random seeds (except Table 3a which is on 5 seeds), and we report the mean $\pm$ 1 standard deviation.

**Main findings.** First, we test *diversity/quality trade-offs*. To do so, for each class we generate images with StyleGAN2-Ada (Karras et al., 2020) under different truncations (Karras et al., 2019): 6K images from a 0.2-truncated model with three randomized truncation centers and 4K images from a 0.6-truncated model with two randomized centers. This produces synthetic data with varying diversity

Table 2: *Covariance matching* performs on par with the best baseline across three training paradigms on CIFAR-10, when the synthetic data is generated via various T2I generative models.

| Method | Scratch | Distillation | Pretrained |
|---|---|---|---|
| No synthetic | $44.36 \pm 1.51$ | $47.33 \pm 0.57$ | $63.40 \pm 1.33$ |
| Center matching (He et al., 2023) | $53.46 \pm 1.95$ | $57.67 \pm 0.58$ | $66.52 \pm 0.81$ |
| Center sampling (Lin et al., 2023) | $50.15 \pm 1.79$ | $56.05 \pm 0.65$ | $65.38 \pm 0.98$ |
| DS3 (Hulkund et al., 2025) | $54.15 \pm 2.17$ | $59.43 \pm 0.73$ | $66.00 \pm 0.94$ |
| K-means (Lin et al., 2023) | $51.63 \pm 1.29$ | $56.77 \pm 0.89$ | $65.23 \pm 0.61$ |
| Random | $51.26 \pm 1.96$ | $55.27 \pm 0.74$ | $65.24 \pm 1.01$ |
| Text matching (Lin et al., 2023) | $51.20 \pm 1.82$ | $56.08 \pm 0.57$ | $65.93 \pm 0.59$ |
| Text sampling (Lin et al., 2023) | $50.31 \pm 1.70$ | $55.79 \pm 0.68$ | $64.93 \pm 1.12$ |
| Covariance matching (ours) | $54.45 \pm 2.11$ | $59.17 \pm 0.64$ | $66.69 \pm 0.70$ |
| Real upper bound | $61.08 \pm 2.54$ | $65.38 \pm 0.51$ | $74.35 \pm 0.56$ |

Table 3: *Covariance matching* performs on par with the best baselines for two additional datasets. In (a), we train a ResNet-18 from scratch on ImageNet-100 with synthetic images from StyleGAN-XL and T2I models. In (b), we train a linear model on top of an ImageNet-pretrained ResNet for perturbation classification on a small subset of RxRx1 (Sypetkowski et al., 2023) augmented with synthetic images from MorphGen (Demirel et al., 2025).

| Method | Truncated models | T2I models |
|---|---|---|
| No synthetic | $40.78 \pm 1.29$ | |
| Center matching (He et al., 2023) | $53.39 \pm 0.37$ | $53.96 \pm 1.06$ |
| DS3 (Hulkund et al., 2025) | $57.47 \pm 0.87$ | $53.51 \pm 0.31$ |
| Random | $54.14 \pm 0.82$ | $49.84 \pm 1.32$ |
| Text matching (Lin et al., 2023) | $53.39 \pm 0.99$ | $53.37 \pm 0.72$ |
| Covariance matching (ours) | $57.52 \pm 0.36$ | $53.07 \pm 0.89$ |
| Real upper bound | $62.67 \pm 0.65$ | |

(a) ImageNet-100 dataset

| Method | MorphGen |
|---|---|
| No synthetic | $86.83 \pm 2.44$ |
| Center matching (He et al., 2023) | $88.17 \pm 2.35$ |
| Random | $87.33 \pm 2.03$ |
| K-means (Lin et al., 2023) | $89.00 \pm 1.70$ |
| DS3 (Hulkund et al., 2025) | $89.67 \pm 1.45$ |
| Center sampling (Lin et al., 2023) | $88.75 \pm 2.27$ |
| Covariance matching (ours) | $90.00 \pm 1.86$ |

(b) RxRx1 dataset

and fidelity, and we note that the diversity we refer to concerns the synthetic dataset obtained from the generative model. The results of Table 1 demonstrate that covariance matching outperforms all baselines for all training paradigms. Table 10 in Appendix C suggests that this superiority is partly due to selecting more diverse samples, evident from the improved Recall (Kynkäänniemi et al., 2019), FID (Heusel et al., 2017), and KID (Bińkowski et al., 2018) scores guaranteed by covariance matching. We further notice that covariance matching selects 268, 245, 333 samples from the three StyleGANs with truncation 0.2, and 3692, 3462 samples from the two StyleGANs with truncation 0.6, pointing out the preference of covariance matching towards more diverse samples. Going beyond ResNets, we also demonstrate the effectiveness of covariance matching for transformer models in Table 4 in Appendix C. Second, we test *text-to-image (T2I) generative models*. To do so, for each class we generate 4K SANA-1.5 (Xie et al., 2025), 4K PixArt-$\alpha$ (Chen et al., 2024), and 2K StableDiffusion1.4 (Rombach et al., 2022) images. Table 2 shows that covariance matching also performs well in this mixed setup.

Finally, to demonstrate the generality of our findings, we consider a broader dataset from computer vision (ImageNet-100) and a specialized dataset from fluorescence microscopy (RxRx1, (Sypetkowski et al., 2023)). Once again, the results reported in Tables 3a-3b show that covariance matching performs on par with the best baselines in all settings. Additionally, the effectiveness is also shown for text classification on the Ironic-Tweet dataset (Van Hee et al., 2018) in Table 13 in Appendix C.

**Additional controlled experiments.** We report additional results in Appendix C. In Table 5, we consider *zero-diversity generators*. Specifically, for each class, we combine 2K StyleGAN2-Ada images with a total of 8K images produced by two zero-diversity generators. Each of these generators emits a single prototype per class: one near the class center of the real samples, and one near the class label's CLIP embedding. This yields high precision, but low diversity relative to the real distribution. Our results show that, again, covariance matching performs well as it avoids selecting many samples with low diversity (collapsed clusters). In contrast, not fully taking into account the diversity of selected samples, methods like DS3 perform rather poorly. In Figure 2, we consider *inserting images from the target distribution* into the pool of synthetic images and test the ability of different methods to select them. Specifically, we form a pool of 4K StableDiffusion1.4 images and 1K images from the target distribution (different from the $n_t = 200$ images forming the training distribution), letting

each method take $n_s = 800$. Our results show that covariance matching selects the highest fraction of images coming from the target distribution, whereas other selectors largely fail to do so.

**Additional ablations.** In Tables 6-7, we repeat the experiments of Tables 1-2 with DINO instead of CLIP features, demonstrating that the gains of covariance matching are not tied to a particular feature extractor. In Table 8, we compare covariance matching with the direct optimization of the objective given by Theorem 4.1. As the outcomes of these two procedures are largely similar, this further justifies the covariance matching objective. In Table 9, we show that our findings replicate in an over-parameterized regime. In Table 10, we examine the distribution of selections produced by each method, quantifying alignment with the test distribution and identifying which metrics best predict downstream accuracy.

In Table 11, we analyze the effect of varying the number of real training samples and synthetic augmentations. Table 12 reports the performance of different variants of *Covariance Matching*.

All these tables and figures are reported and discussed in Appendix C.

## 6 CONCLUSION

This paper advances understanding of the precise connection between training on a mix of real and synthetic data and generalizing on real data. We start with a high-dimensional linear regression analysis, where we find that only covariance shifts, and not mean shifts, affect the error. Even if our theory ignores the interactions between classes that would affect neural network training, the resulting insights transfer to realistic settings. We empirically demonstrate that matching the covariance between samples from real image classification datasets and generative models (irrespective of whether they are from GANs or diffusion model variants) improves the accuracy of deep networks (ResNets and Transformers) under different training regimes (from scratch, distillation, and fine-tuning). In fact, our principled approach even performs on-par or better than existing baselines (Hulkund et al., 2025; He et al., 2023; Lin et al., 2023). Future work could extend the analysis to multiple Gaussian mixtures, which corresponds to optimizing the actual risk as opposed to modeling individual classes. We speculate that this may yield different insights when the training data have extremely imbalanced or fine-grained classes. It would also be interesting to introduce a model shift (different $\beta$ between synthetic and real samples) under covariance shift . In fact, synthetic data often has small differences compared to real data, which a model may overfit on, and the phenomenon could be the cause of the collapse sometimes observed in practice (Shumailov et al., 2024). Finally, we have only focused on generalization, but other quantities may be studied in this framework, including uncertainty calibration (Nixon et al., 2019), differential privacy (Dwork, 2006), fairness (Barocas et al., 2020), and validity for prediction-powered causal inference (Cadei et al., 2025).

## ETHICS STATEMENT

This paper presents work whose goal is to advance the field of machine learning. There are many potential societal consequences of our work, none of which we feel must be specifically highlighted here. Our experiments in computational biology should not be interpreted as a direct clinical application.

## REPRODUCIBILITY STATEMENT

We have made significant efforts to ensure the reproducibility of our results. Full implementation details are provided in Appendix C. Additionally, we provide the codebase as a supplementary material of our submission.

## ACKNOWLEDGEMENTS

P.R., F.K. and M.M. are funded by the European Union (ERC, INF$^2$, project number 101161364). Views and opinions expressed are however those of the author(s) only and do not necessarily reflect those of the European Union or the European Research Council Executive Agency. Neither the European Union nor the granting authority can be held responsible for them. F.L.'s contribution to this research was funded in part by the Austrian Science Fund (FWF) 10.55776/COE12. The authors

thank Pragya Sur for insightful discussions and Berker Demirel for his help with the experiments on cell microscopy images.

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

# A    PROOFS OF THE THEORETICAL RESULTS

**Additional notation.**    We use the shorthand $[n] := \{1, \ldots, n\}$ for an integer $n$. Given a matrix $M$, its operator norm is denoted by $\|M\|_2$, its $i$-th largest singular value by $\sigma_i(M)$ and the corresponding $i$-th left-singular (resp. right-singular) vector of unit norm by $u_i(M)$ (resp. $v_i(M)$). Additionally, when applicable, we denote the $i$-th largest eigenvalue of $M$ by $\lambda_i(M)$. We use $\mathbb{R}^{p \times p}_{\succ 0}$ to denote the set of all $p \times p$ positive definite matrices, and $S^{p-1}$ to denote a $(p-1)$-dimensional unit sphere. We denote by $e_i$ the $i$-th element of the canonical basis of $\mathbb{R}^l$, where the exact exponent $l$ is assumed from context. We will say that an event $\mathcal{E}$ happens *with high probability* (w.h.p.) if and only if $\mathbb{P}(\mathcal{E}) \to 1$ as $p, n \to \infty$. Moreover, we will say that an event $\Xi$ happens *with overwhelming probability* if and only if, for any large constant $D > 0$, $\mathbb{P}(\Xi) \geq 1 - p^{-D}$ for large enough $p$. Lastly, throughout this appendix, we use $c$ to denote a constant (independent of $n, p$) whose value may change from line to line.

For convenience, we recall some notation and definitions from Section 3. Namely, we denote by $Z \in \mathbb{R}^{n \times p}$ a random matrix with i.i.d. entries having zero mean, unit variance and bounded $\psi$-th moment (for some $\psi > 4$). Recall $\mu_{(i)} \in \mathbb{R}^p$, for $(i) \in \{s, t\}$, such that $\|\mu_{(i)}\|_2 = r_{(i)} \sqrt{p}$, where $r_{(i)}$ is a constant, with a constant angle between them $\varphi := |\langle \mu_s, \mu_t \rangle| / (\|\mu_s\|_2 \|\mu_t\|_2)$. Also, let $\Sigma_s, \Sigma_t \in \mathbb{R}^{p \times p}$ be covariance matrices with bounds on their spectrum as in Section 3. Then, we consider a data distribution $X = \begin{bmatrix} Z_t \Sigma_t^{1/2} + 1_{n_t} \mu_t^\top \\ Z_s \Sigma_s^{1/2} + 1_{n_s} \mu_s^\top \end{bmatrix} \in \mathbb{R}^{n \times p}$ and introduce its zero mean counterpart $X^0 := \begin{bmatrix} Z_t \Sigma_t^{1/2} \\ Z_s \Sigma_s^{1/2} \end{bmatrix}$. The corresponding sample covariance matrices are defined as $\hat{\Sigma} = \frac{X^\top X}{n}$ and $\hat{\Sigma}_0 = \frac{X^{0\top} X^0}{n}$. Lastly, unless stated otherwise, we work in the regime $n/p = \gamma$, where $\gamma \neq 1$ is a fixed constant independent of $n$ and $p$.

## A.1    PROOF OF THEOREM 4.1

We first state and prove useful results, in which we analyze the behavior of singular values of a low-rank perturbation of matrices.

**Proposition A.1.** *Let $\sigma_1 \geq \cdots \geq \sigma_{\min(n,p)}$ be the singular values of $\tilde{Z} = \frac{Z + 1_n \mu_s^\top}{\sqrt{n}}$. Then, there exists a constant $c(\gamma) > 0$ independent of $n$, such that, almost surely,*

$$\liminf_{n \to \infty} \sigma_{\min(n,p)} \geq c(\gamma).$$

*Proof.* To simplify notation we will refer to $\sigma_{\min}$ as the smallest singular value of a matrix. Let us choose an orthogonal matrix $Q \in \mathbb{R}^{n \times n}$ such that $Q 1_n = \sqrt{n}\, e_1$. Since singular values are left orthogonally invariant, we may replace $\tilde{Z}$ by

$$\tilde{Z}' = \frac{QZ}{\sqrt{n}} + e_1 \mu_s^\top.$$

Writing the rows of $QZ$ as

$$QZ = \begin{bmatrix} z_1^\top \\ Z_2 \end{bmatrix}, \qquad z_1 \in \mathbb{R}^p, \; Z_2 \in \mathbb{R}^{(n-1) \times p},$$

we have

$$\tilde{Z}' = \begin{bmatrix} \frac{z_1^\top}{\sqrt{n}} + \mu_s^\top \\ \frac{Z_2}{\sqrt{n}} \end{bmatrix}.$$

For any unit vector $x \in \mathbb{R}^p$,

$$\|\tilde{Z}'x\|_2 = \sqrt{\left( \frac{z_1^\top x}{\sqrt{n}} + \mu_s^\top x \right)^2 + \left\| \frac{Z_2 x}{\sqrt{n}} \right\|_2^2} \; \geq \; \left\| \frac{Z_2 x}{\sqrt{n}} \right\|_2.$$

Hence, by the variational definition of singular values, we have

$$\sigma_{\min}(\tilde{Z}) \; \geq \; \sigma_{\min}\left( \frac{Z_2}{\sqrt{n}} \right) = \sqrt{\frac{n-1}{n}}\, \sigma_{\min}\left( \frac{Z_2}{\sqrt{n-1}} \right). \tag{A.1}$$

By the Bai–Yin theorem (Bai & Silverstein, 2010, Theorem 5.11), for an $(n-1) \times p$ random matrix $Z_2$ with i.i.d entries with mean zero, unit variance and bounded fourth moments it holds

$$\sigma_{\min}\left(\frac{Z_2}{\sqrt{n-1}}\right) \xrightarrow[n \to \infty]{a.s.} \left|1 - \sqrt{p/(n-1)}\right|.$$

Therefore, applying $\liminf_{n \to \infty}$ to (A.1), we have

$$\liminf_{n \to \infty} \sigma_{\min}(\tilde{Z}) \geq \liminf_{n \to \infty} \sqrt{\frac{n-1}{n}} \sigma_{\min}\left(\frac{Z_2}{\sqrt{n}}\right)$$

$$= \lim_{n \to \infty} \sqrt{\frac{n-1}{n}} \sigma_{\min}\left(\frac{Z_2}{\sqrt{n}}\right)$$

$$= \left|1 - \gamma^{-1/2}\right| > 0,$$

which gives the desired result as $\gamma \neq 1$. $\qquad\square$

**Proposition A.2.** *Let* $\tilde{X}_n = X/\sqrt{n} = \frac{1}{\sqrt{n}} \begin{bmatrix} Z_t \Sigma_t^{1/2} + 1_{n_t} \mu_t^\top \\ Z_s \Sigma_s^{1/2} + 1_{n_s} \mu_s^\top \end{bmatrix} \in \mathbb{R}^{n \times p}$. *Let* $\sigma_1 \geq \cdots \geq \sigma_n$ *be the singular values of* $\tilde{X}_n$ *and* $v_1, \ldots, v_n$ *be the corresponding right singular vectors. Then, as* $n \to \infty$, *the following results hold:*

1. *For* $\varphi < 1$, *we have*

   1a. $\sigma_1 = \Theta(\sqrt{p})$, $\sigma_2 = \Theta(\sqrt{p})$, *and* $\sigma_3 = O(1)$;

   1b. $\left|\frac{\langle v_1, \mu_s \rangle}{\|v_1\|_2 \|\mu_s\|_2}\right|^2 + \left|\frac{\langle v_2, \mu_s \rangle}{\|v_2\|_2 \|\mu_s\|_2}\right|^2 = 1 - O\left(\frac{1}{p}\right),$

   $\left|\frac{\langle v_1, \mu_t \rangle}{\|v_1\|_2 \|\mu_t\|_2}\right|^2 + \left|\frac{\langle v_2, \mu_t \rangle}{\|v_2\|_2 \|\mu_t\|_2}\right|^2 = 1 - O\left(\frac{1}{p}\right).$

2. *For* $\varphi = 1$, *we have*

   2a. $\sigma_1 = \Theta(\sqrt{p})$, $\sigma_2 = O(1)$;

   2b. $\left|\frac{\langle v_1, \mu_s \rangle}{\|v_1\|_2 \|\mu_s\|_2}\right|^2 = 1 - O\left(\frac{1}{p}\right).$

*Proof.* Let us first abuse notation and write $1_{n_s} = [1, \ldots, 1, 0, \ldots, 0]^\top \in \mathbb{R}^{n \times 1}$ ($n_s$ ones followed by $n_t$ zeros) and $1_{n_t} = [0, \ldots, 0, 1, \ldots, 1]^\top \in \mathbb{R}^{n \times 1}$ ($n_s$ zeros followed by $n_t$ ones). Then if we write

$$X_n := \frac{1}{\sqrt{n}} \begin{bmatrix} Z_t \Sigma_t^{1/2} \\ Z_s \Sigma_s^{1/2} \end{bmatrix},$$

it holds

$$\tilde{X}_n = X_n + P_n, \qquad \text{where} \qquad P_n := \frac{1_{n_s} \mu_s^\top + 1_{n_t} \mu_t^\top}{\sqrt{n}}. \tag{A.2}$$

To obtain the wanted result, we will need to express the non-zero singular values and the corresponding right singular vectors of the rank-2 perturbation $P_n$, that is $\sigma_i(P_n)$ and $v_i(P_n)$, $i \in [2]$. Notice that

$$P_n^\top P_n = \alpha_s^2 \mu_s \mu_s^\top + \alpha_t^2 \mu_t \mu_t^\top,$$

where $\alpha_s := \sqrt{\frac{n_s}{n}}$ and $\alpha_t := \sqrt{\frac{n_t}{n}}$. Moreover, it holds

$$P_n^\top P_n = Q_p^\top Q_p, \tag{A.3}$$

where $Q_p = \begin{bmatrix} \alpha_s \mu_s \\ \alpha_t \mu_t \end{bmatrix} \in \mathbb{R}^{2 \times p}$. Note that

$$Q_p Q_p^\top = \begin{bmatrix} \alpha_s^2 \|\mu_s\|_2^2 & \alpha_s \alpha_t \langle \mu_s, \mu_t \rangle \\ \alpha_s \alpha_t \langle \mu_s, \mu_t \rangle & \alpha_t^2 \|\mu_t\|_2^2 \end{bmatrix} =: \begin{bmatrix} a & b \\ b & d \end{bmatrix},$$

and it is enough to analyze its SVD, since

$$\sigma_i(P_n) = \sqrt{\sigma_i(Q_p Q_p^\top)}, \qquad \text{and} \qquad v_i(P_n) = \frac{1}{\sigma_i(P_n)} v_i(Q_p Q_p^\top)^\top Q_p.$$

The previous equations hold due to (A.3), since $\sigma_i(Q_p) = \sigma_i(P_n)$, and

$$\frac{1}{\sigma_i(P_n)} v_i(Q_p Q_p^\top)^\top Q_p = \frac{1}{\sigma_i(Q_p)} u_i(Q_p)^\top [u_1(Q_p) \quad u_2(Q_p)] \begin{bmatrix} \sigma_1(Q_p) & 0 \\ 0 & \sigma_2(Q_p) \end{bmatrix} \begin{bmatrix} v_1(Q_p) \\ v_2(Q_p) \end{bmatrix}.$$

This implies that, for $i \in [2]$, the singular vectors $v_i(P_n)$ are in the $\mathrm{span}\{\mu_s, \ \mu_t\}$. Recall that the angle between $\mu_s$ and $\mu_t$ is fixed to $\varphi := \frac{|\langle \mu_s, \mu_t \rangle|}{\|\mu_s\|_2 \|\mu_t\|_2}$.

We first consider the case when $\varphi < 1$. It holds that the eigenvalues of $Q_p Q_p^\top$ are

$$
\begin{aligned}
\sigma_{1,2}(Q_p Q_p^\top) &= \frac{a + d \pm \sqrt{(a-d)^2 + 4b^2}}{2} \\
&= \frac{(r_s^2 \alpha_s^2 + r_t^2 \alpha_t^2)p \pm \sqrt{(r_s^2 \alpha_s^2 - r_t^2 \alpha_t^2)^2 p^2 + 4\alpha_s^2 r_s^2 \alpha_t^2 r_t^2 \varphi^2 p^2}}{2} \qquad \text{(A.4)} \\
&\geq p \cdot \frac{(r_s^2 \alpha_s^2 + r_t^2 \alpha_t^2) - \sqrt{(r_s^2 \alpha_s^2 - r_t^2 \alpha_t^2)^2 + 4\alpha_s^2 r_s^2 \alpha_t^2 r_t^2 \varphi^2}}{2} \\
&= p \cdot c_1,
\end{aligned}
$$

with $c_1 = \frac{(r_s^2 \alpha_s^2 + r_t^2 \alpha_t^2) - \sqrt{(r_s^2 \alpha_s^2 - r_t \alpha_t^2)^2 + 4\alpha_s^2 r_s^2 \alpha_t^2 r_t^2 \varphi^2}}{2} > 0$, since $\varphi < 1$. This implies that

$$\sigma_i(P_n) \geq c \cdot \sqrt{p},$$

for some constant $c$.

Furthermore, it almost surely holds that

$$
\begin{aligned}
\sigma_1(X_n) &= \sqrt{\sigma_1(X_n^\top X_n)} \\
&= \sqrt{\sigma_1(\Sigma_s^{1/2} Z_s^\top Z_s \Sigma_s^{1/2} + \Sigma_t^{1/2} Z_t^\top Z_t \Sigma_t^{1/2})} \\
&\leq \sqrt{\sigma_1(\Sigma_s^{1/2} Z_s^\top Z_s \Sigma_s^{1/2}) + \sigma_1(\Sigma_t^{1/2} Z_t^\top Z_t \Sigma_t^{1/2})} \\
&\leq \sqrt{2(1 + \sqrt{\gamma})^2 \cdot \tau^{-1}} = O(1),
\end{aligned}
$$

due to the convergences of the largest eigenvalue of the sample covariance matrices $Z_s^\top Z_s$ and $Z_t^\top Z_t$ by Bai–Yin theorem (Bai & Silverstein, 2010, Theorem 5.11) and the boundedness of the spectrum of $\Sigma_s$ and $\Sigma_t$. Then, from Weyl's inequality for singular values (see e.g. (Horn & Johnson, 2012, Chapter 7)), we have that

$$
\begin{aligned}
\sigma_i(X_n + P_n) &\geq \sigma_i(P_n) - \sigma_1(X_n), \quad \text{for } i = 1, 2, \\
\sigma_3(X_n + P_n) &\leq \sigma_3(P_n) + \sigma_1(X_n) = \sigma_1(X_n),
\end{aligned}
$$

which implies that $\sigma_{1,2}(X_n + P_n) \geq c \cdot \sqrt{p}$, whereas $\sigma_i(X_n + P_n) = O(1)$, for $i \geq 3$. For the upper bound, note that from (A.4) it holds

$$\sigma_{1,2}(QQ^\top) \leq \frac{(r_s^2 \alpha_s^2 + r_t^2 \alpha_t^2)p + (r_s^2 \alpha_s^2 + r_t^2 \alpha_t^2)p}{2} = p \cdot c_2,$$

implying $\sigma_i(P_n) \leq c \cdot \sqrt{p}$. Applying Weyl's inequality for singular values once more, we get

$$\sigma_i(X_n + P_n) \leq \sigma_1(X_n) + \sigma_i(P_n) = O(\sqrt{p}),$$

concluding the proof of **1a**.

Moving onto singular vectors, let us recall the definition of spectral distance between two $k$-dimensional subspaces $\mathcal{W} \leq \mathbb{R}^p$ and $\tilde{\mathcal{W}} \leq \mathbb{R}^p$, as it will be used to conclude the proof. Towards this end, we first introduce principal angles $\theta_1 \ldots \theta_k \in [0, \pi/2]$ between $\mathcal{W}$ and $\tilde{\mathcal{W}}$, which are defined recursively from $i = 1$ as

$$\cos(\theta_i) = \max_{w_i \in \mathcal{W}, \tilde{w}_i \in \tilde{\mathcal{W}}} \frac{\langle w_i, \tilde{w}_i \rangle}{\|w_i\|_2 \|\tilde{w}_i\|_2},$$

subject to $w_i, \tilde{w}_i$ being orthogonal to the previous maximizers. Then, the spectral distance between $\mathcal{W}$ and $\tilde{\mathcal{W}}$ is defined as

$$d(\mathcal{W}, \tilde{\mathcal{W}}) := \max_{i \in [k]} \sin \theta_i.$$

There is an alternative way to express this spectral distance between subspaces, using their orthonormal basis. Namely, let $W \in \mathbb{R}^{p \times k}$ and $\tilde{W} \in \mathbb{R}^{p \times k}$ be such that their columns form an orthonormal basis of $\mathcal{W}$ and $\tilde{\mathcal{W}}$, respectively. Then by (Stewart & Sun, 1990, Chapter II, Corollary 5.4) it holds

$$d(\mathcal{W}, \tilde{\mathcal{W}}) := \left\| (I - WW^\top) \tilde{W} \right\|_2. \tag{A.5}$$

Let us denote by $\tilde{V} := \begin{bmatrix} v_1(P_n) \\ v_2(P_n) \end{bmatrix}^\top, V := \begin{bmatrix} v_1(\tilde{X}_n) \\ v_2(\tilde{X}_n) \end{bmatrix}^\top$ and by $\mathcal{V}, \tilde{\mathcal{V}}$ the subspaces spanned by their columns. Then, by Wedin's $\sin \Theta$ theorem, (Stewart & Sun, 1990, Chapter V, Theorem 4.4.) it holds that

$$d(\mathcal{V}, \tilde{\mathcal{V}}) \leq \frac{\sigma_1(X_n)}{\sigma_2(X_n + P_n) - \sigma_3(X_n + P_n)} = \frac{1}{c \cdot \sqrt{p} + O(1)} = O\left(\frac{1}{\sqrt{p}}\right).$$

As $v_1(P_n), v_2(P_n) \in \mathrm{span}\{\mu_s, \mu_t\}$ and they are linearly independent, this implies that $\mathcal{V} = \mathrm{span}\{\mu_s, \mu_t\}$. Choosing matrices $\tilde{V}_s \in \mathbb{R}^{p \times 2}$ and $\tilde{V}_t \in \mathbb{R}^{p \times 2}$ such that their columns are orthonormal bases of $\tilde{\mathcal{V}}$ and their first column is $\frac{\mu_s}{\|\mu_s\|_2}$ and $\frac{\mu_t}{\|\mu_t\|_2}$ respectively, one gets that

$$\left\| (I - VV^\top) \frac{\mu_s}{\|\mu_s\|_2} \right\|_2 = \left\| (I - VV^\top) \tilde{V}_s e_1 \right\|_2 \leq \left\| (I - VV^\top) \tilde{V}_s \right\|_2 = d(\mathcal{V}, \tilde{\mathcal{V}}) \leq O\left(\frac{1}{\sqrt{p}}\right),$$

$$\left\| (I - VV^\top) \frac{\mu_t}{\|\mu_t\|_2} \right\|_2 = \left\| (I - VV^\top) \tilde{V}_t e_1 \right\|_2 \leq \left\| (I - VV^\top) \tilde{V}_t \right\|_2 = d(\mathcal{V}, \tilde{\mathcal{V}}) \leq O\left(\frac{1}{\sqrt{p}}\right).$$

From this, **1b** directly follows. The case $\varphi = 1$ is handled analogously. $\qquad \square$

**Proposition A.3.** *In the under-parameterized regime, i.e., when $p < n$, it holds that*

$$\frac{1}{n} \mathrm{Tr}[\hat{\Sigma}^+ (\Sigma_t + \mu_t \mu_t^\top)] = \frac{1}{n} \mathrm{Tr}[\hat{\Sigma}_0^+ \Sigma_t] + O\left(\frac{1}{p}\right). \tag{A.6}$$

*Proof.* We break down the LHS of (A.6) into two terms

$$\frac{1}{n} \mathrm{Tr}[\hat{\Sigma}^+ (\Sigma_t + \mu_t \mu_t^\top)] = T_1 + T_2,$$

where

$$T_1 = \frac{1}{n} \mathrm{Tr}[\hat{\Sigma}^+ \Sigma_t], \qquad \text{and} \qquad T_2 = \frac{1}{n} \mathrm{Tr}[\hat{\Sigma}^+ \mu_t \mu_t^\top].$$

We will deal with each of the terms individually.

**Bounding the term $T_1$.** It holds that

$$
\begin{aligned}
T_1 &= \frac{1}{n} \mathrm{Tr}(\hat{\Sigma}^+ \Sigma_t) \\
&= \frac{1}{n} \mathrm{Tr} \left( \left( \left( \frac{X \Sigma_t^{-1/2}}{\sqrt{n}} \right)^\top \frac{X \Sigma_t^{-1/2}}{\sqrt{n}} \right)^{-1} \right) \\
&= \frac{1}{n} \mathrm{Tr} \left( (\bar{X}^\top \bar{X})^{-1} \right) \\
&= \frac{1}{n} \sum_{i=1}^k \frac{1}{\sigma_i^2(\bar{X})},
\end{aligned}
\tag{A.7}
$$

where $\bar{X} := \frac{X \Sigma_t^{-1/2}}{\sqrt{n}} \in \mathbb{R}^{n \times p}$, and $k \leq p$ is the number of non-zero singular values of $\bar{X}$. Let us prove that $\sigma_p(\bar{X}) > c$ for some constant $c$, implying that $k = p$. Towards this end, we write out

$$\bar{X}^\top \bar{X} = \bar{X}_s^\top \bar{X}_s + \bar{X}_t^\top \bar{X}_t,$$

where $\bar{X}_s := \frac{X_s \Sigma_t^{-1/2}}{\sqrt{n}} = \frac{\left(Z_s \Sigma_s^{1/2} + 1_{n_s} \mu_s^\top\right) \Sigma_t^{-1/2}}{\sqrt{n}}$ and $\bar{X}_t := \frac{X_t \Sigma_t^{-1/2}}{\sqrt{n}} = \frac{\left(Z_t \Sigma_t^{1/2} + 1_{n_t} \mu_t^\top\right) \Sigma_t^{-1/2}}{\sqrt{n}}$.
From Proposition A.1, it follows that for large enough $n$, almost surely

$$\sigma_p(\bar{X}_s) \geq c, \qquad \sigma_p(\bar{X}_t) \geq c,$$

for some constant $c$, which is just $c(\gamma)$ from the proposition adjusted by the bound on the eigenvalues of $\Sigma_t^{-1/2}$ and $\Sigma_s^{-1/2}$ (recall that the smallest eigenvalue of $\Sigma_s, \Sigma_t$ is lower bounded by $\tau$). Plugging this in gives

$$\sigma_p(\bar{X})^2 \geq \sigma_p(\bar{X}_s)^2 + \sigma_p(\bar{X}_t)^2 \geq 2c^2. \tag{A.8}$$

Let $\bar{X}^0 := \frac{X^0 \Sigma_t^{-1/2}}{\sqrt{n}}$ and note that $\bar{X}$ is a rank-2 perturbation of $\bar{X}^0$ (see (A.2)). Then, due to Weyl's inequality for singular values, it holds that, for $i \in \{3, \ldots, p-2\}$,

$$\sigma_{i+2}(\bar{X}^0) \leq \sigma_i(\bar{X}) \leq \sigma_{i-2}(\bar{X}^0).$$

Therefore, we have

$$\frac{1}{n} \sum_{i=1}^{p-4} \frac{1}{\sigma_i \left(\bar{X}^0\right)^2} \leq \frac{1}{n} \sum_{i=3}^{p-2} \frac{1}{\sigma_i \left(\bar{X}\right)^2} \leq \frac{1}{n} \sum_{i=3}^{p} \frac{1}{\sigma_i \left(\bar{X}^0\right)^2}.$$

An application of the Bai–Yin theorem (Bai & Silverstein, 2010, Theorem 5.11) gives that there exist constants $a$ and $b$ such that

$$0 < a < \sigma_p(\bar{X}^0) \leq \sigma_1(\bar{X}^0) < b < +\infty,$$

for large enough $n$. Therefore, it holds

$$\frac{1}{n} \sum_{i=1}^{p} \frac{1}{\sigma_i \left(\bar{X}^0\right)^2} - O\left(\frac{1}{n}\right) \leq \frac{1}{n} \sum_{i=3}^{p-2} \frac{1}{\sigma_i \left(\bar{X}\right)^2} \leq \frac{1}{n} \sum_{i=1}^{p} \frac{1}{\sigma_i \left(\bar{X}^0\right)^2},$$

which implies that

$$\frac{1}{n} \sum_{i=3}^{p-2} \frac{1}{\sigma_i \left(\bar{X}\right)^2} = \frac{1}{n} \sum_{i=1}^{p} \frac{1}{\sigma_i \left(\bar{X}^0\right)^2} + \Theta\left(\frac{1}{n}\right).$$

Using the proved fact that $\sigma_i(\bar{X}) > c$ we have

$$\frac{1}{n} \sum_{i=1}^{p} \frac{1}{\sigma_i \left(\bar{X}\right)^2} = \frac{1}{n} \sum_{i=3}^{p-2} \frac{1}{\sigma_i \left(\bar{X}\right)^2} + O\left(\frac{1}{n}\right).$$

Combining all the pieces, it holds that

$$\begin{aligned}
T_1 &= \frac{1}{n} \sum_{i=1}^{p} \frac{1}{\sigma_i^2(\bar{X})} \\
&= \frac{1}{n} \sum_{i=3}^{p-2} \frac{1}{\sigma_i(\bar{X})^2} + O\left(\frac{1}{n}\right) \\
&= \frac{1}{n} \sum_{i=1}^{p} \frac{1}{\sigma_i(\bar{X}^0)^2} + O\left(\frac{1}{n}\right) \\
&= \frac{1}{n} \operatorname{Tr}\left(\left(\bar{X}^{0\top} \bar{X}^0\right)^{-1} \Sigma_t\right) + O\left(\frac{1}{n}\right) \\
&= \frac{1}{n} \operatorname{Tr}[\hat{\Sigma}_0^+ \Sigma_t] + O\left(\frac{1}{p}\right).
\end{aligned}$$

**Bounding the term $T_2$.** First, recall the shorthand $\tilde{X}_n = X/\sqrt{n}$ and note that

$$\sigma_p(\tilde{X}_n) = \sigma_p(\bar{X}\Sigma_t^{1/2}) \geq \sigma_p(\bar{X}) \cdot \sigma_p(\Sigma_t^{1/2}) \geq c \cdot \tau, \tag{A.9}$$

where the last inequality follows from (A.8) and the bounds on the spectrum of $\Sigma_t$. Recall that $n/p = \gamma$, which implies $O\left(\frac{1}{n}\right) = O\left(\frac{1}{p}\right)$. Then, it holds that

$$
\begin{aligned}
T_2 &= \frac{1}{n}\mu_t^\top \hat{\Sigma}^+ \mu_t \\
&= \frac{\mu_t^\top}{\sqrt{n}}(\tilde{X}_n^\top \tilde{X}_n)^+ \frac{\mu_t}{\sqrt{n}} \\
&= \frac{\mu_t^\top}{\sqrt{n}} \sum_{i=1}^p \frac{1}{\sigma_i(\tilde{X}_n)^2} v_i(\tilde{X}_n) v_i(\tilde{X}_n)^\top \frac{\mu_t}{\sqrt{n}} \\
&= \frac{1}{\sigma_1(\tilde{X}_n)^2} \frac{\left\langle v_1(\tilde{X}_n), \mu_t \right\rangle^2}{n} + \frac{1}{\sigma_2(\tilde{X}_n)^2} \frac{\left\langle v_2(\tilde{X}_n), \mu_t \right\rangle^2}{n} + \sum_{i=3}^p \frac{1}{\sigma_i(\tilde{X}_n)^2} \frac{\left\langle v_i(\tilde{X}_n), \mu_t \right\rangle^2}{n} \\
&\leq \Theta\left(\frac{1}{p}\right)\left(1 - O\left(\frac{1}{p}\right)\right) + \frac{1}{c \cdot \tau}O\left(\frac{1}{p}\right) \\
&= O\left(\frac{1}{p}\right),
\end{aligned}
$$

where the penultimate inequality follows directly from (A.9) and Proposition A.2.

Finally, combining the bounds on the two terms we get

$$T_1 + T_2 = \frac{1}{n}\operatorname{Tr}[\hat{\Sigma}_0^+ \Sigma_t] + O\left(\frac{1}{p}\right),$$

proving the claim. $\qquad\square$

We conclude this appendix with the proof of Theorem 4.1.

**Proof of Theorem 4.1.** As proved in Section 4.1, it holds that $B_X(\hat{\beta}; \beta) = 0$, from which follows

$$R_X(\hat{\beta}, \beta) = V_X(\hat{\beta}; \beta) = \frac{\sigma^2}{n}\operatorname{Tr}[\hat{\Sigma}^+(\Sigma_t + \mu_t \mu_t^\top)].$$

By directly applying Proposition A.3, it holds

$$\frac{\sigma^2}{n}\operatorname{Tr}[\hat{\Sigma}^+(\Sigma_t + \mu_t \mu_t^\top)] = \frac{\sigma^2}{n}\operatorname{Tr}[\hat{\Sigma}_0^+ \Sigma_t] + O\left(\frac{1}{p}\right),$$

where $\hat{\Sigma}_0 = \frac{X^{0\top} X^0}{\sqrt{n}}$. Plugging in the expression of $\frac{\sigma^2}{n}\operatorname{Tr}[\hat{\Sigma}_0^+ \Sigma_t]$ given in (Yang et al., 2025, Theorem 3) gives the desired result. $\qquad\square$

## A.2 Proof of Proposition 4.2

Since we are in the setting where $n > p$, it holds that $B_X(\hat{\beta}; \beta) = 0$, which implies

$$R_X(\hat{\beta}, \beta) = V_X(\hat{\beta}; \beta) = \frac{\sigma^2}{n}\operatorname{Tr}[\hat{\Sigma}^+(\Sigma_t + \mu_t \mu_t^\top)].$$

Note that $\gamma_t = 0$ implies that $X_t = 0$ and $X = X_s$. We also note that (A.8) still holds for $X_t = 0$, implying that $\hat{\Sigma}$ is of rank $p$ almost surely and, therefore, invertible. Thus, it holds

$$\operatorname{Tr}[\hat{\Sigma}^+(\Sigma_t + \mu_t \mu_t^\top)] = \operatorname{Tr}[\hat{\Sigma}^{-1}(\Sigma_t + \mu_t \mu_t^\top)].$$

To simplify exposition, we break this down into two terms

$$R_X(\hat{\beta}, \beta) = V_1 + V_2,$$

with $V_1 := \frac{\sigma^2}{n}\operatorname{Tr}[\hat{\Sigma}^{-1}\Sigma_t]$, $V_2 := \frac{\sigma^2}{n}\operatorname{Tr}[\hat{\Sigma}^{-1}\mu_t \mu_t^\top]$, and treat each of them separately.

**Bounding the term** $V_2$. Note that $\gamma_t = 0$ implies $n = n_s$, so we will use these two values interchangeably throughout the proof. From the cyclic property of trace, we have

$$V_2 = \frac{\sigma^2}{n} \operatorname{Tr}[\hat{\Sigma}^{-1} \mu_t \mu_t^\top] = \sigma^2 \frac{\mu_t^\top \hat{\Sigma}^{-1} \mu_t}{n}.$$

Note that

$$\mu_t^\top \hat{\Sigma}^{-1} \mu_t^\top = \mu_t^\top \left( \frac{X^\top X}{n} \right)^{-1} \mu_t$$

$$= \mu_t^\top \left( \frac{(Z_s \Sigma_s^{1/2} + 1_{n_s} \mu_s^\top)^\top (Z_s \Sigma_s^{1/2} + 1_{n_s} \mu_s^\top)}{n} \right)^{-1} \mu_t$$

$$= \left( \Sigma_s^{-1/2} \mu_t \right)^\top \left( \frac{\left( Z_s + 1_{n_s} \left( \Sigma_s^{-1/2} \mu_s \right)^\top \right)^\top \left( Z_s + 1_{n_s} \left( \Sigma_s^{-1/2} \mu_s \right)^\top \right)}{n} \right)^{-1} \left( \Sigma_s^{-1/2} \mu_t \right)$$

$$= \mu_t'^\top \hat{\Sigma}'^{-1} \mu_t',$$

where we use the notation $\mu_t' := \Sigma_s^{-1/2} \mu_t$, $\mu_s' := \Sigma_s^{-1/2} \mu_s$ and $\hat{\Sigma}' := \frac{\left( Z_s + 1_{n_s} \mu_s'^\top \right)^\top \left( Z_s + 1_{n_s} \mu_s'^\top \right)}{n}$. Note that due to the assumed bound on the spectrum of $\Sigma_s$ it holds that $\|\mu_t'\|_2 = O(\sqrt{p})$ and $\|\mu_s'\|_2 = O(\sqrt{p})$. Next, let us break down the vector $\mu_t'$ into its orthogonal projection onto the subspace $\{\mu_s'\}$ and $\{\mu_s'\}^\perp$ as

$$\mu_t' = \mu_{t\|s}' + \mu_{t\perp s}', \quad \text{where} \quad \mu_{t\|s}' := \frac{\langle \mu_t', \mu_s' \rangle}{\|\mu_s'\|_2^2} \mu_s', \quad \mu_{t\perp s}' := \mu_t' - \mu_{t\|s}'. \tag{A.11}$$

Moreover, as a decomposition into orthogonal spaces, it holds $\|\mu_{t\|s}'\|_2^2 + \|\mu_{t\perp s}'\|_2^2 = \|\mu_t'\|_2^2 = O(p)$. By using this decomposition, we will shift the focus from $\mu_t'$ to $\mu_{t\perp s}'$. Namely, it holds

$$V_2 = \sigma^2 \frac{\mu_t'^\top \hat{\Sigma}'^{-1} \mu_t'}{n} = \sigma^2 \frac{(\mu_{t\|s}' + \mu_{t\perp s}')^\top \hat{\Sigma}'^{-1} (\mu_{t\|s}' + \mu_{t\perp s}')}{n}$$

$$= \sigma^2 \frac{\mu_{t\perp s}'^\top}{\sqrt{n}} \hat{\Sigma}'^{-1} \frac{\mu_{t\perp s}'}{\sqrt{n}} + 2\sigma^2 \frac{\mu_{t\perp s}'^\top}{\sqrt{n}} \hat{\Sigma}'^{-1} \frac{\mu_{t\|s}'}{\sqrt{n}} + \frac{\mu_{t\|s}'^\top}{\sqrt{n}} \hat{\Sigma}'^{-1} \frac{\mu_{t\|s}'}{\sqrt{n}}$$

$$= \sigma^2 \frac{\mu_{t\perp s}'^\top}{\sqrt{n}} \hat{\Sigma}'^{-1} \frac{\mu_{t\perp s}'}{\sqrt{n}} + O\left( \frac{1}{\sqrt{p}} \right), \tag{A.12}$$

where the last line follows from derivations analogous to the ones around (A.10), this time applying case 2. of Proposition A.2. To ease further exposition, we introduce $\tilde{\mu}_{t\perp s} := \frac{\mu_{t\perp s}'}{\sqrt{n}}$, noting that $\|\tilde{\mu}_{t\perp s}\|_2 = O(1)$.

In order to bound $V_2$, we relate $\hat{\Sigma}'^{-1}$ to its zero-mean counterpart, as it is easier to work with mean zero data. Towards this end, we write out $\hat{\Sigma}'$ as

$$\hat{\Sigma}' = \frac{\left( Z_s + 1_{n_s} \mu_s'^\top \right)^\top \left( Z_s + 1_{n_s} \mu_s'^\top \right)}{n}$$

$$= \left( \frac{Z_s^\top Z_s}{n} + \frac{Z_s^\top 1_{n_s} \mu_s'^\top}{n} + \frac{\mu_s' 1_{n_s}^\top Z_s}{n} + \mu_s' \mu_s'^\top \right)$$

$$= \left( \hat{\Sigma}_0' + \frac{Z_s^\top 1_{n_s} \mu_s'^\top}{n} + \frac{\mu_s' 1_{n_s}^\top Z_s}{n} + \mu_s' \mu_s'^\top \right),$$

for $\hat{\Sigma}'_0 := \frac{Z_s^\top Z_s}{n}$. All the terms above, except the first one, have rank 1, so we use Woodbury formula to take them out of the inverse when computing $\hat{\Sigma}'$. We introduce the following notation

$$u := \frac{\mu'_s}{\sqrt{n}}, \qquad v := \frac{Z_s^\top 1_{n_s}}{\sqrt{n}},$$

$$U := [u \;\; v] \in \mathbb{R}^{p \times 2}, \;\; \text{and } C := \begin{bmatrix} n & 1 \\ 1 & 0 \end{bmatrix} \in \mathbb{R}^{2 \times 2}.$$

Under this notation it holds

$$\frac{Z_s^\top 1_{n_s} \mu'^\top_s}{n} + \frac{\mu'_s 1^\top_{n_s} Z_s}{n} + \mu'_s \mu'^\top_s = UCU^\top.$$

Then, using Woodbury formula, we have

$$
\begin{aligned}
\hat{\Sigma}'^{-1} &= \left(\hat{\Sigma}'_0 + uv^\top + vu^\top + nuu^\top\right)^{-1} \\
&= \left(\hat{\Sigma}'_0 + UCU^\top\right)^{-1} \\
&= \hat{\Sigma}'^{-1}_0 - \hat{\Sigma}'^{-1}_0 U \left(C^{-1} - U^\top \hat{\Sigma}'^{-1}_0 U\right)^{-1} U^\top \hat{\Sigma}'^{-1}_0.
\end{aligned}
$$

We now compute the $2 \times 2$ block

$$C^{-1} - U^\top \hat{\Sigma}'^{-1}_0 U = \begin{bmatrix} -u^\top \hat{\Sigma}'^{-1}_0 u & 1 - u^\top \hat{\Sigma}'^{-1}_0 v \\ 1 - v^\top \hat{\Sigma}'^{-1}_0 u & -n - v^\top \hat{\Sigma}'^{-1}_0 v \end{bmatrix} = \begin{bmatrix} -a & 1-b \\ 1-b & -n-d \end{bmatrix},$$

where

$$a := u^\top \hat{\Sigma}'^{-1}_0 u, \qquad b := v^\top \hat{\Sigma}'^{-1}_0 u = u^\top \hat{\Sigma}'^{-1}_0 v, \qquad d := v^\top \hat{\Sigma}'^{-1}_0 v.$$

Hence

$$\left(C^{-1} - U^\top \hat{\Sigma}'^{-1}_0 U\right)^{-1} = \frac{1}{\Delta} \begin{bmatrix} -n-d & b-1 \\ b-1 & -a \end{bmatrix}, \qquad \Delta := a(n+d) - (1-b)^2.$$

Plugging back and simplifying gives the explicit formula:

$$\hat{\Sigma}'^{-1} = \hat{\Sigma}'^{-1}_0 - \frac{1}{\Delta} \hat{\Sigma}'^{-1}_0 \left((-n-d)\, uu^\top - (1-b)(uv^\top + vu^\top) - a\, vv^\top\right) \hat{\Sigma}'^{-1}_0,$$

which is valid whenever $\Delta \neq 0$, i.e., whenever $C^{-1} - U^\top \hat{\Sigma}'^{-1}_0 U$ is invertible.

We will now analyze the $a, b, d$ terms. First, for some constants $c_1, c_2 > 0$ it holds almost surely that

$$c_2 > \lambda_1(\hat{\Sigma}'_0) \geq \lambda_p(\hat{\Sigma}'_0) \geq c_1 > 0, \tag{A.13}$$

which follows from Bai–Yin theorem (Bai & Silverstein, 2010, Theorem 5.11), as $Z_s$ has i.i.d entries with mean zero, unit variance and bounded fourth moments. From this, it follows that

$$
\begin{aligned}
|a| &= \left| u^\top \hat{\Sigma}'^{-1}_0 u \right| \\
&= \left\| \frac{\mu'^\top_s}{\sqrt{n}} \hat{\Sigma}'^{-1}_0 \frac{\mu'_s}{\sqrt{n}} \right\|_2 \\
&\leq \left\| \frac{\mu'_s}{\sqrt{n}} \right\|_2 \|\hat{\Sigma}'^{-1}_0\|_2 \left\| \frac{\mu'_s}{\sqrt{n}} \right\|_2 \\
&\leq c,
\end{aligned}
$$

as well as

$$|a| \geq c \cdot (\lambda_1(\hat{\Sigma}'_0))^{-1} \geq c > 0,$$

Similarly, we have

$$|b| = \left| v^\top \hat{\Sigma}'^{-1}_0 u \right| = \left\| \frac{\mu'^\top_s}{\sqrt{n}} \hat{\Sigma}'^{-1}_0 \frac{Z_s^\top 1_{n_s}}{\sqrt{n}} \right\|_2 \leq \left\| \frac{\mu'_s}{\sqrt{n}} \right\|_2 \|\hat{\Sigma}'^{-1}_0\|_2 \left\| \frac{Z_s^\top 1_{n_s}}{\sqrt{n}} \right\|_2 \leq c\sqrt{p}, \tag{A.14}$$

where the last inequality follows with high probability over the sampling of $Z_s$, since $\frac{Z_s^\top 1_{n_t}}{\sqrt{n}}$ is a vector with $p$ i.i.d entries of mean zero and $O(1)$ variance. Finally, we have

$$
\begin{aligned}
|d| &= \left| v^\top \hat{\Sigma}_0'^{-1} v \right| \\
&= \left\| \frac{1_{n_s}^\top Z_s}{\sqrt{n}} \hat{\Sigma}_0'^{-1} \frac{Z_s^\top 1_{n_s}}{\sqrt{n}} \right\|_2 \\
&\leq \left\| \frac{Z_s^\top 1_{n_s}}{\sqrt{n}} \right\|_2 \| \hat{\Sigma}_0'^{-1} \|_2 \left\| \frac{Z_s^\top 1_{n_s}}{\sqrt{n}} \right\|_2 \\
&\leq cp,
\end{aligned}
$$

again with high probability.

We can now prove that, with high probability, $\Delta = \Omega(p)$. Using Cauchy-Schwarz, it holds that

$$
b^2 = |\langle u, v \rangle|_{A^{-1}} \leq \|u\|_{A^{-1}} \|v\|_{A^{-1}} = ad,
$$

from which it follows that

$$
\Delta = a(n+d) - (1-b)^2 \geq an - 1 + 2b = \Omega(p), \tag{A.15}
$$

since $a$ is lower bounded by a constant and $|b| \leq c\sqrt{p}$.

Turning back to the value of interest, we write out

$$
\begin{aligned}
\tilde{\mu}_{t\perp s}^\top \hat{\Sigma}'^{-1} \tilde{\mu}_{t\perp s} &= \tilde{\mu}_{t\perp s}^\top \hat{\Sigma}_0'^{-1} \tilde{\mu}_{t\perp s} \\
&\quad - \tilde{\mu}_{t\perp s}^\top \frac{1}{\Delta} \hat{\Sigma}_0'^{-1} \left( (-n-d)uu^\top - (1-b)(uv^\top + vu^\top) - a\, vv^\top \right) \hat{\Sigma}_0'^{-1} \tilde{\mu}_{t\perp s} \\
&= \tilde{\mu}_{t\perp s}^\top \hat{\Sigma}_0'^{-1} \tilde{\mu}_{t\perp s} + T_{u,u} + T_{u,v} + T_{v,v},
\end{aligned}
$$

where $T_{u,u}$ is the summand corresponding to $uu^\top$, $T_{u,v}$ to $uv^\top + vu^\top$, and $T_{v,v}$ to $vv^\top$. We will prove that each of these terms, except for $\tilde{\mu}_{t\perp s}^\top \hat{\Sigma}_0'^{-1} \tilde{\mu}_{t\perp s}$, is vanishing.

First, we state a useful claim, that for arbitrary deterministic unit vectors $w_1 \in \mathbb{R}^p$ and $w_2 \in \mathbb{R}^p$ it holds with overwhelming probability

$$
w_1^\top \hat{\Sigma}_0'^{-1} w_2 = \frac{\gamma}{\gamma - 1} \langle w_1, w_2 \rangle + O\left(n^{-c_1}\right), \tag{A.16}
$$

for some constant $c_1 > 0$.

**Proof of claim in (A.16).**  The result follows directly from (Yang et al., 2025, Theorem 27). For clarity, we refer to the relevant parts of Section B.3.1 of that work. While Theorem 27 is stated in the more general anisotropic setting, it specializes to our isotropic case by taking $\Lambda$, $U$ and $V$ from (B.3) from their work to be the identity. Substituting these choices into equation (B.6) from Yang et al. (2025) for $z = 0$, implies

$$
\alpha_1(0) + \alpha_2(0) = 1 - \frac{p}{n} = \frac{\gamma - 1}{\gamma}.
$$

Substituting this into (B.7) and applying Theorem 27 from the mentioned paper, yields with overwhelming probability

$$
\left| w_1^\top \hat{\Sigma}_0'^{-1} w_2 - w_1^\top \frac{\gamma}{\gamma - 1} I_p w_2 \right| \leq n^{-c_1},
$$

for any $c_1 < -1/2 + 2/\psi$. Recalling that $Z_s$ has its $\psi$-th moment bounded for $\psi > 4$, implies $c_1 > 0$. ♣

We can now use (A.16) to tackle the terms $T_{u,u}$ and $T_{u,v}$. Namely, we have that

$$
\begin{aligned}
\tilde{\mu}_{t\perp s}^\top \hat{\Sigma}_0'^{-1} u &= \|\tilde{\mu}_{t\perp s}\|_2 \|u\|_2 \left( \frac{\gamma}{\gamma - 1} \langle \tilde{\mu}_{t\perp s}, u \rangle + O(n^{-c_1}) \right) \\
&= c \left( \frac{\gamma}{\gamma - 1} \left\langle \tilde{\mu}_{t\perp s}, \frac{\mu_s'}{\sqrt{n}} \right\rangle + O(n^{-c_1}) \right) \\
&= O(n^{-c_1}),
\end{aligned}
$$

with high probability. From this, it follows that

$$T_{u,u} = \frac{n+d}{\Delta} \tilde{\mu}_{t\perp s} \hat{\Sigma}_0'^{-1} uu^\top \hat{\Sigma}_0'^{-1} \tilde{\mu}_{t\perp s} = O(n^{-2c_1}). \tag{A.17}$$

Similarly,

$$\begin{aligned}
|T_{u,v}| &= \left| \frac{1-b}{\Delta} \tilde{\mu}_{t\perp s} \hat{\Sigma}_0'^{-1} \left( uv^\top + vu^\top \right) \hat{\Sigma}_0'^{-1} \tilde{\mu}_{t\perp s} \right| \\
&= \left| 2 \left( \tilde{\mu}_{t\perp s} \hat{\Sigma}_0'^{-1} u \right) \cdot \left( \frac{1-b}{\Delta} v \hat{\Sigma}_0'^{-1} \tilde{\mu}_{t\perp s} \right) \right| \\
&\leq O(n^{-c_1}) \cdot \left\| \frac{1-b}{\Delta} \frac{Z_s^\top 1_{n_s}}{\sqrt{n}} \right\|_2 \left\| \hat{\Sigma}_0'^{-1} \right\|_2 \| \tilde{\mu}_{t\perp s} \|_2 \\
&= O(n^{-c_1}),
\end{aligned} \tag{A.18}$$

where the last inequality holds with high probability due to (A.13), (A.14), and (A.15).

Let us denote by $\tilde{1}_{n_s} := \frac{1_{n_s}}{\sqrt{n}}$ and turn to the term $T_{v,v}$. Notice that

$$\begin{aligned}
T_{v,v} &= \frac{a}{\Delta} \tilde{\mu}_{t\perp s} \hat{\Sigma}_0'^{-1} vv^\top \hat{\Sigma}_0'^{-1} \tilde{\mu}_{t\perp s} \\
&= \frac{n a}{\Delta} \left( \frac{1_{n_s}^\top}{\sqrt{n}} \frac{Z_s}{\sqrt{n}} \left( \frac{Z_s^\top Z_s}{n} \right)^{-1} \tilde{\mu}_{t\perp s} \right)^2 \\
&= c \left( \tilde{1}_{n_s}^\top \frac{Z_s}{\sqrt{n}} \left( \frac{Z_s^\top Z_s}{n} \right)^{-1} \tilde{\mu}_{t\perp s} \right)^2.
\end{aligned} \tag{A.19}$$

Let us introduce a matrix $Q = [q_1 \quad \cdots \quad q_p] \in \mathbb{R}^{p \times p}$, whose columns form an orthonormal basis, such that $q_1 = \frac{\tilde{\mu}_{t\perp s}}{\| \tilde{\mu}_{t\perp s} \|_2}$. Then, we have that

$$\begin{aligned}
\tilde{1}_{n_s}^\top \frac{Z_s}{\sqrt{n}} \left( \frac{Z_s^\top Z_s}{n} \right)^{-1} \tilde{\mu}_{t\perp s} &= \tilde{1}_{n_s}^\top \frac{Z_s}{\sqrt{n}} QQ^\top \left( \frac{Z_s^\top Z_s}{n} \right)^{-1} \tilde{\mu}_{t\perp s} \\
&= \sum_{k=1}^p \tilde{1}_{n_s}^\top \frac{Z_s}{\sqrt{n}} q_k \cdot q_k^\top \left( \frac{Z_s^\top Z_s}{n} \right)^{-1} \tilde{\mu}_{t\perp s}.
\end{aligned} \tag{A.20}$$

Using (A.16) and a union bound, it holds with overwhelming probability that

$$q_k \left( \frac{Z_s^\top Z_s}{n} \right)^{-1} \tilde{\mu}_{t\perp s} = \frac{\gamma}{\gamma-1} \langle q_k, \tilde{\mu}_{t\perp s} \rangle + O(n^{-c_1}) = \begin{cases} \frac{\gamma}{\gamma-1} \| \tilde{\mu}_{t\perp s} \|_2 + O(n^{-c_1}), & k = 1, \\ O(n^{-c_1}), & k > 1. \end{cases}$$

Plugging this into (A.20) yields

$$\tilde{1}_{n_s}^\top \frac{Z_s}{\sqrt{n}} \left( \frac{Z_s^\top Z_s}{n} \right)^{-1} \tilde{\mu}_{t\perp s} = \tilde{1}_{n_s}^\top \frac{Z_s}{\sqrt{n}} \tilde{\mu}_{t\perp s} \cdot \frac{\gamma}{\gamma-1} + O(n^{-c_1}) \cdot \sum_{k=1}^p \tilde{1}_{n_s}^\top \frac{Z_s}{\sqrt{n}} q_k. \tag{A.21}$$

Let us first analyze the mean and variance of the random variable $\tilde{1}_{n_s}^\top \frac{Z_s}{\sqrt{n}} \tilde{\mu}_{t\perp s}$, namely,

$$\mathbb{E} \left[ \tilde{1}_{n_s}^\top \frac{Z_s}{\sqrt{n}} \tilde{\mu}_{t\perp s} \right] = \mathbb{E} \left[ \frac{1}{\sqrt{n}} \sum_{i=1}^n \sum_{j=1}^p Z_{i,j} (\tilde{1}_{n_s})_i (\tilde{\mu}_{t\perp s})_j \right] = 0,$$

$$\begin{aligned}
\mathrm{Var} \left( \tilde{1}_{n_s}^\top \frac{Z_s}{\sqrt{n}} \tilde{\mu}_{t\perp s} \right) &= \mathrm{Var} \left( \frac{1}{\sqrt{n}} \sum_{i=1}^n \sum_{j=1}^p Z_{i,j} (\tilde{1}_{n_s})_i (\tilde{\mu}_{t\perp s})_j \right) \\
&= \frac{1}{n} \| \tilde{1}_{n_s} \|_2^2 \| \tilde{\mu}_{t\perp s} \|_2^2 = O \left( \frac{1}{n} \right).
\end{aligned}$$

Therefore, using Chebyshev inequality, we have that

$$\left| \tilde{1}_{n_s}^\top \frac{Z_s}{\sqrt{n}} \tilde{\mu}_{t\perp s} \right| = O\left(n^{-c_2}\right),$$

with high probability, for some constant $1/2 > c_2 > 0$. Similarly, we calculate the mean and variance of the random variable $\sum_{k=1}^p \tilde{1}_{n_s}^\top \frac{Z_s}{\sqrt{n}} q_k$ as

$$\mathbb{E}\left[ \sum_{k=1}^p \tilde{1}_{n_s}^\top \frac{Z_s}{\sqrt{n}} q_k \right] = \mathbb{E}\left[ \frac{1}{\sqrt{n}} \sum_{k=1}^p \sum_{i=1}^n \sum_{j=1}^p Z_{i,j} (\tilde{1}_{n_s})_i (q_k)_j \right] = 0,$$

$$\mathrm{Var}\left( \sum_{k=1}^p \tilde{1}_{n_s}^\top \frac{Z_s}{\sqrt{n}} q_k \right) = \mathrm{Var}\left( \frac{1}{\sqrt{n}} \sum_{k=1}^p \sum_{i=1}^n \sum_{j=1}^p Z_{i,j} (\tilde{1}_{n_s})_i (q_k)_j \right)$$

$$= \frac{1}{n} \sum_{k=1}^p \left\| \tilde{1}_{n_s} \right\|_2^2 \left\| q_k \right\|_2^2 = O(1).$$

Again, Chebyshev inequality implies

$$\left| O(n^{-c_1}) \cdot \sum_{k=1}^p \tilde{1}_{n_s}^\top \frac{Z_s}{\sqrt{n}} q_k \right| = O\left(n^{-c_1/2}\right),$$

with high probability. Plugging the obtained results into (A.21) and using a union bound on the probabilities, we get that

$$\left| \tilde{1}_{n_s}^\top \frac{Z_s}{\sqrt{n}} \left( \frac{Z_s^\top Z_s}{n} \right)^{-1} \tilde{\mu}_{t\perp s} \right| \leq \left| O(n^{-c_1}) \cdot \sum_{k=1}^p \tilde{1}_{n_s}^\top \frac{Z_s}{\sqrt{n}} q_k \right| + \left| \tilde{1}_{n_s}^\top \frac{Z_s}{\sqrt{n}} \tilde{\mu}_{t\perp s} \right|$$

$$= O\left(n^{-c_1/2}\right),$$

with high probability. Then, we directly obtain a bound for (A.19) in the form of

$$T_{v,v} = O(n^{-c_1}), \tag{A.22}$$

which holds for some constant $c_1 > 0$ with high probability. Combining the bound in (A.16) and the three bounds on the terms (A.17), (A.18) an (A.22), we get

$$\tilde{\mu}_{t\perp s} \hat{\Sigma}'^{-1} \tilde{\mu}_{t\perp s} = \frac{\gamma}{\gamma - 1} \left\| \tilde{\mu}_{t\perp s} \right\|_2^2 + O(n^{-c}), \tag{A.23}$$

for some $c > 0$. Using this in (A.12) yields

$$V_2 = \sigma^2 \frac{\gamma}{\gamma - 1} \left\| \tilde{\mu}_{t\perp s} \right\|_2^2 + O(n^{-c}),$$

with high probability. Lastly, note that

$$\left\| \tilde{\mu}_{t\perp s} \right\|_2^2 = \frac{1}{n} \left\| \mu'_{t\perp s} \right\|_2^2 = \frac{1}{n} \left( \left\| \mu'_t \right\|_2^2 - \left\| \mu'_{t\|s} \right\|_2^2 \right) = \frac{1}{n} \left( \left\| \Sigma_s^{-1/2} \mu_t \right\|_2^2 - \left( \frac{\mu_t^\top \Sigma_s^{-1} \mu_s}{\left\| \Sigma_s^{-1/2} \mu_s \right\|_2} \right)^2 \right).$$

**Bounding the term $V_1$.** By following exactly the proof of the bound of the term $T_1$ in Proposition A.3, one directly gets the same conclusion that

$$V_1 = \frac{\sigma^2}{n} \mathrm{Tr}[\hat{\Sigma}_0^{-1} \Sigma_t] + O\left(\frac{1}{p}\right).$$

Notice that

$$\hat{\Sigma}_0 = \frac{X^\top X}{n} = \frac{X_s^\top X_s}{n} = \frac{\Sigma_s^{1/2} Z_s^\top Z_s \Sigma_s^{1/2}}{n}.$$

Thus,

$$\frac{\sigma^2}{n}\operatorname{Tr}\left[\hat{\Sigma}_0^{-1}\Sigma_t\right] = \frac{\sigma^2}{n}\operatorname{Tr}\left[\Sigma_s^{-1/2}\left(\frac{Z_s^\top Z_s}{n}\right)^{-1}\Sigma_s^{-1/2}\Sigma_t\right] = \frac{\sigma^2}{n}\operatorname{Tr}\left[\Sigma_s^{-1/2}\Sigma_t\Sigma_s^{-1/2}\left(\frac{Z_s^\top Z_s}{n}\right)^{-1}\right].$$

Let us write the SVD of $\Sigma_s^{-1/2}\Sigma_t\Sigma_s^{-1/2}$ as

$$\Sigma_s^{-1/2}\Sigma_t\Sigma_s^{-1/2} = \sum_{i=1}^p \lambda_i(\Sigma_s^{-1/2}\Sigma_t\Sigma_s^{-1/2})w_i w_i^\top,$$

where $w_i := v_i(\Sigma_s^{-1/2}\Sigma_t\Sigma_s^{-1/2})$. Then it holds with overwhelming probability

$$\begin{aligned}
\frac{\sigma^2}{n}\operatorname{Tr}\left[\Sigma_s^{-1/2}\Sigma_t\Sigma_s^{-1/2}\left(\frac{Z_s^\top Z_s}{n}\right)^{-1}\right] &= \frac{\sigma^2}{n}\sum_{i=1}^p \lambda_i(\Sigma_s^{-1/2}\Sigma_t\Sigma_s^{-1/2})w_i^\top\left(\frac{Z_s^\top Z_s}{n}\right)^{-1}w_i \\
&= \frac{\sigma^2}{n}\sum_{i=1}^p \lambda_i(\Sigma_s^{-1/2}\Sigma_t\Sigma_s^{-1/2})\frac{\gamma}{\gamma-1}\left(\|w_i\|_2^2 + O(n^{-c})\right) \\
&= \left(\frac{\sigma^2}{n}\frac{\gamma}{\gamma-1}\sum_{i=1}^p \lambda_i(\Sigma_s^{-1/2}\Sigma_t\Sigma_s^{-1/2})\right) + O(n^{-c}) \\
&= \frac{\sigma^2}{n}\frac{\gamma}{\gamma-1}\operatorname{Tr}\left(\Sigma_t\Sigma_s^{-1}\right) + O(n^{-c}),
\end{aligned}$$

where the second line holds with overwhelming probability by using (A.16) and the union bound. The previous bound also holds with high probability, since overwhelming probability implies it.

Finally, by combining the bounds on $V_1$ and $V_2$, one gets that, with high probability,

$$\left| R_X(\hat{\beta},\beta) - \frac{\sigma^2}{n}\frac{\gamma}{\gamma-1}\operatorname{Tr}\left(\Sigma_t\Sigma_s^{-1}\right) - \frac{\sigma^2}{n}\frac{\gamma}{\gamma-1}\left(\|\Sigma_s^{-1/2}\mu_t\|_2^2 - \left(\frac{\mu_t^\top\Sigma_s^{-1}\mu_s}{\|\Sigma_s^{-1/2}\mu_s\|_2}\right)^2\right)\right| = O(n^{-c}),$$

for some constant $c > 0$. Taking the limit $n \to \infty$ on both sides yields the desired result.

## A.3 PROOF OF THEOREM 4.3

We start by removing $\alpha_2$ from the fixed point in (4.2) and replacing it by $1 - \frac{p}{n} - \alpha_1$. We rename $\alpha_1$ as $\alpha$ for convenience. Plugging this into the definition of $\mathcal{R}_u(M)$, we get

$$\mathcal{R}_u(M) = \frac{\sigma^2}{n}\operatorname{Tr}\left[\left(\alpha_1 M^\top M + \alpha_2 \operatorname{Id}_{p\times p}\right)^{-1}\right] = \frac{\sigma^2}{n}\sum_{i=1}^p \frac{1}{\lambda_i\alpha + 1 - \frac{p}{n} - \alpha},$$

where as in Theorem 4.1 we refer to $\lambda_1 \geq \cdots \geq \lambda_p$ as the eigenvalues of the matrix $MM^\top$. Furthermore, the fixed point equation (4.2) can be rewritten as follows:

$$\sum_{i=1}^p \frac{1}{\lambda_i\alpha + 1 - \frac{p}{n} - \alpha} = \frac{p + n\alpha - n_s}{1 - \frac{p}{n} - \alpha} = n\left(\frac{n - n_s}{n - p - n\alpha} - 1\right). \tag{A.24}$$

Thus, we have

$$\mathcal{R}_u(M) = \frac{\sigma^2}{n}\cdot n\left(\frac{n - n_s}{n - p - n\alpha} - 1\right) = \sigma^2\left(\frac{1 - \frac{n_s}{n}}{1 - \frac{p}{n} - \alpha} - 1\right). \tag{A.25}$$

Now, due to the RHS of (A.25), it can be seen that $\mathcal{R}_u(M)$ is an increasing function of $\alpha$. Let us denote by $\vec{\lambda} := [\lambda_1,\ldots,\lambda_p]$. Then, for fixed $n, p, n_s$ and $\vec{\lambda}$, we will refer to $\alpha(\vec{\lambda})$ as the solution to the fixed point equation (A.24). Note that following Yang et al. (2025)[Appendix B.3.2] we have that this solutions is unique and $0 < \alpha(\vec{\lambda}) < \frac{n-p}{n}$.

Consider a function $f : \mathbb{R}_{\geq 0}^p \to \mathbb{R}_{\geq 0}^p$. We call a function $f$ *good*, if and only if

$$\sum_{i=1}^p \frac{1}{f(\vec{\lambda})_i\alpha(\vec{\lambda}) + 1 - \frac{p}{n} - \alpha(\vec{\lambda})} < \sum_{i=1}^p \frac{1}{\lambda_i\alpha(\vec{\lambda}) + 1 - \frac{p}{n} - \alpha(\vec{\lambda})}. \tag{A.26}$$

We claim that if $f$ is good, then

$$\alpha(f(\vec{\lambda})) < \alpha(\vec{\lambda}). \tag{A.27}$$

**Proof of the claim.** Consider a good function $f$. Then, we have

$$\sum_{i=1}^{p} \frac{1}{f(\vec{\lambda})_i \alpha(\vec{\lambda}) + 1 - \frac{p}{n} - \alpha(\vec{\lambda})} < \sum_{i=1}^{p} \frac{1}{\lambda_i \alpha(\vec{\lambda}) + 1 - \frac{p}{n} - \alpha(\vec{\lambda})} = n \left( \frac{n - n_s}{n - p - n\alpha(\vec{\lambda})} - 1 \right).$$

Furthermore, setting $\alpha = 0$ we get

$$\sum_{i=1}^{p} \frac{1}{f(\vec{\lambda})_i \cdot 0 + 1 - \frac{p}{n} - 0} = p \frac{n}{n - p}$$

$$> n \frac{p - n_s}{n - p}$$

$$= n \left( \frac{n - n_s}{n - p - n \cdot 0} - 1 \right).$$

By continuity, there exists $\alpha_0 \in (0, \alpha(\vec{\lambda}))$ for which

$$\sum_{i=1}^{p} \frac{1}{f(\vec{\lambda})_i \alpha_0 + 1 - \frac{p}{n} - \alpha_0} = n \left( \frac{n - n_s}{n - p - n\alpha_0} - 1 \right),$$

implying $\alpha(f(\vec{\lambda})) = \alpha_0 < \alpha(\vec{\lambda})$, which concludes the proof. ♣

Next, for $i, j \in [p]$ s.t. $i < j$, we introduce a function $f_c^{i,j} : \mathbb{R}_{\geq 0}^p \to \mathbb{R}_{\geq 0}^p$ defined as

$$f_c^{i,j}(\vec{\lambda})_k = \begin{cases} \lambda_i - c & k = i, \\ \lambda_j + c & k = j, \\ \lambda_k & k \neq i, j, \end{cases}$$

where $c > 0$. We now claim that $f_c^{i,j}$ is good for any $i, j \in [p]$ and $c > 0$, such that $\lambda_i > \lambda_j + c$.

**Proof of the claim.** The claim is equivalent to

$$\frac{1}{(\lambda_i - c)\alpha(\vec{\lambda}) + 1 - \frac{p}{n} - \alpha(\vec{\lambda})} + \frac{1}{(\lambda_j + c)\alpha(\vec{\lambda}) + 1 - \frac{p}{n} - \alpha(\vec{\lambda})}$$

$$< \frac{1}{\lambda_i \alpha(\vec{\lambda}) + 1 - \frac{p}{n} - \alpha(\vec{\lambda})} + \frac{1}{\lambda_j \alpha(\vec{\lambda}) + 1 - \frac{p}{n} - \alpha(\vec{\lambda})}.$$

For simplicity, let $\delta := 1 - \frac{p}{n} - \alpha(\vec{\lambda})$ and $\alpha := \alpha(\vec{\lambda})$. Then,

$$\frac{1}{(\lambda_i - c)\alpha + \delta} + \frac{1}{(\lambda_j + c)\alpha + \delta} < \frac{1}{\lambda_i \alpha + \delta} + \frac{1}{\lambda_j \alpha + \delta}$$

$$\iff \frac{\alpha(\lambda_i + \lambda_j) + 2\delta}{(\lambda_i \alpha - c\alpha + \delta)(\lambda_j \alpha + c\alpha + \delta)} < \frac{\alpha(\lambda_i + \lambda_j) + 2\delta}{(\lambda_i \alpha + \delta)(\lambda_j \alpha + \delta)}$$

$$\iff (\lambda_i \alpha + \delta)(\lambda_j \alpha + \delta) < (\lambda_i \alpha - c\alpha + \delta)(\lambda_j \alpha + c\alpha + \delta)$$

$$\iff c\alpha(\lambda_i \alpha + \delta) - c\alpha(\lambda_j \alpha + \delta) - c^2 \alpha^2 > 0$$

$$\iff c\alpha^2(\lambda_i - \lambda_j) > c^2 \alpha^2$$

$$\iff \lambda_i > \lambda_j + c,$$

which proves the claim. ♣

This implies that, for $t \in (0, 1)$, transformations of the form

$$(\lambda_i, \lambda_j) \to (t\lambda_i + (1 - t)\lambda_j, (1 - t)\lambda_i + t\lambda_j), \tag{A.28}$$

are good.

Let us denote by $\vec{\lambda}' := [1, \ldots, 1]$, which corresponds to eigenvalues of $I_p = M'^{\top} M'$, that is $M' := I_p \in \mathcal{M}$. Pick any $\vec{\lambda}'' \neq \vec{\lambda}'$ that corresponds to some matrix $M'' \in \mathcal{M}$, so it satisfies $\lambda_1'' \geq \lambda_2'' \geq \cdots \geq \lambda_p''$, as well as $\sum_{i=1}^{p} \lambda_i'' = p$.

We recall the definition of *majorization*, as it will be used to conclude the proof. Namely, we say that $\vec{x} \in \mathbb{R}^p$ is *majorized* by $\vec{y} \in \mathbb{R}^p$ whenever for all $k \in [p]$

$$\sum_{i=1}^{k} x_i \leq \sum_{i=1}^{k} y_i,$$

and

$$\sum_{i=1}^{p} x_i = \sum_{i=1}^{p} y_i.$$

Firstly, we claim that $\vec{\lambda}'$ is majorized by $\vec{\lambda}''$. Suppose otherwise, that for some $k \in [p]$

$$\sum_{i=1}^{k} \lambda_i'' < \sum_{i=1}^{k} 1 = k,$$

implying also that $\lambda_k'' < 1$. Then, we have

$$p = \sum_{i=1}^{p} \lambda_i'' < (p-k)\lambda_k'' + k < (p-k) + k = p,$$

which is a contradiction.

Next, as $\vec{\lambda}'$ is majorized by $\vec{\lambda}''$, $\vec{\lambda}'$ can be derived from $\vec{\lambda}''$ by a finite sequence of steps of the form in (A.28) with $t \in [0, 1]$, see (Marshall et al., 1979, Chapter 4, Proposition A.1). Since both vectors $\vec{\lambda}'$ and $\vec{\lambda}''$ are non-increasing, the $t = 0$ transformation can always be omitted. Moreover, $t = 1$ is just the identity transformation, so it can also be omitted and we actually have $t \in (0, 1)$. In formulas, we have that

$$\vec{\lambda}' = f_{c_l}^{i_l, j_l}(\ldots f_{c_1}^{i_1, j_1}(\vec{\lambda}'') \ldots).$$

Since each of the functions above is good, we have that $\alpha(\vec{\lambda}') < \alpha(\vec{\lambda}'')$. As $\mathcal{R}_u(M)$ is increasing with $\alpha$, the smallest $\mathcal{R}_u(M)$ is achieved for $\vec{\lambda}' := [1, \ldots, 1]$, that is, $M_{opt} = M' = I_p$.

## A.4 PROOF OF $\mathcal{R}_u(\eta M) \leq \mathcal{R}_u(M)$

Consider the function $g_\eta : \mathbb{R}_{\geq 0}^p \to \mathbb{R}_{\geq 0}^p$ defined as $g_\eta(\vec{\lambda}) = \eta \vec{\lambda}$, for some $\eta > 1$. Note that, for all $i$,

$$\frac{1}{g_\eta(\vec{\lambda})_i \alpha + 1 - \frac{p}{n} - \alpha} = \frac{1}{\eta \lambda_i \alpha + 1 - \frac{p}{n} - \alpha} < \frac{1}{\lambda_i \alpha + 1 - \frac{p}{n} - \alpha}.$$

Thus, $g_\eta(\vec{\lambda}) = \eta \vec{\lambda}$ is *good* in the sense of (A.26). From (A.27), we obtain that $\alpha(\eta \vec{\lambda}) < \alpha(\vec{\lambda})$. This implies the desired result as $\mathcal{R}_u$ is monotonically increasing in $\alpha$ from (A.25).

## A.5 COEFFICIENT DEFINING SYSTEM OF EQUATIONS OF THEOREM 4.4

The $(a_1, a_2, a_3, a_4)$ is the unique solution, with $a_1, a_2$ positive, to the following system of equations:

$$0 = 1 - \frac{1}{\gamma} \int \frac{a_1 \lambda^s + a_2 \lambda^t}{a_1 \lambda^s + a_2 \lambda^t + 1} d\hat{H}_p(\lambda^s, \lambda^t), \quad 0 = \frac{\gamma_s}{\gamma} - \frac{1}{\gamma} \int \frac{a_1 \lambda^s}{a_1 \lambda^s + a_2 \lambda^t + 1} d\hat{H}_p(\lambda^s, \lambda^t),$$

$$\text{(A.29)}$$

$$a_1 + a_2 = -\frac{1}{\gamma} \int \frac{a_3 \lambda^s + a_4 \lambda^t}{(a_1 \lambda^s + a_2 \lambda^t + 1)^2} d\hat{H}_p(\lambda^s, \lambda^t), \quad a_1 = -\frac{1}{\gamma} \int \frac{a_3 \lambda^s + \lambda^s \lambda^t (a_3 a_2 - a_4 a_1)}{(a_1 \lambda^s + a_2 \lambda^t + 1)^2} d\hat{H}_p(\lambda^s, \lambda^t),$$

and $(b_1, b_2, b_3, b_4)$ is the unique solution, with $b_1, b_2$ positive, to the following system of equations:

$$0 = 1 - \frac{1}{\gamma} \int \frac{b_1 \lambda^s + b_2 \lambda^t}{b_1 \lambda^s + b_2 \lambda^t + 1} d\hat{H}_p(\lambda^s, \lambda^t), \quad 0 = \frac{\gamma_s}{\gamma} - \frac{1}{\gamma} \int \frac{b_1 \lambda^s}{b_1 \lambda^s + b_2 \lambda^t + 1} d\hat{H}_p(\lambda^s, \lambda^t),$$

$$\text{(A.30)}$$

$$0 = \int \frac{\lambda^s(b_3 - b_1 \lambda^t) + \lambda^t(b_4 - b_2 \lambda^t)}{(b_1 \lambda^s + b_2 \lambda^t + 1)^2} d\hat{H}_p(\lambda^s, \lambda^t), \quad 0 = \int \frac{\lambda^s(b_3 - b_1 \lambda^t) + \lambda^s \lambda^t(b_3 b_2 - b_4 b_1)}{(b_1 \lambda^s + b_2 \lambda^t + 1)^2} d\hat{H}_p(\lambda^s, \lambda^t).$$

## A.6 PROOF OF THEOREM 4.4

Recall from (3.5) that bias and variance for non-zero centered data can be expressed as

$$B_X(\hat{\beta}; \beta) = \beta^\top \Pi (\Sigma_t + \mu_t \mu_t^\top) \Pi \beta \quad \text{and} \quad V_X(\hat{\beta}; \beta) = \frac{\sigma^2}{n} \text{Tr}[\hat{\Sigma}^+ (\Sigma_t + \mu_t \mu_t^\top)],$$

where $\hat{\Sigma} = X^\top X / n$ and $\Pi = I - \hat{\Sigma}^+ \hat{\Sigma}$ (projection on the null space of $X$). To obtain the wanted result, we make a connection to zero-mean data and then use results from Song et al. (2024) to handle the zero-mean case. Unlike in the under-parametrized case, the bias term does not necessarily vanish. Thus, we start off by breaking it down into two terms

$$B_X(\hat{\beta}; \beta) = B_X^1(\hat{\beta}; \beta) + B_X^2(\hat{\beta}; \beta),$$

where $B_X^1(\hat{\beta}; \beta) = \beta^\top \Pi \Sigma_t \Pi \beta$ and $B_X^2(\hat{\beta}; \beta) = \beta^\top \Pi \mu_t \mu_t^\top \Pi \beta$. Moreover, we split the variance term as

$$V_X(\hat{\beta}; \beta) = V_X^1(\hat{\beta}; \beta) + V_X^2(\hat{\beta}; \beta),$$

with $V_X^1(\hat{\beta}; \beta) = \frac{\sigma^2}{n} \text{Tr}[\hat{\Sigma}^+ \Sigma_t]$ and $V_X^2(\hat{\beta}; \beta) = \frac{\sigma^2}{n} \text{Tr}[\hat{\Sigma}^+ \mu_t \mu_t^\top]$. We will deal with each of these terms individually.

**Bounding the term $B_X^2(\hat{\beta}, \beta)$.** Recall that $\tilde{X}_n = \frac{X}{\sqrt{n}}$. Then, similarly to (A.10), we can write the SVD of $\hat{\Sigma}$ as

$$\hat{\Sigma} = \sum_{i=1}^{k} \sigma_i^2 (\tilde{X}_n) v_i(\tilde{X}_n) v_i(\tilde{X}_n)^\top,$$

where $k \le \min(n, p) = n$ is the number of non-zero singular values of $\tilde{X}_n$. As in (A.8), we can conclude that $k = n$. Therefore, we have

$$I - \hat{\Sigma}^+ \hat{\Sigma} = I - \sum_{i=1}^{n} v_i(\tilde{X}_n) v_i(\tilde{X}_n)^\top = \sum_{i=n+1}^{p} v_i(\tilde{X}_n) v_i(\tilde{X}_n)^\top.$$

By definition, it holds that $\Pi \mu_t = (I - \hat{\Sigma}^+ \hat{\Sigma}) \mu_t$, from which it follows

$$\Pi \mu_t = \sum_{i=n+1}^{p} v_i(\tilde{X}_n) \left\langle v_i(\tilde{X}_n), \mu_t \right\rangle.$$

Due to Proposition A.2, it holds almost surely that

$$\left| \frac{\langle v_1(\tilde{X}_n), \mu_s \rangle}{\|\mu_s\|_2} \right|^2 + \left| \frac{\langle v_2(\tilde{X}_n), \mu_s \rangle}{\|\mu_s\|_2} \right|^2 \ge 1 - \frac{1}{c \cdot p},$$

from which it follows

$$\|\Pi \mu_t\|_2^2 = \sum_{i=n+1}^{p} \left| \left\langle v_i(\tilde{X}_n), \mu_t \right\rangle \right|^2 \le \frac{1}{c \cdot p} \|\mu_t\|_2^2 = c.$$

Since $\beta$ sampled independently from a sphere of constant radius and $\Pi \mu_t$ is of bounded norm, it is standard result that $|\langle \beta, \Pi \mu_t \rangle|^2$ is sub-exponential and, using Bernstein inequality, we can get that

$$B_X^2(\hat{\beta}, \beta) = \left\| \beta^\top \Pi \mu_t \right\|_2^2 = |\langle \beta, \Pi \mu_t \rangle|^2 = O\left( \frac{1}{p} \right), \tag{A.31}$$

with high probability over the sampling of $\beta$.

**Bounding the term $B_X^1(\hat{\beta}, \beta)$.** We first introduce an object coming from a bias term of a ridge regression estimator with coefficient $\lambda$:

$$B_X^1(\lambda) := \lambda^2 \beta^\top (\hat{\Sigma} + \lambda I)^{-1} \Sigma_t (\hat{\Sigma} + \lambda I)^{-1} \beta, \tag{A.32}$$

defined for any $\lambda > 0$. It is more convenient to work with $B_X^1(\lambda)$ than $B_X^1(\hat{\beta}, \beta)$ and, in addition, $B_X^1(\lambda)$ approximates well $B_X^1(\hat{\beta}, \beta)$ for small $\lambda$. We formalize the second claim as

$$\left| B_X^1(\hat{\beta}, \beta) - B_X^1(\lambda) \right| = O(\lambda) \tag{A.33}$$

proved in the same manner as (Song et al., 2024, D.82). For convenience we also carry out the proof here.

**Proof of the claim in (A.33).** Let us write the SVD $\hat{\Sigma} = UDU^\top$. Moreover, we denote by $1_{D=0}$ and $1_{D>0}$ the diagonal matrices such that

$$(1_{D=0})_{i,i} = \begin{cases} 0, & D_{i,i} \neq 0 \\ 1, & D_{i,i} = 0 \end{cases} \qquad (1_{D>0})_{i,i} = \begin{cases} 1, & D_{i,i} \neq 0 \\ 0, & D_{i,i} = 0 \end{cases}$$

Then it holds that

$$\begin{aligned} B_X^1(\hat{\beta}; \beta) &= \beta^\top (I - \hat{\Sigma}^+ \hat{\Sigma}) \Sigma_t (I - \hat{\Sigma}^+ \hat{\Sigma}) \beta \\ &= \beta^\top U 1_{D=0} U^\top \Sigma_t U 1_{D=0} U^\top \beta \\ &= \beta^\top U 1_{D=0} A 1_{D=0} U^\top \beta \\ &= \|A^{1/2} 1_{D=0} U^\top \beta\|_2^2, \end{aligned}$$

where we set $A := U^\top \Sigma_t U$. Furthermore, we have

$$\begin{aligned} B_X^1(\lambda) &= \lambda^2 \beta^\top (\hat{\Sigma} + \lambda I)^{-1} \Sigma_t (\hat{\Sigma} + \lambda I)^{-1} \beta \\ &= \lambda^2 \beta^\top U (D + \lambda I)^{-1} A (D + \lambda I)^{-1} U^\top \beta \\ &= \|A^{1/2} \lambda (D + \lambda I)^{-1} U^\top \beta\|_2^2. \end{aligned}$$

Therefore, we have

$$\begin{aligned} \left| \sqrt{B_X^1(\hat{\beta}; \beta)} - \sqrt{B_X^1(\lambda)} \right| &\leq \|A^{1/2}(1_{D=0} - \lambda(D + \lambda I)^{-1}) U^\top \beta\|_2 \\ &\leq c \|A\|_2^{1/2} \|\lambda (D + \lambda I)^{-1} 1_{D>0}\|_2 \\ &\leq c \frac{\lambda}{\sigma_n(\hat{\Sigma})} = O(\lambda), \end{aligned}$$

where the third inequality holds as $\|A\|_2 = \|\Sigma_t\|_2 = O(1)$ and the last inequality follows from Proposition A.1 in the same manner as (A.9). Notice that $B_X^1(\lambda), B_X^1(\hat{\beta}; \beta) = O(1)$, since $\|\beta\|_2, \|\Sigma_t\|_2 = O(1)$ and $\sigma_n(\hat{\Sigma}) > c$. This finally implies

$$\left| B_X^1(\hat{\beta}; \beta) - B_X^1(\lambda) \right| = O(\lambda),$$

proving the claim. ♣

The next step is to prove the claim that, for $1 > \lambda > p^{-0.49}$, it holds that

$$B_X^1(\lambda) = \lambda^2 \beta^\top (\hat{\Sigma}_0 + \lambda I)^{-1} \Sigma_t (\hat{\Sigma}_0 + \lambda I)^{-1} \beta + O\left(\frac{\lambda^{-2}}{p}\right). \tag{A.34}$$

**Proof of the claim in (A.34).** Towards this end, we have

$$\begin{aligned} \hat{\Sigma} &= \frac{1}{n}(X^\top X) \\ &= \frac{1}{n}(X^0 + 1_{n_t}\mu_t^\top + 1_{n_s}\mu_s^\top)^\top (X^0 + 1_{n_t}\mu_t^\top + 1_{n_s}\mu_s^\top) \\ &= \left( \frac{X^{0\top}X^0}{n} + \frac{X^{0\top}1_{n_t}\mu_t^\top}{n} + \frac{X^{0\top}1_{n_s}\mu_s^\top}{n} + \frac{\mu_t 1_{n_t}^\top X^0}{n} + \frac{\mu_s 1_{n_s}^\top X^0}{n} + \frac{\gamma_t}{\gamma}\mu_t\mu_t^\top + \frac{\gamma_s}{\gamma}\mu_s\mu_s^\top \right), \end{aligned}$$

where abusing notation we write $1_{n_s} = [1, \ldots, 1, 0, \ldots, 0]^\top \in \mathbb{R}^{n \times 1}$ ($n_s$ ones followed by $n_t$ zeros) and $1_{n_t} = [0, \ldots, 0, 1, \ldots, 1]^\top \in \mathbb{R}^{n \times 1}$ ($n_s$ zeros followed by $n_t$ ones).

All the terms above, except the first one, have rank 1, so we use Woodbury formula to take them out of the inverse when computing $(\hat{\Sigma} + \lambda I)^{-1}$. We consider the case $\varphi \neq 1$, as the case $\varphi = 1$ is analogous (it is in fact easier as some steps can be omitted).

We first focus on the term $(\hat{\Sigma}+\lambda I)^{-1}$ and demonstrate how to handle $\frac{X^{0\top}1_{n_t}\mu_t^\top}{n}+\frac{\mu_t 1_{n_t}^\top X^0}{n}+\frac{\gamma_t}{\gamma}\mu_t\mu_t^\top$. For this purpose, we introduce the following notation

$$
A := \hat{\Sigma} + \lambda I - \frac{X^{0\top}1_{n_t}\mu_t^\top}{n} - \frac{\mu_t 1_{n_t}^\top X^0}{n} - \frac{\gamma_t}{\gamma}\mu_t\mu_t^\top,
$$

$$
u := \frac{\mu_t}{\sqrt{n}}, \qquad v := \frac{X^{0\top}1_{n_t}}{\sqrt{n}}, \tag{A.35}
$$

$$
U := [u \ v] \in \mathbb{R}^{p\times 2}, \quad \text{and } C := \begin{bmatrix} n\frac{\gamma_t}{\gamma} & 1 \\ 1 & 0 \end{bmatrix} \in \mathbb{R}^{2\times 2}.
$$

Under this notation it holds

$$
\frac{X^{0\top}1_{n_t}\mu_t^\top}{n}+\frac{\mu_t 1_{n_t}^\top X^0}{n}+\frac{\gamma_t}{\gamma}\mu_t\mu_t^\top = UCU^\top.
$$

Then, using Woodbury formula, we have

$$
(\hat{\Sigma}+\lambda I)^{-1} = \left( A + uv^\top + vu^\top + n\frac{\gamma_t}{\gamma}uu^\top \right)^{-1}
$$

$$
= (A + UCU^\top)^{-1}
$$

$$
= A^{-1} - A^{-1}U\,(C^{-1} - U^\top A^{-1}U)^{-1}U^\top A^{-1}.
$$

We now compute the $2\times 2$ block

$$
C^{-1} - U^\top A^{-1}U = \begin{bmatrix} -u^\top A^{-1}u & 1 - u^\top A^{-1}v \\ 1 - v^\top A^{-1}u & -n\frac{\gamma_t}{\gamma} - v^\top A^{-1}v \end{bmatrix} = \begin{bmatrix} -a & 1 - b \\ 1 - b & -n\frac{\gamma_t}{\gamma} - d \end{bmatrix},
$$

where

$$
a := u^\top A^{-1}u, \qquad b := v^\top A^{-1}u = u^\top A^{-1}v, \qquad d := v^\top A^{-1}v. \tag{A.36}
$$

Hence

$$
(C^{-1} - U^\top A^{-1}U)^{-1} = \frac{1}{\Delta}\begin{bmatrix} -n\frac{\gamma_t}{\gamma} - d & b - 1 \\ b - 1 & -a \end{bmatrix}, \qquad \Delta := a\left(n\frac{\gamma_t}{\gamma} + d\right) - (1 - b)^2. \tag{A.37}
$$

Plugging back and simplifying gives the explicit formula:

$$
(\hat{\Sigma}+\lambda I)^{-1} = A^{-1} - \frac{1}{\Delta}A^{-1}\left( \left(-n\frac{\gamma_t}{\gamma} - d\right)uu^\top - (1 - b)(uv^\top + vu^\top) - a\,vv^\top \right)A^{-1},
$$

which is valid whenever $\Delta \neq 0$, i.e., whenever $C^{-1} - U^\top A^{-1}U$ is invertible.

We will now analyze the $a, b, d$ terms. First, recall that

$$
A = \frac{X^{0\top}X^0}{n} + \frac{X^{0\top}1_{n_s}\mu_s^\top}{n} + \frac{\mu_s 1_{n_s}^\top X^0}{n} + \frac{\gamma_s}{\gamma}\mu_s\mu_s^\top + \lambda I = \hat{\Sigma}_s + \lambda I,
$$

where $\hat{\Sigma}_s := \frac{(X^0 + 1_{n_s}\mu_s^\top)^\top (X^0 + 1_{n_s}\mu_s^\top)}{n}$. Thus, we have

$$
\left\| A^{-1} \right\|_2 \leq \lambda^{-1}.
$$

From this, it follows that

$$
|a| = \left| u^\top A^{-1}u \right| = \left\| \frac{\mu_t^\top}{\sqrt{n}}A^{-1}\frac{\mu_t}{\sqrt{n}} \right\|_2 \leq \left\| \frac{\mu_t}{\sqrt{n}} \right\|_2 \left\| A^{-1} \right\|_2 \left\| \frac{\mu_t}{\sqrt{n}} \right\|_2 \leq c\lambda^{-1}.
$$

Similarly, we have

$$
|b| = \left| v^\top A^{-1}u \right|
$$

$$
= \left\| \frac{\mu_t^\top}{\sqrt{n}}A^{-1}\frac{X^{0\top}1_{n_t}}{\sqrt{n}} \right\|_2
$$

$$
\leq \left\| \frac{\mu_t}{\sqrt{n}} \right\|_2 \left\| A^{-1} \right\|_2 \left\| \frac{X^{0\top}1_{n_t}}{\sqrt{n}} \right\|_2
$$

$$
\leq c\lambda^{-1}\sqrt{p},
$$

where the last inequality follows with high probability over the sampling of $X^0$, since $\frac{X^{0\top}1_{n_t}}{\sqrt{n}}$ is a vector with $p$ i.i.d entries of mean zero and $O(1)$ variance. Finally, we have

$$
\begin{aligned}
|d| &= \left| v^\top A^{-1} v \right| \\
&= \left\| \frac{1_{n_t}^\top X^0}{\sqrt{n}} A^{-1} \frac{X^{0\top}1_{n_t}}{\sqrt{n}} \right\|_2 \\
&\leq \left\| \frac{X^{0\top}1_{n_t}}{\sqrt{n}} \right\|_2 \|A^{-1}\|_2 \left\| \frac{X^{0\top}1_{n_t}}{\sqrt{n}} \right\|_2 \\
&\leq c\,\lambda^{-1} p,
\end{aligned}
$$

again with high probability.

From a slight adjustment of the second part of Proposition A.2, it holds for the top singular value

$$
\sigma_1(A) = \sigma_1(\hat{\Sigma}_s) + \lambda = \left( \sigma_1 \left( \frac{X^0 + 1_{n_s}\mu_s^\top}{\sqrt{n}} \right) \right)^2 + \lambda = \Theta(p),
$$

and for the corresponding right singular vector

$$
\left| \left\langle v_1(A), \frac{\mu_s}{\|\mu_s\|_2} \right\rangle \right| = \left| \left\langle v_1(\hat{\Sigma}_s), \frac{\mu_s}{\|\mu_s\|_2} \right\rangle \right| = \left| \left\langle v_1 \left( \frac{X^0 + 1_{n_s}\mu_s^\top}{\sqrt{n}} \right), \frac{\mu_s}{\|\mu_s\|_2} \right\rangle \right| = \sqrt{1 - O\left( \frac{1}{p} \right)}.
$$

Note that, for $\varphi < 1$, it holds that $\left| \left\langle \frac{\mu_s}{\|\mu_s\|_2}, \frac{\mu_t}{\|\mu_t\|_2} \right\rangle \right| = \varphi < 1$. Using the triangle inequality and Cauchy-Schwarz gives

$$
\left| \left\langle v_1(A), \frac{\mu_t}{\|\mu_t\|_2} \right\rangle \right| \leq \left| \left\langle \frac{\mu_s}{\|\mu_s\|_2}, \frac{\mu_t}{\|\mu_t\|_2} \right\rangle \right| + \left\| v_1(A) - \frac{\mu_s}{\|\mu_s\|_2} \right\|_2 \left\| \frac{\mu_t}{\|\mu_t\|_2} \right\|_2 \leq \varphi + O\left( \frac{1}{p} \right).
$$

Therefore, it holds that

$$
\begin{aligned}
|a| = \left| u^\top A^{-1} u \right| &= \left\| \frac{\mu_t^\top}{\sqrt{n}} A^{-1} \frac{\mu_t}{\sqrt{n}} \right\|_2 \\
&= \sum_{i=1}^{p} \frac{1}{\sigma_i(A)} \left| \left\langle v_i(A), \frac{\mu_t}{\sqrt{n}} \right\rangle \right|^2 \\
&= c \cdot \sum_{i=1}^{p} \frac{1}{\sigma_i(A)} \left| \left\langle v_i(A), \frac{\mu_t}{\|\mu_t\|_2} \right\rangle \right|^2 \\
&\geq c \sum_{i=2}^{p} \frac{1}{\sigma_i(A)} \left| \left\langle v_i(A), \frac{\mu_t}{\|\mu_t\|_2} \right\rangle \right|^2 \\
&\geq c \frac{1}{\sigma_2(A)} \sum_{i=2}^{p} \left| \left\langle v_i(A), \frac{\mu_t}{\|\mu_t\|_2} \right\rangle \right|^2 \\
&\geq c \left( 1 - \left( \varphi + O\left( \frac{1}{p} \right) \right)^2 \right) > 0,
\end{aligned}
$$

since $\sigma_2(A) = \sigma_2(\hat{\Sigma}_s) + \lambda = O(1)$ due to the second part of Proposition A.2. Note that, for $\varphi = 1$, we do not need this argument, as the $\mu_s$ terms are taken out of the inverse as well. In that case, we take $A = \left( \frac{X^{0\top}X^0}{n} + \lambda I \right)$, which immediately gives $\sigma_1(A) < c$.

We can now prove that, with high probability, $\Delta = \Omega(p)$. Using Cauchy-Schwarz, it holds that

$$
b^2 = |\langle u, v \rangle|_{A^{-1}} \leq \|u\|_{A^{-1}} \|v\|_{A^{-1}} = ad,
$$

from which it follows that

$$
\Delta = a \left( n\frac{\gamma_t}{\gamma} + d \right) - (1 - b)^2 \geq an\frac{\gamma_t}{\gamma} - 1 + 2b = \Omega(p),
$$

since $a$ is lower bounded by a constant and $|b| \leq c\lambda^{-1}\sqrt{p} \leq cp^{0.99}$.

At this point, we have all the necessary bounds and we work towards proving the claim. We first expand the bias term

$$
\begin{aligned}
B_X^1(\lambda) &= \lambda^2 \beta^\top (\hat{\Sigma} + \lambda I)^{-1} \Sigma_t (\hat{\Sigma} + \lambda I)^{-1} \beta \\
&= \lambda^2 \beta^\top (\hat{\Sigma} + \lambda I)^{-1} \Sigma_t (A + UCU^\top)^{-1} \beta \\
&= \lambda^2 \beta^\top (\hat{\Sigma} + \lambda I)^{-1} \Sigma_t \left( A^{-1} - A^{-1} U \left( C^{-1} - U^\top A^{-1} U \right)^{-1} U^\top A^{-1} \right) \beta \\
&= \lambda^2 \beta^\top (\hat{\Sigma} + \lambda I)^{-1} \Sigma_t A^{-1} \beta + S,
\end{aligned}
$$

where $S := -\lambda^2 \beta^\top (\hat{\Sigma} + \lambda I)^{-1} \Sigma_t A^{-1} U \left( C^{-1} - U^\top A^{-1} U \right)^{-1} U^\top A^{-1} \beta$.

We now prove that $S$ is small. To do so, we decompose

$$
\begin{aligned}
S &= -\lambda^2 \beta^\top (\hat{\Sigma} + \lambda I)^{-1} \Sigma_t A^{-1} U \left( C^{-1} - U^\top A^{-1} U \right)^{-1} U^\top A^{-1} \beta \\
&= \lambda^2 \beta^\top (\hat{\Sigma} + \lambda I)^{-1} \Sigma_t \frac{1}{\Delta} A^{-1} \left( \left( n\frac{\gamma_t}{\gamma} + d \right) uu^\top + (1 - b) \left( uv^\top + vu^\top \right) + a \, vv^\top \right) A^{-1} \beta \\
&= T_{u,u} + T_{u,v} + T_{v,v},
\end{aligned}
$$

where $T_{u,u}$ is the summand corresponding to $uu^\top$, $T_{u,v}$ to $uv^\top + vu^\top$, and $T_{v,v}$ to $vv^\top$. Zooming in on one of the terms, it holds that

$$
\begin{aligned}
T_{u,u} &= \lambda^2 \beta^\top (\hat{\Sigma} + \lambda I)^{-1} \Sigma_t \frac{(n\gamma_t/\gamma + d)}{\Delta} A^{-1} uu^\top A^{-1} \beta \\
&= \left\langle \beta, \lambda^2 (\hat{\Sigma} + \lambda I)^{-1} \Sigma_t \frac{(n\gamma_t/\gamma + d)}{\Delta} A^{-1} u \right\rangle \left\langle u^\top A^{-1}, \beta \right\rangle.
\end{aligned}
$$

Note that

$$
\begin{aligned}
\left\| \lambda^2 (\hat{\Sigma} + \lambda I)^{-1} \Sigma_t \frac{(n\gamma_t/\gamma + d)}{\Delta} A^{-1} u \right\|_2 &\leq \lambda^2 \left\| (\hat{\Sigma} + \lambda I)^{-1} \right\|_2 \|\Sigma_t\|_2 \frac{(n\gamma_t/\gamma + d)}{\Delta} \left\| A^{-1} \right\|_2 \|u\|_2 \\
&\leq c\lambda^{-1},
\end{aligned}
$$

and $\left\| u^\top A^{-1} \right\|_2 \leq c\lambda^{-1}$. Using this, we get that, with high probability, it holds

$$
|T_{u,u}| \leq c\frac{\lambda^{-2}}{p}.
$$

This is similar to how we obtained (A.31), since $\beta$ is sampled independently from a sphere of constant radius. With analogous passages, we have that

$$
|T_{u,v}| \leq c\frac{\lambda^{-2}}{p}, \qquad |T_{v,v}| \leq c\frac{\lambda^{-2}}{p}
$$

holds with high probability over the sampling of $\beta$. Putting all together, we get

$$
B_X^1(\lambda) = \lambda^2 \beta^\top (\hat{\Sigma} + \lambda I)^{-1} \Sigma_t A^{-1} \beta + O\left( \frac{\lambda^{-2}}{p} \right).
$$

Using the same argumentation applied now to $(\hat{\Sigma} + \lambda I)^{-1}$ in $\lambda^2 \beta^\top (\hat{\Sigma} + \lambda I)^{-1} \Sigma_t A^{-1} \beta$ gives

$$
B_X^1(\lambda) = \lambda^2 \beta^\top A^{-1} \Sigma_t A^{-1} \beta + O\left( \frac{\lambda^{-2}}{p} \right).
$$

Lastly, doing all of this again to take out the terms containing $\mu_s$ from $A$, i.e., by taking

$$
\tilde{A} := A - \frac{{X^0}^\top 1_{n_s} \mu_s^\top}{n} - \frac{\mu_s 1_{n_s}^\top X^0}{n} - \frac{\gamma_s}{\gamma} \mu_s \mu_s^\top = \hat{\Sigma}_0 + \lambda I,
$$

we get

$$
B_X^1(\lambda) = \lambda^2 \beta^\top \tilde{A}^{-1} \Sigma_t \tilde{A}^{-1} \beta + O\left( \frac{\lambda^{-2}}{p} \right),
$$

proving the claim. ♣

From (Song et al., 2024, D.82), it follows that

$$\left| \beta^\top \Pi_0 \Sigma_t \Pi_0 \beta - \lambda^2 \beta^\top \left( \hat{\Sigma}_0 + \lambda I \right)^{-1} \Sigma_t \left( \hat{\Sigma}_0 + \lambda I \right)^{-1} \beta \right| = O(\lambda), \tag{A.38}$$

where $\Pi_0 = I - \hat{\Sigma}_0^+ \hat{\Sigma}_0$. Thus, by combining (A.33), (A.34) and (A.38), we conclude that

$$\left| B_X^1(\hat{\beta}, \beta) - \beta^\top \Pi_0 \Sigma_t \Pi_0 \beta \right| = O(\lambda) + O\left( \frac{\lambda^{-2}}{p} \right) = O(p^{-1/3}), \tag{A.39}$$

where the last step is obtained by taking $p = \lambda^{-1/3}$ (this also satisfies $1 > \lambda > p^{-0.49}$, which was required to obtain (A.34)). As $B_X(\hat{\beta}, \beta) = B_X^1(\hat{\beta}, \beta) + B_X^2(\hat{\beta}, \beta)$ and $B_X^2(\hat{\beta}, \beta) = O(1/p)$ with high probability by (A.31), we conclude that

$$\left| B_X(\hat{\beta}, \beta) - \beta^\top \Pi_0 \Sigma_t \Pi_0 \beta \right| = O(p^{-1/3}) \tag{A.40}$$

holds with high probability over the sampling of $\beta$ and $X$. Plugging in the expression of $\beta^\top \Pi_0 \Sigma_t \Pi_0 \beta$ given in (Song et al., 2024, Theorem 4.1) yields, with high probability,

$$B_X(\hat{\beta}, \beta) = \int \frac{b_3 \lambda^s + (b_4 + 1)\lambda^t}{(b_1 \lambda^s + b_2 \lambda^t + 1)^2} d\hat{G}_p(\lambda^s, \lambda^t) + O(p^{-c}),$$

where $(b_1, b_2, b_3, b_4)$ is the unique solution, with $b_1, b_2$ positive, to (A.30). Taking the limit $p, n \to \infty$ gives the desired result for the bias term.

**Bounding the term $V_X^2(\hat{\beta}, \beta)$.** Notice that the term $V_X^2(\hat{\beta}, \beta)$ coincides with $T_2$ from Proposition A.3. Moreover, we can follow the proof of the bound on $T_2$ verbatim, only substituting $p$ for $n$ in appropriate places (as we are now in an over-parametrized setting) to get

$$V_X^2(\hat{\beta}, \beta) = \frac{\sigma^2}{n} \operatorname{Tr}[\hat{\Sigma}_0^+ \mu_t \mu_t^\top] = O\left( \frac{1}{p} \right). \tag{A.41}$$

**Bounding the term $V_X^1(\hat{\beta}, \beta)$.** To make a connection with zero-centered data, we will first prove that, with high probability, it holds

$$V_X^1(\hat{\beta}, \beta) = \frac{\sigma^2}{n} \operatorname{Tr}[\hat{\Sigma}^+ \Sigma_t] = \frac{1}{n} \operatorname{Tr}[\hat{\Sigma}_0^+ \Sigma_t] + O\left( \frac{1}{p^{1/7}} \right). \tag{A.42}$$

Similarly to the computation for $B_X^1(\hat{\beta}, \beta)$, we introduce an object coming from a variance term of a ridge regression estimator with coefficient $\lambda$:

$$V_X^1(\lambda) := \frac{1}{n} \operatorname{Tr}[(\hat{\Sigma} + \lambda I)^{-2} \hat{\Sigma} \Sigma_t],$$

defined for any $\lambda > 0$. It is more convenient to work with $V_X^1(\lambda)$ than $V_X^1(\hat{\beta}, \beta)$ and, in addition, $V_X^1(\lambda)$ approximates $V_X^1(\hat{\beta}, \beta)$ well for small $\lambda$. We formalize the second claim as

$$\left| V_X^1(\hat{\beta}, \beta) - V_X^1(\lambda) \right| = O(\lambda), \tag{A.43}$$

proved in the same manner as (Song et al., 2024, D.78). For convenience we also carry out the proof here.

**Proof of claim in (A.43).** Let us write the SVD $\hat{\Sigma} = UDU^\top$. Then it holds that

$$V_X^1(\hat{\beta}, \beta) = \frac{1}{n} \operatorname{Tr}(UD^+ U^\top \Sigma_t),$$

$$V_X^1(\lambda) = \frac{1}{n} \operatorname{Tr}[U(D + \lambda I)^{-2} DU^\top \Sigma_t].$$

Therefore, we have

$$
\begin{aligned}
\left| V_X^1(\hat{\beta}, \beta) - V_X^1(\lambda) \right| &= \frac{1}{n} \left| \mathrm{Tr}\left[ U^\top \Sigma_t U \left( D^+ - (D + \lambda I)^{-2} D \right) \right] \right| \\
&\leq \left\| U^\top \Sigma_t U \right\|_2 \frac{1}{n} \sum_{i=1}^n \left[ \frac{1}{\lambda_i(D)} - \frac{\lambda_i(D)}{(\lambda_i(D) + \lambda)^2} \right] \\
&\leq \frac{1}{\tau} \frac{2\lambda}{\lambda_n(D)^2} \\
&= c \cdot \frac{\lambda}{\lambda_n(\hat{\Sigma})^2} = O(\lambda).
\end{aligned}
$$

Here, we used the inequality $x^{-1} - (x + \lambda)^{-2} x \leq 2\lambda/x^2$ and the fact that $\hat{\Sigma}$ has $n$ non-zero singular values, each bounded below by a constant, which follows from (A.8). This completes the proof of the claim. ♣

Relying on the derivations in (Song et al., 2024, D.2) we have that

$$
V_X^1(\lambda) = \frac{d}{d\lambda} \left( \frac{\lambda}{n} \mathrm{Tr}\left( \Sigma_t (\hat{\Sigma} + \lambda I)^{-1} \right) \right).
$$

Let us denote by

$$
\tilde{V}_X^1(\lambda) := \frac{\lambda}{n} \mathrm{Tr}\left( \Sigma_t (\hat{\Sigma} + \lambda I)^{-1} \right).
$$

We claim that, for any $t > 0$, it holds

$$
\left| V_X^1(\lambda) - \frac{1}{t\lambda} \left( \tilde{V}_X^1(\lambda + t\lambda) - \tilde{V}_X^1(\lambda) \right) \right| = O(t\lambda^{-2}). \tag{A.44}
$$

**Proof of claim in A.44.** We begin by transforming the LHS:

$$
\begin{aligned}
\frac{1}{t\lambda}(\tilde{V}_X^1(\lambda + t\lambda) - \tilde{V}_X^1(\lambda)) &= \frac{1}{n} \mathrm{Tr}\left( \Sigma_t \frac{1}{t\lambda} \left( (\lambda + t\lambda) \left( \hat{\Sigma} + (\lambda + t\lambda)I \right)^{-1} - \lambda \left( \hat{\Sigma} + \lambda I \right)^{-1} \right) \right) \\
&= \frac{1}{n} \mathrm{Tr}\left( \Sigma_t \frac{1}{t\lambda} \left( \left( \frac{1}{\lambda + t\lambda}\hat{\Sigma} + I \right)^{-1} - \left( \frac{1}{\lambda}\hat{\Sigma} + I \right)^{-1} \right) \right) \\
&= \frac{1}{n} \mathrm{Tr}\left( \Sigma_t \frac{1}{t\lambda} \left( \left( \frac{1}{\lambda}\hat{\Sigma} + I \right)^{-1} \left( \frac{1}{\lambda}\hat{\Sigma} + I - \frac{1}{\lambda + t\lambda}\hat{\Sigma} - I \right) \left( \frac{1}{\lambda + t\lambda}\hat{\Sigma} + I \right)^{-1} \right) \right) \\
&= \frac{1}{n} \mathrm{Tr}\left( \Sigma_t \left( \hat{\Sigma} + \lambda I \right)^{-1} \hat{\Sigma} \left( \hat{\Sigma} + (\lambda + t\lambda)I \right)^{-1} \right) \\
&= \frac{1}{n} \mathrm{Tr}\left( \left( \hat{\Sigma} + (\lambda + t\lambda)I \right)^{-1} \left( \hat{\Sigma} + \lambda I \right)^{-1} \hat{\Sigma}\Sigma_t \right),
\end{aligned}
$$

where the last line follows from the cyclic property of the trace and the commutativity of $\hat{\Sigma}$, $\left( \hat{\Sigma} + \lambda I \right)^{-1}$ and $\left( \hat{\Sigma} + (\lambda + t\lambda)I \right)^{-1}$. Plugging this into the LHS of (A.44) yields

$$
\begin{aligned}
&\left| V_X^1(\lambda) - \frac{1}{t\lambda} \left( \tilde{V}_X^1(\lambda + t\lambda) - \tilde{V}_X^1(\lambda) \right) \right| \\
&= \left| \frac{1}{n} \mathrm{Tr}\left( \left( \left( \hat{\Sigma} + \lambda I \right)^{-1} - \left( \hat{\Sigma} + (\lambda + t\lambda)I \right)^{-1} \right) (\hat{\Sigma} + \lambda I)^{-1} \hat{\Sigma}\Sigma_t \right) \right| \\
&= \left| \frac{t\lambda}{n} \mathrm{Tr}\left( \left( \hat{\Sigma} + (\lambda + t\lambda)I \right)^{-1} (\hat{\Sigma} + \lambda I)^{-2} \hat{\Sigma}\Sigma_t \right) \right| \\
&\leq \left\| \Sigma_t \left( \hat{\Sigma} + (\lambda + t\lambda)I \right)^{-1} (\hat{\Sigma} + \lambda I)^{-2} \right\|_2 \frac{t\lambda}{n} \mathrm{Tr}\,\hat{\Sigma} = O(t\lambda^{-2}),
\end{aligned}
$$

where the last line follows from the bound $\frac{1}{n} \mathrm{Tr}\,\hat{\Sigma} = O(1)$, which holds due to Proposition A.2. ♣

Let us denote the zero-centered counterparts of the corresponding $V_X^1$ terms as

$$V_X^0(\hat\beta, \beta) := \frac{1}{n} \operatorname{Tr}[\hat\Sigma_0^+ \Sigma_t]$$

$$V_X^0(\lambda) := \frac{1}{n} \operatorname{Tr}[(\hat\Sigma_0 + \lambda I)^{-2} \hat\Sigma_0 \Sigma_t] = \frac{d}{d\lambda}\left(\frac{\lambda}{n} \operatorname{Tr}\left(\Sigma_t(\hat\Sigma_0 + \lambda I)^{-1}\right)\right),$$

$$\tilde V_X^0(\lambda) := \frac{\lambda}{n} \operatorname{Tr}\left(\Sigma_t(\hat\Sigma_0 + \lambda I)^{-1}\right).$$

Analogously to (A.43) and (A.44), it holds that

$$\left| V_X^0(\hat\beta, \beta) - V_X^0(\lambda) \right| = O(\lambda), \qquad \left| V_X^0(\lambda) - \frac{1}{t\lambda}\left(\tilde V_X^0(\lambda + t\lambda) - \tilde V_X^0(\lambda)\right) \right| = O(t\lambda^{-2}). \quad \text{(A.45)}$$

The next step is to prove that, for $1 > \lambda > p^{-0.49}$,

$$\tilde V_X^1(\lambda) = \tilde V_X^0(\lambda) + O\left(\frac{\lambda^{-2}}{n}\right). \quad \text{(A.46)}$$

**Proof of the claim in (A.46).** Expanding the expression, we want to prove that

$$\tilde V_X^1(\lambda) = \frac{\lambda}{n} \operatorname{Tr}\left(\Sigma_t(\hat\Sigma + \lambda I)^{-1}\right) = \frac{\lambda}{n} \operatorname{Tr}\left(\Sigma_t(\hat\Sigma_0 + \lambda I)^{-1}\right) + O\left(\frac{\lambda^{-2}}{n}\right).$$

Notice that $\tilde V_X^1(\lambda)$ crucially contains $(\hat\Sigma + \lambda I)^{-1}$ in its expression, which we have already analyzed in the context of $B_X^1(\hat\beta, \beta)$. Recalling the definitions of $A, u, v, U, C, a, b, d$, and $\Delta$ from (A.35), (A.36), and (A.37), we can then expand $\tilde V_X^1(\lambda)$ as

$$\frac{\lambda}{n} \operatorname{Tr}\left(\Sigma_t(\hat\Sigma + \lambda I)^{-1}\right) = \frac{\lambda}{n} \operatorname{Tr}\left(\Sigma_t(A + UCU^\top)^{-1}\right)$$

$$= \frac{\lambda}{n} \operatorname{Tr}\left(\Sigma_t\left(A^{-1} - A^{-1}U(C^{-1} - U^\top A^{-1}U)^{-1}U^\top A^{-1}\right)\right)$$

$$= \frac{\lambda}{n} \operatorname{Tr}\left(\Sigma_t A^{-1}\right) + \hat S,$$

where $\hat S := -\frac{\lambda}{n} \operatorname{Tr}\left(\Sigma_t A^{-1}U(C^{-1} - U^\top A^{-1}U)^{-1}U^\top A^{-1}\right)$.

We now prove that $\hat S$ is small. To do so, we decompose

$$\hat S = \frac{\lambda}{n} \operatorname{Tr}\left(\Sigma_t \frac{1}{\Delta} A^{-1}\left(\left(n\frac{\gamma_t}{\gamma} + d\right) uu^\top + (1-b)(uv^\top + vu^\top) + a\, vv^\top\right) A^{-1}\right)$$

$$= \hat T_{u,u} + \hat T_{u,v} + \hat T_{v,v},$$

where $\hat T_{u,u}$ is the summand corresponding to $uu^\top$, $\hat T_{u,v}$ to $uv^\top + vu^\top$, and $\hat T_{v,v}$ to $vv^\top$. Zooming in on one of the terms, it holds that

$$\hat T_{u,u} = \frac{\lambda}{n} \operatorname{Tr}\left(\Sigma_t \frac{1}{\Delta} A^{-1}\left(n\frac{\gamma_t}{\gamma} + d\right) uu^\top A^{-1}\right)$$

$$= \frac{\lambda}{n} \frac{n\frac{\gamma_t}{\gamma} + d}{\Delta} \operatorname{Tr}\left(\Sigma_t A^{-1} uu^\top A^{-1}\right)$$

$$= \frac{\lambda}{n} \frac{n\frac{\gamma_t}{\gamma} + d}{\Delta} u^\top A^{-1} \Sigma_t A^{-1} u.$$

Note that

$$\left\| A^{-1} \Sigma_t A^{-1} \right\|_2 \leq \frac{\lambda^{-2}}{\tau},$$

and $\|u\|_2 \leq c$. Using this, we get that, with high probability, it holds

$$|\hat T_{u,u}| \leq c\frac{\lambda^{-2}}{n}.$$

With analogous passages, we have that

$$|\hat{T}_{u,v}| \le c\frac{\lambda^{-2}}{n}, \qquad |\hat{T}_{v,v}| \le c\frac{\lambda^{-2}}{n}$$

holds with high probability over the sampling of $Z$. Putting all together, we get

$$\frac{\lambda}{n}\operatorname{Tr}\left(\Sigma_t(\hat{\Sigma}+\lambda I)^{-1}\right) = \frac{\lambda}{n}\operatorname{Tr}\left(\Sigma_t A^{-1}\right) + O\left(\frac{\lambda^{-2}}{n}\right).$$

Lastly, doing all of this again to take out the terms containing $\mu_s$ from $A$, i.e., by taking

$$\tilde{A} = A - \frac{X^{0\top}1_{n_s}\mu_s^\top}{n} - \frac{\mu_s 1_{n_s}^\top X^0}{n} - \frac{\gamma_s}{\gamma}\mu_s\mu_s^\top = \hat{\Sigma}_0 + \lambda I,$$

we get

$$\frac{\lambda}{n}\operatorname{Tr}\left(\Sigma_t(\hat{\Sigma}+\lambda I)^{-1}\right) = \frac{\lambda}{n}\operatorname{Tr}\left(\Sigma_t\left(\hat{\Sigma}_0+\lambda I\right)^{-1}\right) + O\left(\frac{\lambda^{-2}}{n}\right),$$

proving the claim. ♣

Finally, combining (A.43), (A.44), (A.45) and (A.46), for $1 > \lambda > p^{-0.49}$ and $t > 0$, we have that

$$\left|V_X^1(\hat{\beta},\beta) - V_X^0(\hat{\beta},\beta)\right| \le \left|V_X^1(\hat{\beta},\beta) - V_X^1(\lambda)\right| + \left|V_X^1(\lambda) - V_X^0(\lambda)\right| + \left|V_X^0(\hat{\beta},\beta) - V_X^0(\lambda)\right|$$

$$\le O(\lambda) + \left|\tilde{V}_X^1(\lambda) - \frac{1}{t\lambda}\left(\tilde{V}_X^1(\lambda+t\lambda) - \tilde{V}_X^1(\lambda)\right)\right|$$

$$+ \left|\tilde{V}_X^0(\lambda) - \frac{1}{t\lambda}\left(\tilde{V}_X^0(\lambda+t\lambda) - \tilde{V}_X^0(\lambda)\right)\right|$$

$$+ \left|\frac{1}{t\lambda}\left(\tilde{V}_X^1(\lambda+t\lambda) - \tilde{V}_X^1(\lambda)\right) - \frac{1}{t\lambda}\left(\tilde{V}_X^0(\lambda+t\lambda) - \tilde{V}_X^0(\lambda)\right)\right|$$

$$\le O(\lambda) + O\left(\frac{t}{\lambda^2}\right) + \frac{1}{t\lambda}\left|\tilde{V}_X^1(\lambda+t\lambda) - \tilde{V}_X^0(\lambda+t\lambda)\right| + \frac{1}{t\lambda}\left|\tilde{V}_X^1(\lambda) - \tilde{V}_X^0(\lambda)\right|$$

$$= O(\lambda) + O(t\lambda^{-2}) + O\left(\frac{t^{-1}\lambda^{-3}}{n}\right).$$

Taking $t = \lambda^3$ and $\lambda = n^{-1/7}$, we get $\left|V_X^1(\hat{\beta},\beta) - V_X^0(\hat{\beta},\beta)\right| = O(n^{-1/7})$, proving the claim from (A.42). As $V_X(\hat{\beta};\beta) = V_X^1(\hat{\beta};\beta) + V_X^2(\hat{\beta};\beta)$, and $V_X^2(\hat{\beta},\beta) = O(1/p)$ by (A.41) we conclude that

$$V_X(\hat{\beta};\beta) = \frac{\sigma^2}{n}\operatorname{Tr}[\hat{\Sigma}^+(\Sigma_t + \mu_t\mu_t^\top)] = \frac{\sigma^2}{n}\operatorname{Tr}[\hat{\Sigma}_0^+\Sigma_t] + O\left(p^{-1/7}\right).$$

Plugging in the expression of $\frac{\sigma^2}{n}\operatorname{Tr}[\hat{\Sigma}_0^+\Sigma_t]$ given in (Song et al., 2024, Theorem 4.1) yields, with high probability,

$$V_X(\hat{\beta};\beta) = -\frac{\sigma^2}{\gamma}\int\frac{\lambda^t(a_3\lambda^s + a_4\lambda^t)}{(a_1\lambda^s + a_2\lambda^t + 1)^2}d\hat{H}_p(\lambda^s,\lambda^t) + O(p^{-c}),$$

where $(a_1, a_2, a_3, a_4)$ is the unique solution, with $a_1, a_2$ positive, to (A.29). Taking the limit $p, n \to \infty$ gives the desired result for the variance term and concludes the proof.

## A.7 Proof of Theorem 4.5

For $\Sigma_t = I_p$ and $\Sigma_s \in \mathbb{R}_{>0}^{p\times p}$, it holds that

$$\mathcal{R}_o(\Sigma_s, I_p, \beta) = \mathcal{V}(\Sigma_s, I_p) + \mathcal{B}(\Sigma_s, I_p, \beta).$$

We analyze each of the two terms separately.

**Calculating** $\mathcal{B}(\Sigma_s, I_p, \beta)$**.** Note that $\Sigma_t = I_p$ implies $\lambda_i^t = 1$ in all the equations in (A.30). Plugging this in, one gets that the third and fourth equation in (A.30) are satisfied for $b_4 = b_2$ and $b_3 = b_1$. From the uniqueness of a solution $(b_1, b_2, b_3, b_4)$ to the whole system of equations in $(A.30)$, and the fact that $b_3$ and $b_4$ only show up in the mentioned third and fourth equation, we get that it must hold $b_4 = b_2$ and $b_3 = b_1$. Plugging this into the bias term we get that

$$\mathcal{B}(\Sigma_s, I_p, \beta) = \int \frac{b_3 \lambda^s + (b_4 + 1)\lambda^t}{(b_1 \lambda^s + b_2 \lambda^t + 1)^2} d\hat{G}_p(\lambda^s, \lambda^t)$$

$$= \int \frac{b_1 \lambda^s + b_2 + 1}{(b_1 \lambda^s + b_2 + 1)^2} d\hat{G}_p(\lambda^s, \lambda^t)$$

$$= \sum_{i=1}^{p} \frac{\langle \beta, u_i \rangle^2}{b_1 \lambda_i^s + b_2 + 1},$$

noting that $u_i \in \mathbb{R}^p$ is the eigenvector of the matrix $\Sigma_s$ corresponding to the eigenvalue $\lambda_i^s$.

Recall that we have assumed in the setup of Section 4.2 that $\beta$ is sampled from a sphere of constant radius, which we will denote by $rS^{p-1}$, i.e., $r = \|\beta\|_2$. We now prove concentration of $\mathcal{B}(\Sigma_s, I_p, \beta)$ over this sampling of $\beta$. Towards this end, we introduce a matrix $A \in \mathbb{R}^{p \times p}$ such that

$$\mathcal{B}(\Sigma_s, I_p, \beta) = \beta^\top A \beta, \qquad A := \sum_{i=1}^{p} \frac{1}{b_1 \lambda_i^s + b_2 + 1} u_i u_i^\top.$$

Notice that first equation of (A.30) yields

$$\frac{1}{\gamma p} \sum_{i=1}^{p} \frac{b_1 \lambda_i^s + b_2}{b_1 \lambda_i^s + b_2 + 1} = 1,$$

which gives

$$\mathrm{Tr}(A) = \sum_{i=1}^{p} \frac{1}{b_1 \lambda_i^s + b_2 + 1} = p - n.$$

Since both $b_1$ and $b_2$ are positive, as stated in Theorem 4.4, it holds

$$\|A\|_2 = \lambda_1(A) = \frac{1}{b_1 \lambda_p^s + b_2 + 1} \leq 1.$$

Note that

$$\mathbb{E}_{\beta \sim rS^{p-1}} \beta^\top A \beta = \mathbb{E} \sum_{i=1}^{p} \frac{\langle \beta, u_i \rangle^2}{b_1 \lambda_i^s + b_2 + 1}$$

$$= \sum_{i=1}^{p} \frac{1}{b_1 \lambda_i^s + b_2 + 1} \mathbb{E} \langle \beta, u_i \rangle^2$$

$$= \frac{1}{p} \sum_{i=1}^{p} \frac{1}{b_1 \lambda_i^s + b_2 + 1} r^2$$

$$= \frac{p - n}{p} r^2. \tag{A.47}$$

Furthermore, the function $\beta \to \beta^\top A \beta$ is Lipschitz over the sphere. Namely, for two vectors $\beta_1, \beta_2 \in rS^{p-1}$, it holds that

$$|\beta_1^\top A \beta_1 - \beta_2^\top A \beta_2| \leq |\beta_1^\top A(\beta_1 - \beta_2)| + |\beta_2^\top A(\beta_1 - \beta_2)| \leq 2r \|A\|_2 \|\beta_1 - \beta_2\|_2 \leq 2r \|\beta_1 - \beta_2\|_2.$$

Then, due to the concentration of Lipschitz functions over the sphere (Vershynin, 2018, Theorem 5.1.4), we get that, with overwhelming probability,

$$\left| \beta^\top A \beta - \mathbb{E} \beta^\top A \beta \right| = O(n^{-c_1}),$$

for any constant $c_1 < 1/2$. Plugging (A.47) gives

$$\mathcal{B}(\Sigma_s, I_p, \beta) = \beta^\top A \beta = \frac{p - n}{p} r^2 + O(n^{-c_1}),$$

with overwhelming probability. We can readily calculate the bias term for $\Sigma_s = I_p$:

$$\mathcal{B}(I_p, I_p, \beta) = \sum_{i=1}^{p} \frac{\langle \beta, u_i \rangle^2}{b_1 + b_2 + 1} = \frac{p - n}{p} r^2.$$

Thus, for any $\Sigma_s \in \mathbb{R}_{\succ 0}^{p \times p}$, we have

$$\mathcal{B}(I_p, I_p, \beta) \le \mathcal{B}(\Sigma_s, I_p, \beta) + O(n^{-c_1}), \tag{A.48}$$

with overwhelming probability.

**Calculating $\mathcal{V}(\Sigma_s, I_p)$.** Note that

$$
\begin{aligned}
\mathcal{V}(\Sigma_s, I_p) &= -\sigma^2 \frac{1}{\gamma} \int \frac{\lambda^t (a_3 \lambda^s + a_4 \lambda^t)}{(a_1 \lambda^s + a_2 \lambda^t + 1)^2} d\hat{H}_p(\lambda^s, \lambda^t) \\
&= -\sigma^2 \frac{1}{\gamma} \int \frac{a_3 \lambda^s + a_4}{(a_1 \lambda^s + a_2 + 1)^2} d\hat{H}_p(\lambda^s, \lambda^t) \\
&= \sigma^2 (a_1 + a_2),
\end{aligned}
\tag{A.49}
$$

where the last equality follows from the third equation in (A.29) and the fact that $\lambda_i^t = 1$ for all $i \in [p]$. Moreover, subtracting the second from the first equation in (A.29) yields

$$0 = 1 - \frac{\gamma_s}{\gamma} - \frac{1}{\gamma} \frac{1}{p} \sum_{i=1}^{p} \frac{a_2}{a_1 \lambda_i^s + a_2 + 1}. \tag{A.50}$$

Analyzing just the first equation in (A.29), we get

$$\frac{1}{\gamma p} \left( p - \sum_{i=1}^{p} \frac{1}{a_1 \lambda_i^s + a_2 + 1} \right) = \frac{1}{\gamma p} \sum_{i=1}^{p} \frac{a_1 \lambda_i^s + a_2}{a_1 \lambda_i^s + a_2 + 1} = 1,$$

which gives

$$\sum_{i=1}^{p} \frac{1}{a_1 \lambda_i^s + a_2 + 1} = p - n.$$

Plugging this into (A.50) we get that $a_2 = \frac{\gamma_t}{1 - \gamma}$. Therefore, $a_1$ is the unique solution to

$$\sum_{i=1}^{p} \frac{1}{a_1 \lambda_i^s + c_2} = p - n, \tag{A.51}$$

for $c_2 = \frac{\gamma_t}{1 - \gamma} + 1 > 0$. From (A.49), we have that $\mathcal{V}(\Sigma_s, I_p)$ only depends on $\Sigma_s$ through $a_1$, with which it monotonically increases. To conclude this section, we will apply the majorization argument from the proof of Theorem 4.3 with a slight modification. Almost all parts of the argument are analogous, and we restate them mainly for convenience.

Let us denote by $\vec{\lambda}^s := [\lambda_1^s, \ldots, \lambda_p^s]$. Then, for fixed $n, p$ and $\vec{\lambda}^s$, we will refer to $a_1(\vec{\lambda}^s)$ as the positive solution to (A.51). Note that from Theorem 4.4 we have that this solution is unique. Consider a function $f : \mathbb{R}_{\ge 0}^p \to \mathbb{R}_{\ge 0}^p$. We call a function $f$ *good*, if and only if

$$\sum_{i=1}^{p} \frac{1}{a_1(\vec{\lambda}^s) f(\vec{\lambda}^s)_i + c_2} < \sum_{i=1}^{p} \frac{1}{a_1(\vec{\lambda}^s) \lambda_i^s + c_2}. \tag{A.52}$$

We claim that, if $f$ is good, then

$$a_1(f(\vec{\lambda}^s)) < a_1(\vec{\lambda}^s). \tag{A.53}$$

**Proof of the claim.** Consider a good function $f$. Then, we have

$$\sum_{i=1}^{p} \frac{1}{a_1(\vec{\lambda}^s) f(\vec{\lambda}^s)_i + c_2} < \sum_{i=1}^{p} \frac{1}{a_1(\vec{\lambda}^s) \lambda_i^s + c_2} = p - n.$$

Furthermore, setting $a_1 = 0$ we get

$$\sum_{i=1}^{p} \frac{1}{0 \cdot f(\vec{\lambda}^s)_i + c_2} = p \, \frac{1}{\frac{\gamma_t}{1-\gamma} + 1}$$

$$= p \, \frac{p-n}{p-n_s}$$

$$> p - n.$$

By continuity, there exists $a_1' \in (0, a_1(\vec{\lambda}^s))$ for which

$$\sum_{i=1}^{p} \frac{1}{a_1' f(\vec{\lambda}^s)_i + c_2} = n - p,$$

implying $a_1(f(\vec{\lambda}^s)) = a_1' < a_1(\vec{\lambda}^s)$, which concludes the proof. ♣

Next, for $i, j \in [p]$ s.t. $i < j$, we introduce a function $f_c^{i,j} : \mathbb{R}_{\geq 0}^p \to \mathbb{R}_{\geq 0}^p$ defined as

$$f_c^{i,j}(\vec{\lambda})_k = \begin{cases} \lambda_i^s - c & k = i, \\ \lambda_j^s + c & k = j, \\ \lambda_k^s & k \neq i, j, \end{cases}$$

where $c > 0$ is a constant. We now claim that $f_c^{i,j}$ is good for any $i, j \in [p]$ and $c > 0$, such that $\lambda_i^s > \lambda_j^s + c$.

**Proof of the claim.** The claim is equivalent to

$$\frac{1}{a_1(\vec{\lambda}^s)(\lambda_i^s - c) + c_2} + \frac{1}{a_1(\vec{\lambda}^s)(\lambda_j^s + c) + c_2} < \frac{1}{a_1(\vec{\lambda}^s)\lambda_i^s + c_2} + \frac{1}{a_1(\vec{\lambda}^s)\lambda_j^s + c_2}.$$

For simplicity, let us denote $a := a_1(\vec{\lambda}^s)$. Then,

$$\frac{1}{a(\lambda_i^s - c) + c_2} + \frac{1}{a(\lambda_j^s + c) + c_2} < \frac{1}{a\lambda_i^s + c_2} + \frac{1}{a\lambda_j^s + c_2}$$

$$\iff \frac{a(\lambda_i^s + \lambda_j^s) + 2c_2}{(\lambda_i^s a - ca + c_2)(\lambda_j^s a + ca + c_2)} < \frac{a(\lambda_i^s + \lambda_j^s) + 2c_2}{(\lambda_i^s a + c_2)(\lambda_j^s a + c_2)}$$

$$\iff (\lambda_i^s a + c_2)(\lambda_j^s a + c_2) < (\lambda_i^s a - ca + c_2)(\lambda_j^s a + ca + c_2)$$

$$\iff ca(\lambda_i^s a + c_2) - ca(\lambda_j^s a + c_2) - c^2 a^2 > 0$$

$$\iff ca^2(\lambda_i^s - \lambda_j^s) > c^2 a^2$$

$$\iff \lambda_i^s > \lambda_j^s + c,$$

which proves the claim. ♣

This implies that, for $t \in (0, 1)$, transformations of the form

$$(\lambda_i^s, \lambda_j^s) \to (t\lambda_i^s + (1-t)\lambda_j^s, (1-t)\lambda_i^s + t\lambda_j^s) \tag{A.54}$$

are good. Let us denote by $\vec{\lambda}^{id} := [1, \ldots, 1]$, which corresponds to the matrix $I_p$. Pick any $\vec{\lambda}^s \neq \vec{\lambda}^{id}$ that corresponds to some matrix $\Sigma_s \in \mathcal{S}$, so it satisfies $\lambda_1^s \geq \lambda_2^s \geq \cdots \geq \lambda_p^s$, as well as $\sum_{i=1}^{p} \lambda_i^s = p$.

Firstly, we claim that $\vec{\lambda}^{id}$ is majorized by $\vec{\lambda}^s$. Suppose otherwise, that for some $k \in [p]$

$$\sum_{i=1}^{k} \lambda_i^s < \sum_{i=1}^{k} 1 = k,$$

implying also that $\lambda_k^s < 1$. Then, we have

$$p = \sum_{i=1}^{p} \lambda_i^s < (p-k)\lambda_k^s + k < (p-k)1 + k = p,$$

which is a contradiction.

Next, as $\vec{\lambda}^{id}$ is majorized by $\vec{\lambda}^s$, $\vec{\lambda}^{id}$ can be derived from $\vec{\lambda}^s$ by a finite sequence of steps of the form in (A.54) with $t \in [0, 1]$, see (Marshall et al., 1979, Chapter 4, Proposition A.1). Since both vectors $\vec{\lambda}^{id}$ and $\vec{\lambda}^s$ are non-increasing, the $t = 0$ transformation can always be omitted. Moreover, $t = 1$ is just the identity transformation, so it can also be omitted and we actually have $t \in (0, 1)$. In formulas, we have that

$$\vec{\lambda}^{id} = f_{c_l}^{i_l, j_l}(\dots f_{c_1}^{i_1, j_1}(\vec{\lambda}^s) \dots).$$

Since each of the functions above is good, we have that $a_1(\vec{\lambda}^{id}) < a_1(\vec{\lambda}^s)$. As $\mathcal{V}(\Sigma_s, I_p)$ is increasing with $a_1$, this directly implies that, for any $\Sigma_s \in \mathbb{R}_{\succ 0}^{p \times p}$,

$$\mathcal{V}(I_p, I_p) \leq \mathcal{V}(\Sigma_s, I_p).$$

Combining this with (A.48), we get

$$\mathcal{R}_o(I_p, I_p, \beta) \leq \mathcal{R}_o(\Sigma_s, I_p, \beta) + o(1),$$

with overwhelming probability, which concludes the proof.

### A.8 WEIGHTED OBJECTIVES UNDER DIFFERENT LABEL NOISE

Let us suppose we work in the setting where the label noise $\varepsilon_{(i)}$ differs between real and synthetic data. A natural way to account for this in training is to assign weights exactly inversely proportional to the corresponding noise levels. Thus, let $w_1$ be the weight assigned to the real data, and $w_2$ the weight assigned to synthetic data. This leads to three possible regimes:

- $w_1 \ll w_2$. Here, synthetic data effectively determines the estimator. In this context, we would be in the scenario of Proposition 4.2. As discussed there, both the mean and covariance discrepancy play a role, and thus the resulting optimality condition differs from the mixed-data case.

- $w_1 \gg w_2$. In this regime, the estimator is driven almost entirely by the real data, reducing to the classical setting studied by Hastie et al. (2022). Consequently, the choice of synthetic data has negligible impact.

- $w_1 \sim w_2$. This is arguably the most interesting case. The current theory applies directly, since weighting real and synthetic observations is equivalent to scaling $(\Sigma_t, \mu_t)$ by $w_1$ and $(\Sigma_s, \mu_s)$ by $w_2$. Then, by adjusting Theorems 4.1, 4.3, 4.4 and 4.5, one can prove that the optimal choice becomes $\Sigma_s \sim \frac{w_1}{w_2} \Sigma_t$.

Thus, the qualitative conclusion remains unchanged: matching the synthetic covariance to that of the real data is optimal, where the weighting reflects the appropriate scaling depending on the trust placed in the labels.

## B MODEL SHIFT

In this section we give a precise estimate of the excess risk of the min-norm interpolator using both training and augmenting synthetic data under model shift, assuming no covariance shift is present. Interestingly, the difference in means once again does not affect the characterization. Furthermore, depending on the intensity of model shift, synthetic data may have limited or even negative effect. By characterizing this, the resulting formula suggests a simple training heuristic for synthetic data training in the presence of model shift. We focus on the under-parameterized regime, which offers a technically cleaner setting for analysis. Extending these results to the over-parameterized regime is more involved, though we anticipate that existing techniques should make this feasible. We leave this extension for future work.

### B.1 PRELIMINARIES

**Data model.** We consider the setup of Section 3, with the addition of possible model shift. This is modeled as

$$y_{(i)} = X_{(i)}\beta_{(i)} + \varepsilon_{(i)}, \qquad (i) \in \{t, s\}, \tag{B.1}$$

where $\beta_{(i)} \in \mathbb{R}^p$, and it is no longer necessary that $\beta_s = \beta_t$. Note that this is a common modeling in the transfer learning literature, as done in prior work (Song et al., 2024; Yang et al., 2025). This setting allows us to explore the impact of model shift on generalization error, as the true parameter is no longer shared between training and synthetic data. Lastly, to emphasize difference in means we will assume that the synthetic data is normalized, i.e., $\mu_s = 0$, while keeping $\|\mu_t\|_2 = r_t\sqrt{p}$.

**Risk and estimator.** As in Section 3, we test estimators on data sampled from the same distribution as the training data $(X_t, y_t)$. Then, under the adjusted data model, the formula for the risk can be broken down into a bias and variance term as in (3.3). Namely, it holds

$$R_X(\hat{\beta}; \beta_t, \beta_s) = \|\mathbb{E}[\hat{\beta} \mid X] - \beta_t\|^2_{\Sigma_t + \mu_t\mu_t^\top} + \mathrm{Tr}[\mathrm{Cov}(\hat{\beta} \mid X)(\Sigma_t + \mu_t\mu_t^\top)]$$
$$:= B_X(\hat{\beta}; \beta_t, \beta_s) + V_X(\hat{\beta}; \beta_t, \beta_s). \tag{B.2}$$

We are interested in the performance of the minimum norm interpolator. Its closed-form solution is unaffected by adjustment to the data model, i.e. it remains the same as in (3.4), i.e.,

$$\hat{\beta} := \arg\min \left\{ \|b\|_2 : b \text{ minimizes } \|y - Xb\|_2^2 \right\} = (X^\top X)^+ X^\top y. \tag{B.3}$$

Substituting (B.3) into the excess risk decomposition (B.2) yields closed-form expressions for bias

$$B_X(\hat{\beta}; \beta_t, \beta_s) = \beta_t^\top \Pi(\Sigma_t + \mu_t\mu_t^\top)\Pi\beta_t - 2\beta_t^\top \Pi(\Sigma_t + \mu_t\mu_t^\top)\hat{\Sigma}^+ \left( \frac{X_s^\top X_s}{n} \right) (\beta_s - \beta_t)$$
$$+ (\beta_s - \beta_t)^\top \left( \frac{X_s^\top X_s}{n} \right) \hat{\Sigma}^+ (\Sigma_t + \mu_t\mu_t^\top)\hat{\Sigma}^+ \left( \frac{X_s^\top X_s}{n} \right) (\beta_s - \beta_t) \tag{B.4}$$

and variance

$$V_X(\hat{\beta}; \beta_t, \beta_s) = \frac{\sigma^2}{n} \mathrm{Tr}[\hat{\Sigma}^+ (\Sigma_t + \mu_t\mu_t^\top)], \tag{B.5}$$

where $\hat{\Sigma} = X^\top X/n$ and $\Pi = I - \hat{\Sigma}^+\hat{\Sigma}$ (projection on the null space of $X$).

## B.2 THEORETICAL RESULTS

As mentioned at the start of Appendix B, we aim to characterize the excess risk of the min-norm interpolator under model shift, when no covariance shift is present.

Let us assume that $1 + \tau \leq \gamma \leq 1/\tau$, implying that $n > p$, which makes the setting under-parameterized. Thus, $\hat{\Sigma} = X^\top X/n$ is full rank almost surely, which implies that $\Pi = I - \hat{\Sigma}^+\hat{\Sigma} = I - \hat{\Sigma}^{-1}\hat{\Sigma} = 0$. From (B.4), it follows that

$$B_X(\hat{\beta}; \beta_t, \beta_s) = (\beta_s - \beta_t)^\top \left( \frac{X_s^\top X_s}{n} \right) \hat{\Sigma}^{-1} (\Sigma_t + \mu_t\mu_t^\top)\hat{\Sigma}^{-1} \left( \frac{X_s^\top X_s}{n} \right) (\beta_s - \beta_t). \tag{B.6}$$

We additionally constrain the number of samples as $1 + \tau \leq \gamma_t, \gamma_s \leq 1/\tau$ and $0 < \gamma_s/\gamma_t \leq 1/\tau$. Similarly to Section 4.2, $\beta_s$ and $\beta_t$ are sampled from a sphere of constant radius, independently from $X, \varepsilon_t, \varepsilon_s$. We will work in the setting where population covariances match, i.e. $\Sigma_s = \Sigma_t$, highlighting the influence of model shift.

The following result provides a precise asymptotic characterization of the excess risk and, in doing so, it extends results by Yang et al. (2025) to non-zero centered data.

**Theorem B.1.** *Let $\Sigma_t = \Sigma_s$. Then under the assumptions from Section 3 and the start of this section it holds*

$$\lim_{n \to \infty} \left| R_X(\hat{\beta}; \beta_t, \beta_s) - \left\| \Sigma_s^{1/2}(\beta_s - \beta_t) \right\|_2^2 \cdot s_1 - \sigma^2 \cdot s_2 \right| = 0,$$

*where*

$$s_1 = \frac{n_s^2(n-p) + pn_sn_t}{n^2(n-p)}, \qquad s_2 = \frac{p}{n-p}.$$

This theorem has an interesting implication. Unlike in the under-parameterized setting without model shift, the inclusion of synthetic data can degrade performance in the presence of model shift.

As discussed in the random-effects framework (Section 4.2 of (Yang et al., 2025)), different shift intensities can cause different effects from training on synthetic data. This observation suggests that when covariances are matched, a practical training strategy would be to gradually downsample the synthetic dataset until the performance begins to decrease. We refer the reader to the above-cited work for a more detailed discussion of these phenomena.

We now proceed to prove the main theorem of this appendix.

*Proof of Theorem B.1.* Firstly, note that under the assumption $\mu_s = 0$ it holds that $X_s = X_s^0$, i.e., synthetic data is centered. Next, from (B.2), we see that the risk can be broken down into the bias and variance term. We will handle each of them separately. Let us start by splitting the bias term in (B.6) into two components, i.e., $B_X(\hat{\beta}; \beta_t, \beta_s) = B_1 + B_2$, where

$$B_1 := (\beta_s - \beta_t)^\top \left( \frac{X_s^{0\top} X_s^0}{n} \right) \hat{\Sigma}^{-1} \Sigma_t \hat{\Sigma}^{-1} \left( \frac{X_s^{0\top} X_s^0}{n} \right) (\beta_s - \beta_t),$$

$$B_2 := (\beta_s - \beta_t)^\top \left( \frac{X_s^{0\top} X_s^0}{n} \right) \hat{\Sigma}^{-1} \mu_t \mu_t^\top \hat{\Sigma}^{-1} \left( \frac{X_s^{0\top} X_s^0}{n} \right) (\beta_s - \beta_t).$$

**Bounding the term $B_1$.** We will first prove that

$$B_1 = (\beta_s - \beta_t)^\top \left( \frac{X_s^{0\top} X_s^0}{n} \right) \hat{\Sigma}_0^{-1} \Sigma_t \hat{\Sigma}_0^{-1} \left( \frac{X_s^{0\top} X_s^0}{n} \right) (\beta_s - \beta_t) + O\left( \frac{1}{p} \right). \qquad \text{(B.7)}$$

The argument mirrors the proof of (A.34), and we include it here to point out the necessary modifications.

**Proof of the claim in (B.7).** It holds that

$$\hat{\Sigma} = \frac{1}{n}(X^\top X)$$

$$= \frac{1}{n}(X^0 + 1_{n_t} \mu_t^\top)^\top (X^0 + 1_{n_t} \mu_t^\top)$$

$$= \left( \frac{X^{0\top} X^0}{n} + \frac{X^{0\top} 1_{n_t} \mu_t^\top}{n} + \frac{\mu_t 1_{n_t}^\top X^0}{n} + \frac{\gamma_t}{\gamma} \mu_t \mu_t^\top \right),$$

where abusing notation we write $1_{n_t} = [0, \ldots, 0, 1, \ldots, 1]^\top \in \mathbb{R}^{n \times 1}$ ($n_s$ zeros followed by $n_t$ ones). All the terms above, except the first one, have rank 1, so we use Woodbury formula to take them out of the inverse when computing $\hat{\Sigma}^{-1}$. We introduce the following notation

$$A := \frac{X^{0\top} X^0}{n} = \hat{\Sigma}_0,$$

$$u := \frac{\mu_t}{\sqrt{n}}, \qquad v := \frac{X^{0\top} 1_{n_t}}{\sqrt{n}}, \qquad \text{(B.8)}$$

$$U := [u \ v] \in \mathbb{R}^{p \times 2}, \quad \text{and } C := \begin{bmatrix} n\frac{\gamma_t}{\gamma} & 1 \\ 1 & 0 \end{bmatrix} \in \mathbb{R}^{2 \times 2}.$$

Under this notation, it holds that

$$\frac{X^{0\top} 1_{n_t} \mu_t^\top}{n} + \frac{\mu_t 1_{n_t}^\top X^0}{n} + \frac{\gamma_t}{\gamma} \mu_t \mu_t^\top = UCU^\top.$$

Then, using Woodbury formula, we have

$$\hat{\Sigma}^{-1} = \left( A + uv^\top + vu^\top + n\frac{\gamma_t}{\gamma} uu^\top \right)^{-1}$$

$$= (A + UCU^\top)^{-1}$$

$$= A^{-1} - A^{-1} U (C^{-1} - U^\top A^{-1} U)^{-1} U^\top A^{-1}.$$

We now compute the $2 \times 2$ block

$$C^{-1} - U^\top A^{-1} U = \begin{bmatrix} -u^\top A^{-1} u & 1 - u^\top A^{-1} v \\ 1 - v^\top A^{-1} u & -n\frac{\gamma_t}{\gamma} - v^\top A^{-1} v \end{bmatrix} = \begin{bmatrix} -a & 1-b \\ 1-b & -n\frac{\gamma_t}{\gamma} - d \end{bmatrix},$$

where

$$a := u^\top A^{-1} u, \qquad b := v^\top A^{-1} u = u^\top A^{-1} v, \qquad d := v^\top A^{-1} v. \tag{B.9}$$

Hence

$$(C^{-1} - U^\top A^{-1} U)^{-1} = \frac{1}{\Delta} \begin{bmatrix} -n\frac{\gamma_t}{\gamma} - d & b-1 \\ b-1 & -a \end{bmatrix}, \qquad \Delta := a\left(n\frac{\gamma_t}{\gamma} + d\right) - (1-b)^2. \tag{B.10}$$

Plugging back and simplifying gives the explicit formula:

$$\hat\Sigma^{-1} = A^{-1} - \frac{1}{\Delta} A^{-1} \left( \left( -n\frac{\gamma_t}{\gamma} - d \right) uu^\top - (1-b)\left( uv^\top + vu^\top \right) - a\, vv^\top \right) A^{-1},$$

which is valid whenever $\Delta \neq 0$, i.e., whenever $C^{-1} - U^\top A^{-1} U$ is invertible. We will now analyze the $a, b, d$ terms. First, note that there exist constants $c_1, c_2 > 0$ such that

$$0 < c_1 \leq \lambda_p(A) \leq \lambda_1(A) \leq c_2. \tag{B.11}$$

By the Bai–Yin theorem (Bai & Silverstein, 2010, Theorem 5.11), this directly implies that $\left\| A^{-1} \right\|_2 \leq c$. From this, it follows that

$$|a| = \left| u^\top A^{-1} u \right| = \left\| \frac{\mu_t^\top}{\sqrt{n}} A^{-1} \frac{\mu_t}{\sqrt{n}} \right\|_2 \leq \left\| \frac{\mu_t}{\sqrt{n}} \right\|_2 \left\| A^{-1} \right\|_2 \left\| \frac{\mu_t}{\sqrt{n}} \right\|_2 \leq c.$$

Similarly, we have

$$\begin{aligned} |b| &= \left| v^\top A^{-1} u \right| \\ &= \left\| \frac{\mu_t^\top}{\sqrt{n}} A^{-1} \frac{X^{0\top} \mathbf{1}_{n_t}}{\sqrt{n}} \right\|_2 \\ &\leq \left\| \frac{\mu_t}{\sqrt{n}} \right\|_2 \left\| A^{-1} \right\|_2 \left\| \frac{X^{0\top} \mathbf{1}_{n_t}}{\sqrt{n}} \right\|_2 \\ &\leq c\sqrt{p}, \end{aligned}$$

where the last inequality follows with high probability over the sampling of $X^0$, since $\frac{X^{0\top} \mathbf{1}_{n_t}}{\sqrt{n}}$ is a vector with $p$ i.i.d entries of mean zero and $O(1)$ variance. Finally, we have

$$\begin{aligned} |d| &= \left| v^\top A^{-1} v \right| \\ &= \left\| \frac{\mathbf{1}_{n_t}^\top X^0}{\sqrt{n}} A^{-1} \frac{X^{0\top} \mathbf{1}_{n_t}}{\sqrt{n}} \right\|_2 \\ &\leq \left\| \frac{X^{0\top} \mathbf{1}_{n_t}}{\sqrt{n}} \right\|_2 \left\| A^{-1} \right\|_2 \left\| \frac{X^{0\top} \mathbf{1}_{n_t}}{\sqrt{n}} \right\|_2 \\ &\leq c\, p, \end{aligned}$$

again with high probability. Therefore, it holds that

$$|a| = \left| u^\top A^{-1} u \right| = \left\| \frac{\mu_t^\top}{\sqrt{n}} A^{-1} \frac{\mu_t}{\sqrt{n}} \right\|_2 \geq c\, \lambda_p\left(A^{-1}\right) > c_1 > 0,$$

where last inequality follows from (B.11). We can now prove that, with high probability, $\Delta = \Omega(p)$. Using Cauchy-Schwarz, it holds that

$$b^2 = |\langle u, v \rangle|_{A^{-1}} \leq \|u\|_{A^{-1}} \|v\|_{A^{-1}} = ad,$$

from which it follows that

$$\Delta = a\left(n\frac{\gamma_t}{\gamma} + d\right) - (1-b)^2 \geq an\frac{\gamma_t}{\gamma} - 1 + 2b = \Omega(p),$$

since $a$ is lower bounded by a constant and $|b| \leq c\sqrt{p}$. At this point, we have all the necessary bounds and we work towards proving the claim. To simplify notation we denote by $\hat{\Sigma}_s^0 := \left(\frac{X_s^{0\top} X_s^0}{n}\right)$ and $\tilde{\beta} = \beta_s - \beta_t$. We first expand the bias term

$$
\begin{aligned}
B_1 &= \tilde{\beta}^\top \hat{\Sigma}_s^0 \hat{\Sigma}^{-1} \Sigma_t \hat{\Sigma}^{-1} \hat{\Sigma}_s^0 \tilde{\beta} \\
&= \tilde{\beta}^\top \hat{\Sigma}_s^0 \hat{\Sigma}^{-1} \Sigma_t (A + UCU^\top)^{-1} \hat{\Sigma}_s^0 \tilde{\beta} \\
&= \tilde{\beta}^\top \hat{\Sigma}_s^0 \hat{\Sigma}^{-1} \Sigma_t \left(A^{-1} - A^{-1} U \left(C^{-1} - U^\top A^{-1} U\right)^{-1} U^\top A^{-1}\right) \hat{\Sigma}_s^0 \tilde{\beta} \\
&= \tilde{\beta}^\top \hat{\Sigma}_s^0 \hat{\Sigma}^{-1} \Sigma_t \hat{\Sigma}_0^{-1} \hat{\Sigma}_s^0 \tilde{\beta} + S,
\end{aligned}
$$

where $S := -\tilde{\beta}^\top \hat{\Sigma}_s^0 \hat{\Sigma}^{-1} \Sigma_t A^{-1} U \left(C^{-1} - U^\top A^{-1} U\right)^{-1} U^\top A^{-1} \hat{\Sigma}_s^0 \tilde{\beta}$.

We now prove that $S$ is small. To do so, we decompose

$$
\begin{aligned}
S &= -\tilde{\beta}^\top \hat{\Sigma}_s^0 \hat{\Sigma}^{-1} \Sigma_t A^{-1} U \left(C^{-1} - U^\top A^{-1} U\right)^{-1} U^\top A^{-1} \hat{\Sigma}_s^0 \tilde{\beta} \\
&= \tilde{\beta}^\top \hat{\Sigma}_s^0 \hat{\Sigma}^{-1} \Sigma_t \frac{1}{\Delta} A^{-1} \left(\left(n\frac{\gamma_t}{\gamma} + d\right) uu^\top + (1-b)\left(uv^\top + vu^\top\right) + a\, vv^\top\right) A^{-1} \hat{\Sigma}_s^0 \tilde{\beta} \\
&= T_{u,u} + T_{u,v} + T_{v,v},
\end{aligned}
$$

where $T_{u,u}$ is the summand corresponding to $uu^\top$, $T_{u,v}$ to $uv^\top + vu^\top$, and $T_{v,v}$ to $vv^\top$. Zooming in on one of the terms, it holds that

$$
\begin{aligned}
T_{u,u} &= \tilde{\beta}^\top \hat{\Sigma}_s^0 \hat{\Sigma}^{-1} \Sigma_t \frac{(n\gamma_t/\gamma + d)}{\Delta} A^{-1}\, uu^\top A^{-1} \hat{\Sigma}_s^0 \tilde{\beta} \\
&= \left\langle \tilde{\beta}, \hat{\Sigma}_s^0 \hat{\Sigma}^{-1} \Sigma_t \frac{(n\gamma_t/\gamma + d)}{\Delta} A^{-1}\, u \right\rangle \left\langle u^\top A^{-1} \hat{\Sigma}_s^0, \tilde{\beta} \right\rangle.
\end{aligned}
$$

Note that

$$
\left\| \hat{\Sigma}_s^0 \hat{\Sigma}^{-1} \Sigma_t \frac{(n\gamma_t/\gamma + d)}{\Delta} A^{-1}\, u \right\|_2 \leq \left\|\hat{\Sigma}_s^0\right\|_2 \left\|\hat{\Sigma}^{-1}\right\|_2 \|\Sigma_t\|_2 \frac{(n\gamma_t/\gamma + d)}{\Delta} \left\|A^{-1}\right\|_2 \|u\|_2 \leq c
$$

and $\left\|u^\top A^{-1} \hat{\Sigma}_s^0\right\|_2 \leq c$. Using this, we get that, with high probability, it holds

$$
|T_{u,u}| \leq \frac{c}{p}.
$$

This is similar to how we obtained (A.31), since $\beta_s$ and $\beta_t$ are sampled independently from a sphere of constant radius. With analogous passages, we have that

$$
|T_{u,v}| \leq \frac{c}{p}, \qquad |T_{v,v}| \leq \frac{c}{p}
$$

holds with high probability over the sampling of $\beta_s$ and $\beta_t$. Putting all together, we get

$$
B_1 = \tilde{\beta}^\top \hat{\Sigma}_s^0 \hat{\Sigma}^{-1} \Sigma_t \hat{\Sigma}_0^{-1} \hat{\Sigma}_s^0 \tilde{\beta} + O\left(\frac{1}{p}\right).
$$

Using the same argumentation applied to the remaining term $\hat{\Sigma}^{-1}$ in $\tilde{\beta}^\top \hat{\Sigma}_s^0 \hat{\Sigma}^{-1} \Sigma_t \hat{\Sigma}_0^{-1} \hat{\Sigma}_s^0 \tilde{\beta}$ gives

$$
B_X^1(\lambda) = \tilde{\beta}^\top \hat{\Sigma}_s^0 \hat{\Sigma}_0^{-1} \Sigma_t \hat{\Sigma}_0^{-1} \hat{\Sigma}_s^0 \tilde{\beta} + O\left(\frac{1}{p}\right),
$$

proving the claim. ♣

By directly applying Yang et al. (2025)[Theorem 6] we obtain that

$$
\left| \tilde{\beta}^\top \hat{\Sigma}_s^0 \hat{\Sigma}_0^{-1} \Sigma_t \hat{\Sigma}_0^{-1} \hat{\Sigma}_s^0 \tilde{\beta} - \left\|\Sigma_s^{1/2}(\beta_s - \beta_t)\right\|_2^2 \cdot s_1 \right| = O\left(\frac{1}{p^{1/2-c}}\right).
$$

Finally plugging in (B.7) yields

$$
\left| B_1 - \left\|\Sigma_s^{1/2}(\beta_s - \beta_t)\right\|_2^2 \cdot s_1 \right| = O\left(\frac{1}{p^{1/2-c}}\right).
$$

**Bounding the term $B_2$.** We follow the recipe devised in the proof of Proposition A.3. If we denote by $\tilde{X}_n = \frac{X}{\sqrt{n}}$, then it holds

$$\left\| \hat{\Sigma}^{-1} \mu_t \right\|_2^2 = \left\| \left( \tilde{X}_n^\top \tilde{X}_n \right)^{-1} \mu_t \right\|_2^2$$

$$= \left\| \sum_{i=1}^p \frac{1}{\sigma_i(\tilde{X}_n)^2} v_i(\tilde{X}_n) v_i(\tilde{X}_n)^\top \mu_t \right\|_2^2$$

$$= \frac{\left\langle v_1(\tilde{X}_n), \mu_t \right\rangle^2}{\sigma_1(\tilde{X}_n)^4} + \frac{\left\langle v_2(\tilde{X}_n), \mu_t \right\rangle^2}{\sigma_2(\tilde{X}_n)^4} + \sum_{i=3}^p \frac{\left\langle v_i(\tilde{X}_n), \mu_t \right\rangle^2}{\sigma_i(\tilde{X}_n)^4}$$

$$\leq \Theta\left(\frac{1}{p}\right)\left(1 - O\left(\frac{1}{p}\right)\right) + \frac{1}{c \cdot \tau} O(1)$$

$$= O(1),$$

where the penultimate inequality follows directly from (A.9) and Proposition A.2. This means that the vector $\mu_t^\top \hat{\Sigma}^{-1} \left( \frac{X_s^{0\top} X_s^0}{n} \right)$ is bounded, since

$$\left\| \mu_t^\top \hat{\Sigma}^{-1} \left( \frac{X_s^{0\top} X_s^0}{n} \right) \right\|_2 \leq \left\| \mu_t^\top \hat{\Sigma}^{-1} \right\|_2 \left\| \left( \frac{X_s^{0\top} X_s^0}{n} \right) \right\|_2 = O(1),$$

as an application of the Bai–Yin theorem (Bai & Silverstein, 2010, Theorem 5.11) gives a constant bound on $\left\| \left( \frac{X_s^{0\top} X_s^0}{n} \right) \right\|_2$. Since $\beta_s$ and $\beta_t$ are sampled independently from a sphere of constant radius and $\mu_t^\top \hat{\Sigma}^{-1} \left( \frac{X_s^{0\top} X_s^0}{n} \right)$ is of bounded norm, we have that $\left| \left\langle \left( \frac{X_s^{0\top} X_s^0}{n} \hat{\Sigma}^{-1} \mu_t \right), \beta_{(i)} \right\rangle \right|^2$ is sub-exponential. Using Bernstein inequality, we obtain that, for $(i) \in \{s, t\}$,

$$\left( \mu_t^\top \hat{\Sigma}^{-1} \left( \frac{X_s^{0\top} X_s^0}{n} \right) \beta_{(i)} \right)^2 = \left| \left\langle \left( \frac{X_s^{0\top} X_s^0}{n} \hat{\Sigma}^{-1} \mu_t \right), \beta_{(i)} \right\rangle \right|^2 = O\left(\frac{1}{p}\right), \tag{B.12}$$

with high probability over the sampling of $\beta_s$ and $\beta_t$. Therefore, we have

$$B_2 = \left( \mu_t^\top \hat{\Sigma}^{-1} \left( \frac{X_s^{0\top} X_s^0}{n} \right) (\beta_s - \beta_t) \right)^2 = O\left(\frac{1}{p}\right),$$

where the last inequality follows by separating $\beta_s$ and $\beta_t$ from the square and using (B.12) on each term.

**Bounding the term $V_X(\hat{\beta}; \beta_t, \beta_s)$.** Note that the variance term is the same as in the case without model shift. This means that we could simply use the already derived results from Section 4. Namely, directly from the proof of Theorem 4.1, we have that for $M = \Sigma_s^{1/2} \Sigma_t^{-1/2}$ and $\lambda_1 \geq \cdots \geq \lambda_p$ eigenvalues of $M^\top M$, it holds

$$\left| V_X(\hat{\beta}; \beta_t, \beta_s) - \frac{\sigma^2}{n} \mathrm{Tr}\left[ \left( \alpha_1 M^\top M + \alpha_2 I_p \right)^{-1} \right] \right| = O(p^{-1/2}),$$

where $\alpha_1$ and $\alpha_2$ are the unique positive solutions to the following two equations

$$\alpha_1 + \alpha_2 = 1 - \frac{p}{n}, \quad \alpha_1 + \frac{1}{n} \sum_{i=1}^p \frac{\lambda_i \alpha_1}{\lambda_i \alpha_1 + \alpha_2} = \frac{n_s}{n}.$$

As we assume that $\Sigma_t = \Sigma_s$, it follows that $M^\top M = I_p$ and

$$\frac{\sigma^2}{n} \mathrm{Tr}\left[ \left( \alpha_1 M^\top M + \alpha_2 I_p \right)^{-1} \right] = \sigma^2 \frac{p}{n-p},$$

which ultimately yields

$$\left| V_X(\hat{\beta}; \beta_t, \beta_s) - \sigma^2 \frac{p}{n-p} \right| = O(p^{-1/2}).$$

Finally by combining the bounds on $B_X(\hat{\beta}; \beta_t, \beta_s) = B_1 + B_2$ and $V_X(\hat{\beta}; \beta_t, \beta_s)$, the desired claim follows. $\qquad\square$

## C ADDITIONAL NUMERICAL RESULTS

**Setup details.** We train for 200 epochs using SGD as optimizer, and we use cosine annealing; the initial learning rate is 0.1 for *Scratch* (0.2 for the experiment of Table 3a) and 0.01 for *Distillation* and *Pretrained*. The *Distillation* teacher is a ResNet-50 trained on CIFAR-10. We use an early stopping with patience 20 based on a validation subset (10% of the full training dataset). We avoid up-scaling images in the *Pretrained* experiments to better demonstrate the effect of synthetic data augmentation. On the generation side, to generate the images by T2I models, we use CLIP's text encoder prompt template on CIFAR-10 and ImageNet labels. Moreover, as models like StableDiffusion1.4 sometimes generate low quality data or images discarded by the safety checker, before applying all the algorithms, we do an initial pruning of 2% of the generated pool based on the distance to the CLIP embedding of the label. For RxRx1, we train a linear classifier on frozen features from an ImageNet-pretrained ResNet. For each class, MorphGen generates a pool of 500 synthetic images; we augment the real training set (30 images/class) with 60 selected synthetic images/class and evaluate on a disjoint test set of 20 images/class. We repeat the experiment 10 times by resampling the real subset from 120 images/class. As in the main setup, CLIP features are used for the selection algorithms.

**Transformer-based models.** In Table 4, we use the same setup as Table 1, but instead of ResNet, we train a ViT and a Swin-T model from scratch. We use a patch size of 4 and Adam optimizer with learning rate 0.0001 for this experiment. We observe that, in accordance with our previous findings, covariance matching surpasses other algorithms.

Table 4: *Covariance matching* outperforms all baselines when fully training a transformer model on a mix of real and synthetic data.

| Method | ViT | | Swin-T | |
|---|---|---|---|---|
| | Scratch | Distillation | Scratch | Distillation |
| No synthetic | $40.11 \pm 0.59$ | $40.32 \pm 1.01$ | $40.02 \pm 0.70$ | $40.84 \pm 0.73$ |
| Center matching (He et al., 2023) | $43.89 \pm 0.97$ | $45.61 \pm 0.68$ | $44.39 \pm 0.54$ | $46.64 \pm 0.53$ |
| Center sampling (Lin et al., 2023) | $43.89 \pm 0.95$ | $46.29 \pm 0.80$ | $43.94 \pm 1.76$ | $46.97 \pm 0.59$ |
| DS3 (Hulkund et al., 2025) | $45.92 \pm 0.49$ | $48.61 \pm 0.67$ | $46.57 \pm 0.68$ | $49.55 \pm 0.72$ |
| K-means (Lin et al., 2023) | $44.24 \pm 1.13$ | $47.44 \pm 0.97$ | $44.71 \pm 0.32$ | $48.49 \pm 0.64$ |
| Random | $44.07 \pm 0.82$ | $46.50 \pm 0.78$ | $44.38 \pm 0.77$ | $47.35 \pm 0.50$ |
| Text matching (Lin et al., 2023) | $44.57 \pm 0.57$ | $46.02 \pm 1.00$ | $45.15 \pm 0.58$ | $46.55 \pm 2.52$ |
| Text sampling (Lin et al., 2023) | $43.80 \pm 0.98$ | $46.00 \pm 0.98$ | $44.59 \pm 0.93$ | $47.62 \pm 0.71$ |
| Covariance matching (ours) | $46.09 \pm 0.91$ | $49.53 \pm 0.61$ | $46.64 \pm 0.96$ | $50.73 \pm 0.44$ |
| Real upper bound | $51.85 \pm 0.47$ | $53.11 \pm 0.43$ | $52.43 \pm 1.39$ | $54.80 \pm 0.69$ |

**Zero-diversity generators.** To assess the importance of filtering low-diversity data, we construct a pool per CIFAR-10 class with 2K images from StyleGAN2-Ada and 8K images from two collapsed generators. The first collapsed model emits the image whose CLIP embedding is closest to the class label; the second produces images near the mean embedding of the class's real subset. We sample 4K images from each collapsed generator, yielding a total 10K images per class. As shown in Table 5, most baselines over-select from the collapsed generators because they ignore the diversity of selected samples. In particular, DS3 retains the two clusters formed by the collapsed outputs and thus fails to filter them. By contrast, K-means and Covariance matching draw more from the 2K non-collapsed subset and achieve higher classification accuracy.

**Leak experiment.** We consider inserting ("leaking") images from the target distribution into the pool of synthetic images and test the ability of different methods to select them. We use 1K leaked CIFAR-10 images, disjoint from the 200 ($n_t$) real reference samples. From a pool of 4K StableDiffusion1.4 images and 1K leaked images, each method selects 800 ($n_s$). Figure 2 shows, for each method, the fraction of selected samples drawn from the leak. Because replacing synthetic with real augmentations yields the best accuracy (*Real upper bound*), an effective selector should prioritize leaked real images: covariance matching does, achieving the highest leaked fraction among all methods.

Table 5: *Covariance matching* performs on par with the best baselines across three training paradigms on CIFAR-10, when the synthetic data is generated via a StyleGAN2-Ada model and two zero-diversity generators.

| Method | Scratch | Distillation | Pretrained |
|---|---|---|---|
| No synthetic | $44.36 \pm 1.51$ | $47.33 \pm 0.57$ | $63.40 \pm 1.33$ |
| Center matching (He et al., 2023) | $45.33 \pm 2.43$ | $47.50 \pm 0.55$ | $62.96 \pm 1.26$ |
| Center sampling (Lin et al., 2023) | $46.88 \pm 2.59$ | $51.11 \pm 0.60$ | $65.38 \pm 1.14$ |
| DS3 (Hulkund et al., 2025) | $53.74 \pm 1.92$ | $59.16 \pm 1.56$ | $69.43 \pm 0.93$ |
| K-means (Lin et al., 2023) | $60.20 \pm 1.35$ | $65.03 \pm 0.81$ | $72.83 \pm 0.48$ |
| Random | $50.31 \pm 1.28$ | $51.82 \pm 0.91$ | $66.27 \pm 1.21$ |
| Text matching (Lin et al., 2023) | $42.89 \pm 1.89$ | $47.38 \pm 0.76$ | $62.82 \pm 1.31$ |
| Text sampling (Lin et al., 2023) | $48.13 \pm 1.81$ | $50.81 \pm 0.77$ | $66.12 \pm 1.06$ |
| Covariance matching (ours) | $58.97 \pm 1.67$ | $64.85 \pm 0.63$ | $72.38 \pm 0.66$ |
| Real upper bound | $61.08 \pm 2.54$ | $65.38 \pm 0.51$ | $74.35 \pm 0.56$ |

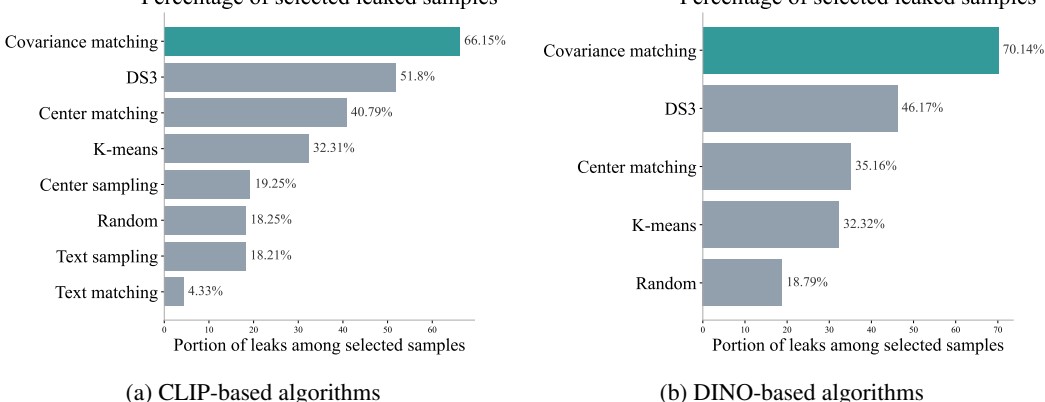

(a) CLIP-based algorithms          (b) DINO-based algorithms

Figure 2: The portion of samples chosen from the set of leaked images shows that our proposed algorithm reliably selects real samples among the pool of generated examples.

**Changing the feature extractor.** In the main experiments, we use CLIP features for all selection methods. To test the dependence on the feature extractor, we repeat the setups of Tables 1-2 with DINO-v2 features. As shown in Tables 6-7, covariance matching matches or surpasses the best baseline across settings, indicating that its effectiveness is not tied to a specific feature extractor. We also repeat the leak experiment of Figure 2, see the bar plot in (b), showing again similar results.

Table 6: *Covariance matching* outperforms all baselines across three training paradigms on CIFAR-10, when the synthetic data is generated via truncated generative models and features are extracted with DINO-v2.

| Method | Scratch | Distillation | Pretrained |
|---|---|---|---|
| No synthetic | $44.36 \pm 1.51$ | $47.33 \pm 0.57$ | $63.40 \pm 1.33$ |
| Center matching (He et al., 2023) | $50.06 \pm 1.45$ | $54.50 \pm 0.62$ | $66.23 \pm 0.72$ |
| DS3 (Hulkund et al., 2025) | $52.93 \pm 1.65$ | $58.69 \pm 0.81$ | $68.04 \pm 0.71$ |
| K-means (Lin et al., 2023) | $51.66 \pm 2.10$ | $55.97 \pm 0.58$ | $67.00 \pm 0.84$ |
| Random | $49.97 \pm 2.45$ | $54.79 \pm 0.68$ | $66.57 \pm 0.92$ |
| Text matching (Lin et al., 2023) | $51.52 \pm 1.67$ | $55.17 \pm 0.57$ | $67.13 \pm 0.45$ |
| Covariance matching (ours) | $54.97 \pm 2.60$ | $59.41 \pm 0.81$ | $68.87 \pm 0.41$ |
| Real upper bound | $61.08 \pm 2.54$ | $65.38 \pm 0.51$ | $74.35 \pm 0.56$ |

**Optimizing the theoretical objective.** We also implement a greedy algorithm that, at each step, adds the sample minimizing the objective in (4.1) (*Alpha matching*). This method requires computing the eigenvalues of the current sample covariance and is therefore more costly than *Covariance*

Table 7: *Covariance matching* performs on par with the best baseline across three training paradigms on CIFAR-10, when the synthetic data is generated via text-to-image (T2I) generative models and features are extracted with DINO-v2.

| Method | Scratch | Distillation | Pretrained |
|---|---|---|---|
| No synthetic | $44.36 \pm 1.51$ | $47.33 \pm 0.57$ | $63.40 \pm 1.33$ |
| Center matching (He et al., 2023) | $51.75 \pm 2.01$ | $55.67 \pm 0.63$ | $66.00 \pm 0.58$ |
| DS3 (Hulkund et al., 2025) | $52.33 \pm 2.07$ | $58.80 \pm 0.96$ | $66.68 \pm 0.63$ |
| K-means (Lin et al., 2023) | $51.14 \pm 1.90$ | $56.93 \pm 0.46$ | $65.71 \pm 0.71$ |
| Random | $50.45 \pm 1.41$ | $55.86 \pm 0.73$ | $65.67 \pm 0.82$ |
| Text matching (Lin et al., 2023) | $51.38 \pm 1.51$ | $55.81 \pm 0.65$ | $65.76 \pm 1.00$ |
| Covariance matching (ours) | $52.65 \pm 1.47$ | $58.78 \pm 0.53$ | $67.04 \pm 0.83$ |
| Real upper bound | $61.08 \pm 2.54$ | $65.38 \pm 0.51$ | $74.35 \pm 0.56$ |

*matching*. As in *Covariance matching*, we first fit PCA on the real samples and project all features, then iteratively add the sample that yields the smallest value of (4.1). Without loss of generality, we drop the noise variance term since it scales all candidates equally. The results of Table 8 show that *Alpha matching* performs similarly to *Covariance matching*.

Table 8: *Covariance matching* performs on par with *Alpha matching* across the experiments on CIFAR-10.

| Experiment | Method | Scratch | Distillation | Pretrained |
|---|---|---|---|---|
| Zero-diversity models | Covariance matching | $58.97 \pm 1.67$ | $64.85 \pm 0.63$ | $72.38 \pm 0.66$ |
| | Alpha matching | $59.30 \pm 2.50$ | $64.72 \pm 0.55$ | $72.76 \pm 0.73$ |
| Truncated models | Covariance matching | $54.00 \pm 1.89$ | $59.77 \pm 0.61$ | $69.20 \pm 0.56$ |
| | Alpha matching | $52.25 \pm 2.11$ | $59.18 \pm 0.68$ | $68.32 \pm 0.58$ |
| T2I models | Covariance matching | $54.45 \pm 2.11$ | $59.17 \pm 0.64$ | $66.69 \pm 0.70$ |
| | Alpha matching | $53.37 \pm 1.85$ | $59.03 \pm 0.64$ | $66.23 \pm 0.66$ |

**Over-parameterized setting.** We repeat the setup of Table 1 taking $n_s = 200$ (instead of $n_s = 800$). This gives a total of $n_s + n_t = 400$ samples, which is less than the number of features $p = 512$, thus placing us in an over-parameterized regime. As shown in Table 9, the quantitative trends mirror those in the under-parameterized case.

Table 9: *Covariance matching* outperforms all baselines across three training paradigms on CIFAR-10, when the synthetic data is generated via truncated StyleGAN2-Ada models (Karras et al., 2019) in the over-parameterized regime with 200 training and 200 augmenting synthetic samples.

| Method | Scratch | Distillation | Pretrained |
|---|---|---|---|
| No synthetic | $44.36 \pm 1.51$ | $47.33 \pm 0.57$ | $63.40 \pm 1.33$ |
| Center matching (He et al., 2023) | $46.45 \pm 1.97$ | $50.83 \pm 0.50$ | $64.40 \pm 1.11$ |
| Center sampling (Lin et al., 2023) | $47.29 \pm 1.33$ | $50.89 \pm 0.78$ | $65.64 \pm 0.74$ |
| DS3 (Hulkund et al., 2025) | $48.09 \pm 2.04$ | $52.65 \pm 0.61$ | $66.41 \pm 1.35$ |
| K-means (Lin et al., 2023) | $47.75 \pm 0.82$ | $51.56 \pm 0.68$ | $65.47 \pm 0.99$ |
| Random | $47.39 \pm 1.63$ | $50.96 \pm 0.22$ | $65.49 \pm 1.12$ |
| Text matching (Lin et al., 2023) | $47.56 \pm 1.09$ | $51.67 \pm 0.65$ | $65.74 \pm 0.78$ |
| Text sampling (Lin et al., 2023) | $46.93 \pm 1.95$ | $50.64 \pm 0.49$ | $65.13 \pm 1.13$ |
| Covariance matching (ours) | $48.95 \pm 1.28$ | $53.28 \pm 0.45$ | $66.62 \pm 0.57$ |
| Real upper bound | $50.79 \pm 1.70$ | $54.66 \pm 0.91$ | $68.97 \pm 0.88$ |

**Distribution of selected samples.** Beyond accuracy, we assess how well each method's selections match the test distribution. In the CIFAR-10 setup of Table 1, each method selects 800 samples per class given 200 real samples. We then calculate how well these samples match the CIFAR-10 training dataset. The selection obtained via *Covariance matching* consistently achieves lower FID/KID and covariance distance than all other baselines. Metrics that couple fidelity and diversity (e.g., FID/KID)

show larger gains than quality metrics (e.g., Precision (Kynkäänniemi et al., 2019), Density (Naeem et al., 2020)), indicating improved distributional alignment rather than mere sample quality. The results are reported in Table 10.

Table 10: *Covariance matching* selects samples that better match the target distribution according to various evaluation metrics.

| Method | FID ↓ | KID ↓ | Precision ↑ | Recall ↑ | Density ↑ | Coverage ↑ | Covariance Shift ↓ |
|---|---|---|---|---|---|---|---|
| K-means (Lin et al., 2023) | $366.52 \pm 2.62$ | $0.59 \pm 0.04$ | $0.77 \pm 0.01$ | $0.41 \pm 0.00$ | $0.87 \pm 0.04$ | $0.58 \pm 0.01$ | $118.91 \pm 0.62$ |
| Center matching (He et al., 2023) | $544.56 \pm 5.57$ | $0.83 \pm 0.06$ | $0.78 \pm 0.01$ | $0.33 \pm 0.01$ | $0.82 \pm 0.03$ | $0.49 \pm 0.01$ | $212.55 \pm 3.03$ |
| Center sampling (Lin et al., 2023) | $450.27 \pm 3.86$ | $0.61 \pm 0.04$ | $0.77 \pm 0.01$ | $0.44 \pm 0.01$ | $0.86 \pm 0.03$ | $0.53 \pm 0.01$ | $150.49 \pm 0.79$ |
| DS3 (Hulkund et al., 2025) | $273.59 \pm 6.72$ | $0.42 \pm 0.04$ | $0.79 \pm 0.01$ | $0.45 \pm 0.01$ | $0.84 \pm 0.03$ | $0.64 \pm 0.01$ | $106.52 \pm 2.44$ |
| Random | $458.39 \pm 4.16$ | $0.63 \pm 0.04$ | $0.77 \pm 0.02$ | $0.44 \pm 0.01$ | $0.86 \pm 0.05$ | $0.53 \pm 0.01$ | $150.66 \pm 1.08$ |
| Text matching (Lin et al., 2023) | $454.23 \pm 2.66$ | $0.69 \pm 0.05$ | $0.81 \pm 0.01$ | $0.36 \pm 0.00$ | $0.90 \pm 0.03$ | $0.54 \pm 0.01$ | $172.70 \pm 0.66$ |
| Text sampling (Lin et al., 2023) | $447.53 \pm 3.99$ | $0.61 \pm 0.04$ | $0.77 \pm 0.01$ | $0.44 \pm 0.01$ | $0.86 \pm 0.03$ | $0.53 \pm 0.01$ | $149.98 \pm 0.95$ |
| Covariance matching (ours) | $242.09 \pm 1.93$ | $0.41 \pm 0.04$ | $0.78 \pm 0.01$ | $0.50 \pm 0.01$ | $0.84 \pm 0.03$ | $0.68 \pm 0.01$ | $95.55 \pm 0.58$ |

**Real and synthetic dataset sizes.** Next, we analyze the behavior of selection methods for different values of real dataset size and synthetic samples gathered. We use the setup of Table 1, with training from scratch, and change the size of real samples $n_t \in \{200, 300\}$ and synthetic samples $n_s \in \{600, 800, 1000\}$ in Table 11.

Table 11: An ablation on the size of the real and synthetic datasets used for training shows results consistent with those reported in Table 1.

| Method | (200, 600) | (200, 800) | (200, 1000) | (300, 600) | (300, 800) | (300, 1000) |
|---|---|---|---|---|---|---|
| No synthetic | $44.36 \pm 1.51$ | $44.36 \pm 1.51$ | $44.36 \pm 1.51$ | $47.85 \pm 1.21$ | $47.85 \pm 1.21$ | $47.85 \pm 1.21$ |
| Center matching | $47.51 \pm 3.04$ | $50.04 \pm 2.84$ | $50.60 \pm 2.87$ | $51.66 \pm 2.19$ | $53.61 \pm 3.57$ | $54.75 \pm 3.44$ |
| Center sampling | $50.14 \pm 2.42$ | $50.48 \pm 2.03$ | $51.40 \pm 2.84$ | $51.00 \pm 2.22$ | $52.97 \pm 3.11$ | $55.00 \pm 3.45$ |
| DS3 | $50.62 \pm 1.25$ | $52.83 \pm 2.19$ | $53.02 \pm 3.18$ | $53.33 \pm 2.26$ | $55.91 \pm 2.80$ | $55.79 \pm 2.70$ |
| K-means | $49.65 \pm 2.52$ | $50.74 \pm 1.77$ | $51.31 \pm 3.32$ | $52.10 \pm 2.53$ | $52.65 \pm 3.65$ | $54.22 \pm 3.13$ |
| Random | $48.07 \pm 2.80$ | $49.38 \pm 2.43$ | $49.34 \pm 2.37$ | $52.38 \pm 2.50$ | $52.96 \pm 3.82$ | $53.47 \pm 2.66$ |
| Text matching | $49.90 \pm 2.70$ | $50.94 \pm 1.40$ | $50.72 \pm 2.54$ | $52.27 \pm 2.52$ | $53.04 \pm 2.64$ | $52.03 \pm 1.05$ |
| Text sampling | $49.19 \pm 2.30$ | $50.28 \pm 1.18$ | $49.98 \pm 3.40$ | $52.26 \pm 2.78$ | $51.86 \pm 1.96$ | $53.23 \pm 3.28$ |
| Covariance matching (ours) | $51.91 \pm 2.05$ | $54.00 \pm 1.89$ | $54.64 \pm 3.79$ | $55.38 \pm 3.27$ | $55.63 \pm 2.70$ | $55.80 \pm 2.25$ |
| Upper bound | $57.85 \pm 1.86$ | $61.08 \pm 2.54$ | $60.99 \pm 2.11$ | $58.76 \pm 1.85$ | $61.11 \pm 2.87$ | $62.79 \pm 2.56$ |

**Variations for covariance matching.** We experiment with two other variations for covariance matching. First, incorporating more compute, we analyze a look-ahead strategy. At each round, we select the top $k$ samples for minimizing the covariance. We then exhaustively test each pair of these $k$ samples (total of $\binom{k}{2}$ possibilities) and find out which pair minimizes the covariance shift. We then only add the first element of the pair. This models a look-ahead strategy for adding a sample during each round. Second, we perform a method to find the best samples globally. To find $n_s$ data among $M$ samples, we repeat the $n_t$ dataset and sample to get $n_s$ real data points. We then use the Hungarian algorithm to match the $n_s$ real and $M$ synthetic datasets. After matching, we select only the $n_s$ samples matched from the $M$ total synthetic data points as our selected synthetic data. This imitates a global strategy to match the covariance of the data points. We use the setup from Table 1 and report the accuracies in Table 12, observing that the performance is on par with covariance matching.

Table 12: The look-ahead algorithm with $k \in \{50, 100\}$ and the Hungarian algorithm perform on par with the greedy implementation of covariance matching across three training paradigms on CIFAR-10, when the synthetic data is generated via five truncated StyleGAN2-Ada models.

| Method | Scratch | Distillation | Pretrained |
|---|---|---|---|
| Look-ahead (50) | $54.91 \pm 1.60$ | $59.56 \pm 0.71$ | $68.10 \pm 0.84$ |
| Look-ahead (100) | $53.84 \pm 0.95$ | $59.79 \pm 0.67$ | $68.19 \pm 0.74$ |
| Hungarian | $53.10 \pm 1.37$ | $58.54 \pm 1.07$ | $68.60 \pm 0.87$ |
| Greedy Covariance Matching | $54.00 \pm 1.89$ | $59.77 \pm 0.61$ | $69.20 \pm 0.56$ |

**Experiment on a text dataset.** To further demonstrate the generality of our approach beyond vision tasks, we evaluate *Covariance Matching* on a text classification problem. Following observations reported by Li et al. (2023), we use the Tweet Irony dataset (Van Hee et al., 2018) as the real dataset and GPT-generated tweets from Kuo et al. (2025) as the synthetic dataset. The synthetic corpus contains 3K prompts per class. We sample 100 tweets from the real dataset and augment them with 300 synthetic samples. We extract sentence embeddings using the all-mpnet-base-v2 model (Song et al., 2020), and apply our selection algorithms in this feature space. A linear classifier is then trained on the combined real and selected synthetic samples to classify ironic vs. non-ironic tweets. Table 13 shows that synthetic augmentations improve performance, and *Covariance Matching* outperforms all baselines, serving as a theory-driven heuristic for selecting synthetic data.

Table 13: *Covariance matching* outperforms all baselines on the Ironic-Tweet dataset.

| Method | Tweet Irony |
| --- | --- |
| No synthetic | $64.60 \pm 3.16$ |
| Center Matching | $70.80 \pm 1.54$ |
| Random | $67.97 \pm 1.20$ |
| K-means | $68.85 \pm 1.26$ |
| DS3 | $69.73 \pm 2.51$ |
| Covariance Matching (ours) | $71.49 \pm 1.59$ |

