# OpenReview forum: "High-dimensional Analysis of Synthetic Data Selection"
_ICLR.cc/2026/Conference — ICLR 2026 Oral_

### Official Review · Reviewer_D1Gm · 2025-10-17

**Soundness:** 3
**Presentation:** 3
**Contribution:** 3
**Rating:** 6
**Confidence:** 3

**Summary:**

This paper investigates the incorporation of both synthetic and real datasets in a high-dimensional linear regression context to minimize the test error associated with the real data distribution.
The two principal conclusions are:
(1) In the high-dimensional limit, the mean shift between the real and synthetic distributions does not impact the asymptotic test error.
(2) An optimal synthetic data distribution has covariance proportional to that of the real distribution.
The authors also introduces a simple and greedy covariance-matching algorithm for data selection, and show it succeeds empirically.

**Strengths:**

This paper is clearly written and easy to understand.

The findings are surprising to me: I do not know any previous results showing that the mean shift does not matter in combining real and synthetic data for training. Also, the optimal covariance expression makes a lot of sense.

Though the theory is built on toy models, the covariance matching algorithm seems to work in more realistic problems like the imagine classification tasks in this paper.

**Weaknesses:**

In this paper, only direct mixing of the synthetic and real data is considered. If we assume that we are able to assign different weights to synthetic and real data, will we get different conclusions? For example, under this circumstance, can we still get the same conclusion on the optimal covariance matrix?

Also, it seems that the assumption of synthetic and real data sharing the same true parameter $\beta$ is too strong, can you argue if this is common in practice? If no, how to relax this assumption?

The experiments are only conducted in the image dataset. Could you add more experiments on the language models?

**Questions:**

None

---

> ### Author Response · Authors · 2025-11-20
>
> We thank the reviewer for the comments and for the positive evaluation of our work. We address all concerns and questions below, one by one. We have edited our paper accordingly and uploaded a revised version, where the changes are marked in blue color to make them more visible.
>
> >Weakness 1. In this paper, only direct mixing of the synthetic and real data is considered. If we assume that we are able to assign different weights to synthetic and real data, will we get different conclusions? For example, under this circumstance, can we still get the same conclusion on the optimal covariance matrix?
>
> **Response**: Assigning different weights to synthetic and real data can be readily handled by the theory we have developed without much adjustment, and it leads to the same conclusions. This has also been noted in the zero-centered case by Yang et al., (2025), and Song et al., (2024).
>
> Putting this into context, let us suppose we work in the setting where expected label noise intensity of $\varepsilon_{(i)}$ differs between real and synthetic data. A natural way to account for this in training is to assign weights inversely proportional to the corresponding noise levels, i.e. assigning greater weight on datapoints we trust more. Thus, let $w_1$ be the weight assigned to the real data, and $w_2$ the weight assigned to synthetic data. This leads to three possible regimes:
>
> 1. $w_1 \ll w_2$. Here synthetic data effectively determines the estimator. In this context, we would be in the scenario of Proposition 4.2 of our paper. As discussed there, both the mean and covariance discrepancy play a role, and thus the resulting optimality condition differs from the mixed-data case.
>
> 2. $w_1 \gg w_2$. In this regime, the estimator is driven almost entirely by the real data, reducing to the classical setting studied by Hastie et al. (2022). Consequently, the choice of synthetic data has negligible impact.
>
> 3. $w_1\sim w_2$. This is arguably the most interesting case. The current theory applies directly, since weighting real and synthetic observations is equivalent to scaling $(\Sigma_t,\mu_t)$ by $w_1$ and $(\Sigma_s,\mu_s)$ by $w_2$. Then, by adjusting Theorems 4.1, 4.3, 4.4 and 4.5, one can prove that the optimal choice becomes  $\Sigma_s \sim \frac{w_1}{w_2} \Sigma_t$. Thus, the qualitative conclusion remains unchanged: matching the synthetic covariance to that of the real data is optimal, where the weighting reflects the appropriate scaling depending on the trust placed in the labels.
>
> We clarify this point in l. 195-198, and l. 2178-2196. of the revision.
>
> >Weakness 2. Also, it seems that the assumption of synthetic and real data sharing the same true parameter \beta  is too strong, can you argue if this is common in practice? If no, how to relax this assumption?
>
> **Response**: We would like to point out that the assumption that real and synthetic data share the same underlying parameter $\beta$ is rather common in practice. In modern synthetic data pipelines, such as diffusion-based label synthesis, or augmentations derived from fitted predictive models, the synthetic labels are produced precisely to approximate the real data generating mechanism, see (Pham et al., 2021). Thus, the shared $\beta$ assumption reflects a common and intentional design choice, and therefore serves as a realistic first-order modeling approximation.
>
> That being said, the assumption can indeed be relaxed. In the statistical literature, a possible difference in the hidden parameters that generate real and synthetic data is known as model shift, see (Yang et al., 2025; Song et al., 2024). Allowing for such a shift complicates the analysis, because the excess risk now depends on the interaction between the hidden parameter and covariance mismatch, requiring additional technical tools. Nevertheless, partial progress is possible. In the under-parameterized regime, assuming that covariances of synthetic and real data are already matched, we derive an explicit excess-risk expression that accounts for model shift. Consequently, we can conclude that depending on the intensity of the model shift, synthetic data may have limited or even negative effects. By characterizing this, the derived formula suggests a simple training heuristic of subsampling synthetic data until performance decreases.
>
> A comprehensive treatment of model shift in the presence of covariance shift is significantly harder and requires new technical tools. We view this as an interesting direction for future work as discussed in Section 6.
>
> The added model shift results and discussion can be found l.65-66, l. 165-166, l. 504, and in Appendix B, l. 2199 - 2483 of the revision.

---

> ### Author Response · Authors · 2025-11-20
>
> >Weakness 3. The experiments are only conducted in the image dataset. Could you add more experiments on the language models?
>
> **Response.** As suggested by the reviewer, we have added an experiment on language models in Appendix C. Following observations reported by Li et al. (2023), we use the Tweet Irony dataset (Van Hee et al., 2018) as the real dataset and GPT-generated tweets from (Kuo et al., 2025) as the synthetic dataset. The synthetic corpus contains 3K prompts per class. We sample 100 tweets from the real dataset and augment them with 300 synthetic samples. We extract sentence embeddings using the all-mpnet-base-v2 model (Song et al., 2020), and apply our selection algorithms in this feature space. A linear classifier is then trained on the combined real and selected synthetic samples to classify ironic vs. non-ironic tweets. The following table (Table 13 of the revised paper) shows that synthetic augmentations improve performance, and covariance matching outperforms all baselines, serving as a theory-driven heuristic for selecting synthetic data.
>
> | Method                      | Tweet Irony         |
> |----------------------------|----------------|
> | No synthetic               | 64.60 ± 3.16    |
> | Center Matching            | 70.80 ± 1.54    |
> | Random                     | 67.97 ± 1.20    |
> | K-means                    | 68.85 ± 1.26    |
> | DS3                        | 69.73 ± 2.51    |
> | Covariance Matching (ours) | 71.49 ± 1.59    |
>
> (Li et al., 2023) Zhuoyan Li, Hangxiao Zhu, Zhuoran Lu, and Ming Yin. Synthetic data generation with large language models for text classification: Potential and limitations. In _Proceedings of the 2023 Conference on Empirical Methods in Natural Language Processing_, pp. 10443–10461, 2023.
>
> (Kuo et al., 2025) Hsun-Yu Kuo, Yin-Hsiang Liao, Yu-Chieh Chao, Wei-Yun Ma, and Pu-Jen Cheng. Not all LLM- generated data are equal: Rethinking data weighting in text classification. In _The Thirteenth International Conference on Learning Representations_, 2025.
>
> (Song et al., 2020) Kaitao Song, Xu Tan, Tao Qin, Jianfeng Lu, and Tie-Yan Liu. Mpnet: Masked and permuted pre-training for language understanding. _Advances in neural information processing systems_, 33: 16857–16867, 2020.
>
> (Van Hee et al., 2018) Cynthia Van Hee, Els Lefever, and Véronique Hoste. Semeval-2018 task 3: Irony detection in english tweets. In _Proceedings of the 12th international workshop on semantic evaluation_, pp. 39–50, 2018.
>
> (Pham et al., 2021) Hieu Pham, Zihang Dai, Qizhe Xie, and Quoc V. Le. "Meta pseudo labels." In _Proceedings of the IEEE/CVF conference on computer vision and pattern recognition_, pp. 11557-11568. 2021.
>
> (Hastie et al., 2022) Trevor Hastie, Andrea Montanari, Saharon Rosset, and Ryan J. Tibshirani. "Surprises in high-dimensional ridgeless least squares interpolation." _Annals of statistics_ 50, no. 2: 949, 2022.
>
> (Song et al., 2024) Yanke Song, Sohom Bhattacharya, and Pragya Sur. "Generalization error of min-norm interpolators in transfer learning." _arXiv preprint arXiv:2406_.13944, 2024.
>
> (Yang et al., 2025) Fan Yang, Hongyang R. Zhang, Sen Wu, Christopher Ré, and Weijie J. Su. "Precise high-dimensional asymptotics for quantifying heterogeneous transfers." _Journal of Machine Learning Research_ 26, no. 113: 1-88, 2025.

---

> > ### Comment · Reviewer_D1Gm · 2025-11-20
> >
> > Thank you for the rebuttal. Most of my concerns are addressed.
> >
> > I've updated my score to 8.

---

> > > ### Author Response · Authors · 2025-11-24
> > >
> > > Thank you for your appreciation of our work and of our rebuttal, and for raising the score. We remain at disposal in case any additional question arises.

---

### Official Review · Reviewer_1CSm · 2025-10-27

**Soundness:** 4
**Presentation:** 4
**Contribution:** 2
**Rating:** 6
**Confidence:** 4

**Summary:**

This paper investigates how to optimally select synthetic data to minimize test error and enhance generalization performance.. The authors first analyze this problem in a linear ridgeless regression setting and derive an optimal data selection criterion that minimizes the test risk of the trained regressor. Specifically, the optimal strategy consists of selecting the set of synthetic samples whose covariance matrix is closest (in Frobenius distance) to that of the real data, a method they call covariance matching, while paying less attention to discrepancies in the first-order moment (the mean vector). Based on these theoretical insights, the authors design a practical algorithm for synthetic data selection, evaluate it on multiple classification tasks on CIFAR-10 and ImageNet, and validate their theoretical findings empirically.

**Strengths:**

This paper has many strengths and advantages, among which:

1) The authors prove an interesting and somewhat counterintuitive result stating that the gap between the mean vector of the true data $\mu_t$ and the synthetic one $\mu_s$ **does not** impact the generalization error in their linear regression setting when training on a mixture of real + synthetic data. Consequently, the selection criterion needs only to consider the second-order moment (covariance).

2) They provide a rigorous theoretical study of their problem through a linear regression setting both in the low (under-parametreized) and high-dimensional (overparameterized) regimes.

3) The paper is overall well-written and the experiments section is extensive and well-detailed and showing promising results.

**Weaknesses:**

The main weaknesses of this paper are related to the novelty of their theoretical analysis and the absence of some important references in the same subject:

1) **Novelty:** I am somewhat concerned about the originality of the covariance matching idea. In fact, the notion that discrepancies between the covariance matrices of real and synthetic data degrade the quality of the generated samples was already introduced and analyzed in a recent ICLR 2025 paper [1]. That work also studied training on a mixture of real and synthetic data under a theoretical high-dimensional binary classification setting. Therefore, in my view, this raises questions about the novelty of the present paper’s contributions.

2) **Lack of key references:** The authors claim in the conclusion that "they take the first step in understanding the precise connection between training on a mix of real and synthetic data and generalizing on real data". This does not hold as previous works (that were not cited) have also tackled this same problem: [1], [2] and [3].


[1] Aymane El Firdoussi, Mohamed El Amine Seddik, Soufiane Hayou, Reda Alami, Ahmed Alzubaidi, Hakim Hacid. Maximizing the Potential of Synthetic data: Insights from Random Matrix Theory. ICLR 2025

[2] Bertrand, Q., Bose, A. J., Duplessis, A., Jiralerspong, M., and Gidel, G. On the stability of iterative retraining of generative models on their own data, ICLR 2024 spotlight

[3] Mohamed El Amine Seddik, Suei-Wen Chen, Soufiane Hayou, Pierre Youssef, Merouane Debbah. How bad is training on synthetic data? a statistical analysis of language model collapse. COLM 2024

**Questions:**

My questions are mostly related to the weaknesses discussed earlier (see **Weaknesses** section), along with few other minor remarks.

1) How does your work connect to or extend previous studies that have theoretically analyzed the problem of mixing real and synthetic data, particularly the work of Firdoussi et al. (2024) [1]?

2) I am somewhat skeptical about the theoretical result claiming that the mean vector does not affect generalization performance. Could this outcome stem from simplifying assumptions in your theoretical setup ? I also believe that discrepancies in the mean vectors should not pose a serious issue, since the real data mean can typically be estimated consistently, whereas the covariance matrix cannot in high dimensions (as described by the Marchenko–Pastur law).

3) In Figure 1, the authors evaluated their theoretical findings using mean vectors $\mu_s$ and $\mu_t$ of norms in the order of $\mathcal{O}(\sqrt{p})$. Could you justify this choice, knowing that the dimension $p$ scales to infinity in high-dimensions ?

4) Regarding the covariance matching algorithm described in the experiments section, do you think that adding multiple synthetic samples at once (rather than one at a time) would result in a different final synthetic dataset ?

I would be happy to reconsider my score once my concerns have been addressed.

---

> ### Author Response · Authors · 2025-11-20
>
> We thank the reviewer for the comments and for the appreciation of our work. We address all concerns and questions below, one by one. We have edited our paper accordingly and uploaded a revised version, where the changes are marked in blue color to make them more visible.
>
> >Weakness 1. Novelty: I am somewhat concerned about the originality of the covariance matching idea. In fact, the notion that discrepancies between the covariance matrices of real and synthetic data degrade the quality of the generated samples was already introduced and analyzed in a recent ICLR 2025 paper [1]. That work also studied training on a mixture of real and synthetic data under a theoretical high-dimensional binary classification setting. Therefore, in my view, this raises questions about the novelty of the present paper’s contributions.
>
> **Response.** We thank the reviewer for pointing out the work (Firdoussi et al., 2025) which also studies training on a mix of real and synthetic data in a high-dimensional regime. While (Firdoussi et al., 2025) is related to our paper, let us point out several important differences in the setting and in the resulting insights, showcasing that these two works crucially have different objectives.
> Our paper focuses on the multi-class classification setting in which data augmentations occur per class. For this reason, we analyze each class separately, modeling them as a linear regression problem where the means and covariances of real and synthetic data may be arbitrary. This modelling setting is common to the statistical transfer learning literature, see (Yang et al., 2025; Song et al., 2024). Our work then studies how an arbitrary distribution shift in the augmented data affects the performance and, from there, derives a procedure to select samples e.g. obtained via a generative model.
>
> In contrast, (Firdoussi et al., 2025) works in a binary classification setting in which the two classes share the isotropic covariance matrix and have means that differ by a sign, determining class membership. In their setup the synthetic data is generated using estimates derived from the real data. The goal of (Firdoussi et al., 2025) is to understand the factors influencing performance when generating synthetic data using estimated statistics from real data.
>
> The main insights of our paper are that when selecting synthetic data per class (i) the mean of the distributions does not matter, and (ii) it is best to match covariance. On the other hand, (Firdoussi et al., 2025) analyzes performance under estimation of covariance and label noise, pointing to degrading performance in case of bad estimation. Thus, the two works are complementary: they both operate in the high-dimensional regime, but have distinct settings and derive different high-level insights.
>
> We now discuss the related work (Firdoussi et al., 2025) in  l. 120-128 of the revision.

---

> ### Author Response · Authors · 2025-11-20
>
> >Weakness 2. Lack of key references: The authors claim in the conclusion that "they take the first step in understanding the precise connection between training on a mix of real and synthetic data and generalizing on real data". This does not hold as previous works (that were not cited) have also tackled this same problem: [1], [2] and [3].
>
> **Response.** We thank the reviewer for pointing out the related works (Firdoussi et al., 2025, Bertrand et. al., 2024, Seddik et al., 2024) . These papers are indeed related to our work, each with a distinct perspective on training on a mix of real and synthetic data. While they have a different setting and pursue a different objective, they all work with a distribution shift that arises when synthetic data is incorporated into training, ultimately complementing the insights we make.
> As discussed above in our response to Weakness 1, the work most directly related to ours is (Firdoussi et al., 2025), which analyzes high-dimensional binary classification. Its goal is to understand the factors influencing performance when generating synthetic data using estimated statistics from real data, pointing to degrading performance in case of bad estimation. In contrast, works (Bertrand et. al., 2024) and (Seddik et al., 2024) focus on a different aspect of training on real and synthetic data, namely, the behavior of iterative retraining that repeatedly uses synthetic data. Both papers focus on training stability and the mechanisms underlying model collapse, a phenomenon linked in part to statistical approximation error as identified in prior work (Shumailov et al., 2024). Specifically, (Bertrand et. al., 2024) develops a theoretical framework for iterative retraining of generative models on mixtures of real and self-generated data. One conclusion is that, with sufficiently large real training data, iterative training using self-generated synthetic data can in fact remain stable. Conversely, when these conditions fail, (Bertrand et. al., 2024)  shows both theoretically and empirically that iterative retraining leads to collapse. Meanwhile, (Seddik et al., 2024) studies analogous effects in the context of language modeling, offering a statistical analysis of distribution shift and demonstrating that repeated training on purely synthetic generations inevitably drives the model toward collapse of the next-token distribution.
>
> We now discuss the related works (Firdoussi et al., 2025, Bertrand et. al., 2024, Seddik et al., 2024) in l. 120-128 of the revision, and we have replaced the claim “This paper offers the first step in understanding the precise connection between training on a mix of real and synthetic data and generalizing on real data.” with “This paper advances understanding of the precise connection between training on a mix of real and synthetic data and generalizing on real data.”(see l. 492 of the revision).
>
> >Question 1. How does your work connect to or extend previous studies that have theoretically analyzed the problem of mixing real and synthetic data, particularly the work of Firdoussi et al. (2024) [1]?
>
> **Response.** While both works aim to shine light on the use of synthetic data in a high-dimensional regime, they are complementary in their goals. Each investigates a different problem setting, which consequently yields complementary high-level insights, see our response above to Weakness 1 and l. 120-124 of the revision.

---

> ### Author Response · Authors · 2025-11-20
>
> >Question 2. I am somewhat skeptical about the theoretical result claiming that the mean vector does not affect generalization performance. Could this outcome stem from simplifying assumptions in your theoretical setup ? I also believe that discrepancies in the mean vectors should not pose a serious issue, since the real data mean can typically be estimated consistently, whereas the covariance matrix cannot in high dimensions (as described by the Marchenko–Pastur law).
>
> **Response.** We agree with the intuition of the reviewer: the mean vector is indeed implicitly estimated by the estimator $\hat{\beta}$ in the high-dimensional proportional regime, which is why the means do not influence the risk. This can be seen from the closed form expression for $\hat{\beta}$​, i.e. Equation (3.4), where the pseudo-inverse of the uncentered sample covariance appears. Proposition A.2 in the appendix then shows that the dominant eigendirection of the uncentered sample covariance is precisely the mean. Consequently, the direction of the mean is always recovered by the estimator $\hat{\beta}$ during training.
>
> We also note that our conclusion that the generalization performance is independent of the mean relies on the way we formalize the problem. In fact, by doing our analysis per class via linear regression, the mean vectors should not be interpreted as the primary quantity correlated to the class label, but rather the class identity is encoded by the hidden signal $\beta$. In contrast, in the related work (Firdoussi et al., 2025), estimating the mean is important because the mean vector is identified with the class label. We further remark that our assumptions in the linear regression setting are standard in the transfer learning literature, see (Yang et al., 2025; Song et al., 2024).
>
> Let us finally mention that the validity of the proposed method (covariance matching) which does not rely on the means of the distributions is extensively validated in practical settings in Section 5, as well as in the ablations of Appendix C.
>
>
> >Question 3. In Figure 1, the authors evaluated their theoretical findings using mean vectors  and  of norms in the order of $O(\sqrt{p})$. Could you justify this choice, knowing that the dimension scales to infinity in high-dimensions?
>
> **Response.** Choosing the norm of the mean vector of order $\sqrt{p}$ is tightly connected to our modelling via linear regression. Recall that, as mentioned above, the distribution mean is actually not the only thing indicative of class membership, since our analysis is performed within each class. The class identity in the broader sense is encoded by the signal vector $\beta$, which is assumed to be uncorrelated with the class-conditional mean. Given that, consider the quantity $y_t$ as given in Equation (3.1). It holds that $\mathbb{E}[y_t] = \langle\mu_t,\beta\rangle$ and Var$[y_t] = \beta^\top \Sigma_t \beta \sim \Theta(1)$. Therefore, the only scaling ensuring that the mean plays a non-trivial role in the selection is $||\mu_t||_2\sim \sqrt{p}$, which implies that $\langle\mu_t,\beta\rangle \sim\Theta(1)$. In fact, if $||\mu_t||_2\ll\sqrt{p}$, then $\langle\mu_t,\beta\rangle\ll 1$, so the mean has a vanishing effect on the risk, implying trivially the conclusion that the means do not matter. Furthermore, if $||\mu_t||_2\gg \sqrt{p}$, then $\langle\mu_t,\beta\rangle \gg 1$, so the output $y_t$ diverges. In this case the mean would dominate the risk entirely, again making the analysis trivial. We clarify this point in l. 212-214 of the revision.

---

> ### Author Response · Authors · 2025-11-20
>
> >Question 4. Regarding the covariance matching algorithm described in the experiments section, do you think that adding multiple synthetic samples at once (rather than one at a time) would result in a different final synthetic dataset?
>
> **Response.** While adding multiple synthetic samples at once may result in a slightly different selection, the greedy method adding one sample at a time considered in Section 5 offers an excellent trade-off between computational complexity and generalization performance. To demonstrate this, we conducted an additional ablation (Table 12, revised paper) comparing two new variants of covariance matching.
>
> (1) Look-ahead: At each iteration, we first select the top-$k$ candidates that most reduce the covariance shift. We then exhaustively evaluate all $k \choose 2$ pairs, identify the pair that yields the lowest shift, and add only the first element of that pair. This simulates a more computationally intensive, look-ahead selection strategy. (2) Hungarian: To select $n_s$ samples from $M$ synthetic candidates, we replicate the $n_t$ real samples to form a comparison set of size $n_s$, then use the Hungarian algorithm to match these against the $M$ synthetic samples. We keep only the $n_s$ matched synthetic samples.
>
> Using the same experimental configuration as Table 1, results in the following table (Table 12 of the revised paper) show that both variants perform comparably to Covariance Matching, despite requiring more computation.
>
> | Method                     | Scratch           | Distillation      | Pretrained       |
> |---------------------------|------------------|------------------|------------------|
> | Look-ahead (50)           | 54.91 ± 1.60      | 59.56 ± 0.71      | 68.10 ± 0.84      |
> | Look-ahead (100)          | 53.84 ± 0.95      | 59.79 ± 0.67      | 68.19 ± 0.74      |
> | Hungarian                 | 53.10 ± 1.37      | 58.54 ± 1.07      | 68.60 ± 0.87      |
> | Greedy Covariance Matching| 54.00 ± 1.89      | 59.77 ± 0.61      | 69.20 ± 0.56      |
>
>
> (Firdoussi et al., 2025) Aymane  El Firdoussi, Mohamed El Amine Seddik, Soufiane Hayou, Reda Alami, Ahmed Alzubaidi, and Hakim Hacid. "Maximizing the Potential of Synthetic Data: Insights from Random Matrix Theory." In _The Thirteenth International Conference on Learning Representations_, 2025.
>
> (Bertrand et. al., 2024) Quentin Bertrand, Joey Bose, Alexandre Duplessis, Marco Jiralerspong, and Gauthier Gidel. "On the Stability of Iterative Retraining of Generative Models on their own Data." In _The Twelfth International Conference on Learning Representations_, 2024.
>
> (Seddik et al., 2024) Mohamed El Amine Seddik, Suei-Wen Chen, Soufiane Hayou, Pierre Youssef, and Merouane Abdelkader Debbah. "How bad is training on synthetic data? A statistical analysis of language model collapse." In _First Conference on Language Modeling_, 2024.
>
> (Shumailov et al., 2024) Ilia Shumailov, Zakhar Shumaylov, Yiren Zhao, Nicolas Papernot, Ross Anderson, and Yarin Gal. "AI models collapse when trained on recursively generated data." _Nature_ 631, no. 8022 (2024): 755-759.
>
> (Song et al., 2024) Yanke Song, Sohom Bhattacharya, and Pragya Sur. "Generalization error of min-norm interpolators in transfer learning." _arXiv preprint arXiv:2406_.13944, 2024.
>
> (Yang et al., 2025) Fan Yang, Hongyang R. Zhang, Sen Wu, Christopher Ré, and Weijie J. Su. "Precise high-dimensional asymptotics for quantifying heterogeneous transfers." _Journal of Machine Learning Research_ 26, no. 113: 1-88, 2025.

---

> > ### Comment · Reviewer_1CSm · 2025-11-21
> > **Response to Authors' rebuttal**
> >
> > I thank the authors for this excellent rebuttal.
> >
> > They have addressed all my questions and remarks, included new experiments to support their claims and answers, and revised their manuscript to include other important works on the same problem.
> >
> > Overall, I am very satisfied by the quality of their responses, and I have therefore raised my score from 6 to 8 and highly recommend acceptance of this work as it introduces a simple and efficient algorithm to address a critical problem spanning almost all Machine Learning fields.
> >
> > - **Additional question:** Regarding my previous Question 3, the authors claimed that having a mean vector of norm $\Vert \mu \Vert_2 = \mathcal{O}(\sqrt{p})$ implies that $\langle \beta, \mu \rangle = \Theta(1)$. Can you please justify this claim ? What is the order of magnitude of the norm of the vector $\beta$ ?

---

> > > ### Author Response · Authors · 2025-11-24
> > >
> > > We thank the reviewer for the appreciation of our work and of our rebuttal, and for raising the score.
> > >
> > > Regarding the question, if $\beta$ has norm of constant order and it is sampled uniformly at random independently of other quantities, then the claim $\langle\beta, \mu\rangle = \Theta(1)$ follows from the concentration of Lipschitz functions on the sphere, see e.g. Theorem 5.1.4 in the book “High Dimensional Probability” by Vershynin. More precisely, we have that $|\langle\beta, \mu\rangle| \le c \lVert\beta \rVert_2 \sqrt{ln(1/\delta)}$ for a universal constant $c$ (independent of $p, n$), with probability at least $1-\delta$.
> > >
> > > We note that these assumptions on $\beta$ are not formally required to hold in order to obtain the theoretical results in the under-parameterized regime. In fact, in that case the risk does not depend on $\beta$ since the bias term is $0$. Once the bias term is non-trivial, which is the case in the over-parametrized regime, we make these assumptions explicit (see the beginning of Section 4.2, at l. 323). In the over-parametrized regime, the bias term scales directly with $\lVert\beta\rVert_2^2$ (see equation (3.5)), so under a different scaling it would either diverge or vanish.
> > > Lastly, a complementary way to see this scaling is natural, is to consider $\text{Var} {[y_t]}  = \beta^\top \Sigma_t \beta \sim \lVert\beta\rVert_2^2$, and $\mathbb{E}[y_t]=\langle\mu,\beta\rangle\sim \frac{\lVert \mu \rVert_2 \lVert \beta \rVert_2}{ \sqrt{p}}$. Namely, setting $\lVert \beta \rVert_2 \sim \Theta(1)$ and $\lVert \mu \rVert_2  \sim \sqrt{p}$ is the only way to obtain a normalized output with non-trivial impact from the mean.

---

### Official Review · Reviewer_GUb9 · 2025-11-01

**Soundness:** 3
**Presentation:** 3
**Contribution:** 2
**Rating:** 4
**Confidence:** 3

**Summary:**

This paper presents a theoretical and empirical study regarding synthetic data selection for augmenting training datasets. It focuses on the effects of mean shift and covariance shift by theoretical analysis in high-dimensional linear regression and experiments on vision models. Their findings suggest that covariance shift rather than mean shift affects generalization error when training on a mixture of real and synthetic data, and show that the covariance matching approach for selecting synthetic data would improve model performance.

**Strengths:**

1. The paper is theoretically sound by using high-dimensional linear regression scenarios and findings from experiments align with their theoretical analysis.
2. Although the conclusion regarding covariance shift matters is expected, the paper offers a theoretical framework that formalizes and explains this intuition.
3. The connection between covariance matching selection and evaluation metrics such as FID and recall is quite interesting and provides insights into generative data quality.

**Weaknesses:**

1. In the introduction, the notation ($X_t,y_t$) is used to denote both the training and test datasets, which may confuse readers.
2. The theoretical analysis holds strong, simplified assumptions, building on high-dimensional linear regression with Gaussian data. It doesn't account for nonlinearities, non-Gaussian feature distributions that characterize deep learning models in practice.
3.  All experiments are conducted on vision tasks. It is unclear whether the findings hold for language tasks, which typically involve more complex architectures.
4.  It states that training data should not be too small compared to synthetic data, but it is not formally quantified. In the experiment, the authors use 200 real and 800 synthetic samples per class, but it is unclear how varying the real/synthetic ratio affects the theoretical predictions or empirical results. Similarly, the paper mentions an upper bound on diversity scaling but does not provide a concrete characterization of when diversity stops being beneficial.
5. The definition of diversity remains ambiguous. It is not clear whether it refers to information within training data or generated diversity beyond it. If it refers to within, does covariance matching include the concepts of diversity?

**Questions:**

1. How is covariance matching different from data quality measures? What makes covariance matching fundamentally different from these existing measures of data fidelity? Such as FID or other metrics.
2. What is the relationship between covariance matching and diversity? Does matching covariance automatically mean the data are more diverse, or is diversity a broader concept that includes more than covariance?

---

> ### Author Response · Authors · 2025-11-20
>
> We thank the reviewer for the comments. We address all concerns and questions below, one by one. We have edited our paper accordingly and uploaded a revised version, where the changes are marked in blue color to make them more visible.
>
> >Weakness 1. In the introduction, the notation $(X_t,y_t)$ is used to denote both the training and test datasets, which may confuse readers.
>
> **Response.** Thanks for pointing this out. We have added an additional, different notation for the test dataset in page 1, in order to avoid confusion.
>
> >Weakness 2. The theoretical analysis holds strong, simplified assumptions, building on high-dimensional linear regression with Gaussian data. It doesn't account for nonlinearities, non-Gaussian feature distributions that characterize deep learning models in practice.
>
> **Response.** While we agree that the theoretical analysis comes with strong assumptions, we would like to note that we do not make any Gaussianity assumption on the data. Furthermore, all experimental results discussed in Section 5 (as well as the additional ablations of Appendix C) are performed on non-linear deep learning models used in practice (ResNets, transformers).
>
> >Weakness 3. All experiments are conducted on vision tasks. It is unclear whether the findings hold for language tasks, which typically involve more complex architectures.
>
> **Response.** As suggested by the reviewer, to analyze the findings for language tasks, we added an experiment on tweet classification in Appendix C. Following observations reported by Li et al. (2023), we use the Tweet Irony dataset (Van Hee et al., 2018) as the real dataset and GPT-generated tweets from (Kuo et al., 2025) as the synthetic dataset. The synthetic corpus contains 3K prompts per class. We sample 100 tweets from the real dataset and augment them with 300 synthetic samples. We extract sentence embeddings using the all-mpnet-base-v2 model (Song et al., 2020), and apply our selection algorithms in this feature space. A linear classifier is then trained on the combined real and selected synthetic samples to classify ironic vs. non-ironic tweets. The following table (Table 13 of the revised paper) shows that synthetic augmentations improve performance, and covariance matching outperforms all baselines, serving as a theory-driven heuristic for selecting synthetic data.
>
> | Method                      | Tweet Irony         |
> |----------------------------|----------------|
> | No synthetic               | 64.60 ± 3.16    |
> | Center Matching            | 70.80 ± 1.54    |
> | Random                     | 67.97 ± 1.20    |
> | K-means                    | 68.85 ± 1.26    |
> | DS3                        | 69.73 ± 2.51    |
> | Covariance Matching (ours) | 71.49 ± 1.59    |

---

> ### Author Response · Authors · 2025-11-20
>
> >Weakness 4. It states that training data should not be too small compared to synthetic data, but it is not formally quantified. In the experiment, the authors use 200 real and 800 synthetic samples per class, but it is unclear how varying the real/synthetic ratio affects the theoretical predictions or empirical results. Similarly, the paper mentions an upper bound on diversity scaling but does not provide a concrete characterization of when diversity stops being beneficial.
>
> **Response.** Formally, our theory requires that the sample size of the synthetic data $n_s$ is comparable with the dimension $p$, but the proportionality constant can be arbitrarily small: in l. 176, we assume that $n_s\ge \tau p$ for some constant $\tau>0$ which can be taken arbitrarily small (as long as it does not depend on $p$).
>
> In practice, we found that varying the ratio between real and synthetic sample sizes does not affect our conclusions significantly. To demonstrate this, we now report in Appendix C additional experiments using the same setup as in Table 1 but changing the sample sizes to combinations of (200, 300) real and (600,800,1000) synthetic samples. In Table 11 of the revised paper (see also below), we observe a similar behavior as in Table 1.
>
> | Method                    | (200, 600)        | (200, 800)        | (200, 1000)       | (300, 600)        | (300, 800)        | (300, 1000)       |
> |--------------------------|-------------------|-------------------|-------------------|-------------------|-------------------|-------------------|
> | No synthetic             | 44.36 ± 1.51      | 44.36 ± 1.51      | 44.36 ± 1.51      | 47.85 ± 1.21      | 47.85 ± 1.21      | 47.85 ± 1.21      |
> | Center matching          | 47.51 ± 3.04      | 50.04 ± 2.84      | 50.60 ± 2.87      | 51.66 ± 2.19      | 53.61 ± 3.57      | 54.75 ± 3.44      |
> | Center sampling          | 50.14 ± 2.42      | 50.48 ± 2.03      | 51.40 ± 2.84      | 51.00 ± 2.22      | 52.97 ± 3.11      | 55.00 ± 3.45      |
> | DS3                      | 50.62 ± 1.25      | 52.83 ± 2.19      | 53.02 ± 3.18      | 53.33 ± 2.26      | 55.91 ± 2.80      | 55.79 ± 2.70      |
> | K-means                  | 49.65 ± 2.52      | 50.74 ± 1.77      | 51.31 ± 3.32      | 52.10 ± 2.53      | 52.65 ± 3.65      | 54.22 ± 3.13      |
> | Random                   | 48.07 ± 2.80      | 49.38 ± 2.43      | 49.34 ± 2.37      | 52.38 ± 2.50      | 52.96 ± 3.82      | 53.47 ± 2.66      |
> | Text matching            | 49.90 ± 2.70      | 50.94 ± 1.40      | 50.72 ± 2.54      | 52.27 ± 2.52      | 53.04 ± 2.64      | 52.03 ± 1.05      |
> | Text sampling            | 49.19 ± 2.30      | 50.28 ± 1.18      | 49.98 ± 3.40      | 52.26 ± 2.78      | 51.86 ± 1.96      | 53.23 ± 3.28      |
> | Covariance matching (ours) | 51.91 ± 2.05    | 54.00 ± 1.89      | 54.64 ± 3.79      | 55.38 ± 3.27      | 55.63 ± 2.70      | 55.80 ± 2.25      |
> | Upper bound              | 57.85 ± 1.86      | 61.08 ± 2.54      | 60.99 ± 2.11      | 58.76 ± 1.85      | 61.11 ± 2.87      | 62.79 ± 2.56      |
>
> Lastly, note that, in our theoretical model, the noise levels of real and synthetic data are assumed to be identical. Under this assumption, fixing the noise level also effectively fixes the admissible diversity scaling of the synthetic data. Consequently, increasing diversity scaling is beneficial only up to the point where it remains comparable to that of the original dataset, as formalized by Theorems 4.3 and 4.5. If, however, the expected noise intensities of the real and synthetic datasets differ, then a natural way to account for this in training is to assign weights inversely proportional to the corresponding noise levels, i.e. assigning greater weight on data points we trust more. This setting is exactly captured by the weighted objective extension introduced in line in l. 195-198, and l. 2178-2196 of the revision, see also our response to weakness 1 of Reviewer D1Gm. The optimal diversity scaling is the one proportional to the confidence we place in the synthetic labels, see l. 2193. Thus, increasing synthetic diversity remains beneficial only insofar as it remains consistent with the reliability of the corresponding labels.
>
>
> >Weakness 5. The definition of diversity remains ambiguous. It is not clear whether it refers to information within training data or generated diversity beyond it. If it refers to within, does covariance matching include the concepts of diversity?
>
> **Response.** Generally, we’d rather refrain from giving a formal definition of ‘diversity’, see also our answer to Question 2 below which discusses the relationship between covariance matching and diversity. However, we would like to clarify that diversity refers to the ‘generated diversity beyond the training data’ and we have now added this clarification in l. 419-421 of the revision.

---

> ### Author Response · Authors · 2025-11-20
>
> >Question 1. How is covariance matching different from data quality measures? What makes covariance matching fundamentally different from these existing measures of data fidelity? Such as FID or other metrics.
>
> **Response.** What makes covariance matching fundamentally different from other existing measures of data fidelity is that it is provably optimal in the sense of the analysis of Section 4. Having said that, we do expect covariance matching to be linked to other existing measures of data fidelity. In fact, Table 10 in Appendix C shows that covariance matching selects samples that better match the target distribution according to various evaluation metrics, such as FID, KID, recall, coverage and covariance shift.
>
> >Question 2. What is the relationship between covariance matching and diversity? Does matching covariance automatically mean the data are more diverse, or is diversity a broader concept that includes more than covariance?
>
> **Response.** As our analysis is based on linear models, we intuitively interpret the covariance as a measure of diversity (e.g., samples with small covariance will be more concentrated towards the mean). In the generative modeling case, it is less obvious that the covariance is the only relevant feature to capture diversity. In our experiments, we use StyleGANs with different truncation factors, as the original paper refers to the truncation factor as a way of controlling the diversity of the generated samples. There, we find that our approach works well and, when mixing samples from multiple StyleGANs with different truncations, our model prefers the variant that generates more diverse samples (high truncation). To demonstrate this, we have analyzed the number of samples chosen from each StyleGAN in the experiment of Table 1. We observe that 268, 245, 333 samples are chosen from the three StyleGANs with truncation 0.2, and 3692, 3462 samples are chosen from the two StyleGANs with truncation 0.6, which shows the preference of covariance matching towards more diverse samples. We have clarified this point in l. 421-423 of the revision.
>
> (Li et al., 2023) Zhuoyan Li, Hangxiao Zhu, Zhuoran Lu, and Ming Yin. Synthetic data generation with large language models for text classification: Potential and limitations. In _Proceedings of the 2023 Conference on Empirical Methods in Natural Language Processing_, pp. 10443–10461, 2023.
>
> (Kuo et al., 2025) Hsun-Yu Kuo, Yin-Hsiang Liao, Yu-Chieh Chao, Wei-Yun Ma, and Pu-Jen Cheng. Not all LLM- generated data are equal: Rethinking data weighting in text classification. In _The Thirteenth International Conference on Learning Representations_, 2025.
>
> (Song et al., 2020) Kaitao Song, Xu Tan, Tao Qin, Jianfeng Lu, and Tie-Yan Liu. Mpnet: Masked and permuted pre-training for language understanding. _Advances in neural information processing systems_, 33: 16857–16867, 2020.
>
> (Van Hee et al., 2018) Cynthia Van Hee, Els Lefever, and Véronique Hoste. Semeval-2018 task 3: Irony detection in english tweets. In _Proceedings of the 12th international workshop on semantic evaluation_, pp. 39–50, 2018.

---

> > ### Author Response · Authors · 2025-11-27
> >
> > We kindly ask the reviewer whether our responses have addressed their questions to their satisfaction or if anything remains unresolved. We remain available for any further discussion.

---

### Author Response · Authors · 2025-11-20
**General Response**

We thank the reviewers for all their constructive feedback, which has allowed us to improve our revised manuscript.

In this rebuttal we have carefully addressed the reviewers' questions. All the updates in the revision are highlighted in blue for clarity. We summarize here the main changes in the revision, and respond below to each reviewer individually:
- As asked by the Reviewers GUb9 and D1Gm, we have added an experiment on language tasks, strengthening the theoretical finding about covariance matching.
- As asked by the Reviewer GUb9, we have added an experiment which ablates the ratio between the size of real and synthetic data, obtaining results in agreement with theoretical predictions.
- As pointed out by Reviewer 1CSm, we have found the paper (Firdoussi et al., 2025) very relevant and added a discussion, which also concerns the additional related works (Bertrand et. al., 2024,Seddik et al., 2024).
- As asked by the Reviewer 1CSm, we have clarified the choice of mean norm in the theoretical setup.
- As asked by the Reviewer 1CSm, we have added experiments evaluating alternatives to greedily adding one synthetic sample at a time, all of which achieve similar performance.
- As asked by the Reviewer D1Gm, we have included a discussion on adding different weights to synthetic and real data, which yields the same conclusion on the optimality of covariance matching.
- As asked by the Reviewer D1Gm, we have added new theoretical results that relax the assumption that synthetic and real data share the same true parameter $\beta$.
- We have separated and updated the proofs of the bound on the variance term in the under-parameterized (Proposition A.3) and over-parameterized regime (this now includes an intermediate result, Claim A.42), in order to correct an imprecision and improve the structure.



(Firdoussi et al., 2025) Aymane  El Firdoussi, Mohamed El Amine Seddik, Soufiane Hayou, Reda Alami, Ahmed Alzubaidi, and Hakim Hacid. "Maximizing the Potential of Synthetic Data: Insights from Random Matrix Theory." In _The Thirteenth International Conference on Learning Representations_, 2025.

(Bertrand et. al., 2024) Quentin Bertrand, Joey Bose, Alexandre Duplessis, Marco Jiralerspong, and Gauthier Gidel. "On the Stability of Iterative Retraining of Generative Models on their own Data." In _The Twelfth International Conference on Learning Representations_, 2024.

(Seddik et al., 2024) Mohamed El Amine Seddik, Suei-Wen Chen, Soufiane Hayou, Pierre Youssef, and Merouane Abdelkader Debbah. "How bad is training on synthetic data? A statistical analysis of language model collapse." In _First Conference on Language Modeling_, 2024.

---

### Meta-Review · Area_Chair_2W9N · 2026-01-06

**Summary:**

The reviewers questioned whether the theoretical results, derived under a high-dimensional linear regression framework, would remain valid beyond simplified assumptions, in particular in non-linear and non-Gaussian settings relevant to practice.
They also noted that the initial experimental evaluation was largely vision-focused and asked whether the conclusions would extend to language tasks.
Further concerns were raised about the treatment of the real-to-synthetic data ratio and the notion of “diversity,” which were initially insufficiently quantified and defined.
In addition, reviewers asked for a clearer positioning with respect to prior work, especially Firdoussi et al. (2025), and whether the proposed contributions were genuinely distinct.
The counter-intuitive claim that mean shift does not affect generalization prompted questions about its dependence on modeling assumptions and on the specific scaling choices used in the theory.
There were also methodological questions about the robustness of the proposed greedy covariance-matching algorithm compared to selecting multiple samples at once.
 Finally, reviewers inquired about extensions beyond the base setting, including the effect of weighting real versus synthetic data and the strength of the assumption that real and synthetic data share the same underlying parameter.

**Reviewer Concerns:**

The rebuttal and revised manuscript addressed all of the major reviewer concerns raised during the review process.

Specifically, concerns about the restrictive theoretical assumptions were addressed by clarifying that the analysis does not rely on Gaussianity and by adding extensive experiments with non-linear deep models.
The lack of language experiments was resolved through the inclusion of a new language task, demonstrating that the proposed method generalizes beyond vision.
Questions about the real/synthetic data ratio and the notion of diversity were addressed via additional ablation studies and clearer definitions, supported by new empirical analyses.

Concerns regarding novelty and positioning relative to prior work, in particular Firdoussi et al. (2025), were addressed through an expanded discussion that clearly distinguishes the problem settings and contributions.
Skepticism about the irrelevance of mean shift and the chosen scaling was resolved through additional theoretical explanations and clarifications in the revised text.
Methodological concerns about the greedy selection strategy were addressed by comparing it to alternative multi-sample selection procedures, which showed similar performance.
Finally, questions about weighted mixtures of real and synthetic data and the shared-parameter assumption were addressed through theoretical extensions and additional discussion.

**Reviewer Scores:**

The reviewers who provided substantive evaluations appear to have had their main concerns adequately addressed by the rebuttal, and have reached a level of satisfaction.

---

### Decision · Program_Chairs · 2026-01-26

Accept (Oral)